# A Multi-Fidelity Control Variate Approach for Policy Gradient Estimation

**Xinjie Liu**[†]                                                  *xinjie-liu@utexas.edu*
*The University of Texas at Austin, Austin, TX, USA*

**Cyrus Neary**[†]                                                *cyrus.neary@ubc.ca*
*The University of British Columbia, Vancouver, BC, Canada*

**Kushagra Gupta**                                            *kushagrag@utexas.edu*
*The University of Texas at Austin, Austin, TX, USA*

**Wesley A Suttle**                                    *wesley.a.suttle.ctr@army.mil*
*DEVCOM Army Research Laboratory, Adelphi, MD, USA*

**Christian Ellis**                              *christian.ellis@austin.utexas.edu*
*The University of Texas at Austin, Austin, TX, USA*
*DEVCOM Army Research Laboratory, Adelphi, MD, USA*

**Ufuk Topcu**                                                    *utopcu@utexas.edu*
*The University of Texas at Austin, Austin, TX, USA*

**David Fridovich-Keil**                                          *dfk@utexas.edu*
*The University of Texas at Austin, Austin, TX, USA*

**Reviewed on OpenReview:** *https://openreview.net/forum?id=zAoOL7Dcqt*

## Abstract

Many reinforcement learning (RL) algorithms are impractical for deployment in operational systems or for training with computationally expensive high-fidelity simulations, as they require large amounts of data. Meanwhile, low-fidelity simulators—such as reduced-order models, heuristic reward functions, or generative world models—can cheaply provide useful data for RL training, even if they are too coarse for direct sim-to-real transfer. We propose *multi-fidelity policy gradients (MFPGs)*, an RL framework that mixes a small amount of data from the target environment with a control variate formed from a large volume of low-fidelity simulation data to construct an unbiased, variance-reduced estimator for on-policy policy gradients. We instantiate the framework by developing a practical, multi-fidelity variant of the classical REINFORCE algorithm. We show that under standard assumptions, the MFPG estimator guarantees asymptotic convergence of multi-fidelity REINFORCE to locally optimal policies in the target environment, and achieves faster finite-sample convergence rates compared to training with high-fidelity data alone. We evaluate the MFPG algorithm across a suite of simulated robotics benchmark tasks in scenarios with limited high-fidelity data but abundant off-dynamics, low-fidelity data. In our baseline comparisons, for scenarios where low-fidelity data are neutral or beneficial and dynamics gaps are mild to moderate, MFPG is, among the evaluated off-dynamics RL and low-fidelity-only approaches, *the only* method that consistently achieves statistically significant improvements in mean performance over a baseline trained solely on high-fidelity data. When low-fidelity data become harmful, MFPG exhibits the strongest robustness against performance degradation

---

[†]Equal contribution.
  Project website: `https://xinjie-liu.github.io/mfpg-rl/`

among the evaluated methods, whereas strong off-dynamics RL methods tend to exploit low-fidelity data aggressively and fail substantially more severely. An additional experiment in which the high- and low-fidelity environments are assigned anti-correlated rewards shows that MFPG can remain effective even when the low-fidelity environment exhibits reward misspecification. Thus, MFPG not only offers a *reliable* and *robust* paradigm for exploiting low-fidelity data, e.g., to enable efficient sim-to-real transfer, but also provides a principled approach to managing the trade-off between policy performance and data collection costs.

# 1 Introduction

Reinforcement learning (RL) algorithms offer significant capabilities in systems that work with unknown, or difficult-to-specify, dynamics and objectives. The flexibility and performance of RL algorithms have led to their adoption in applications as diverse as controlling plasma configurations in nuclear fusion reactors (Degrave et al., 2022), piloting high-speed aerial vehicles (Kaufmann et al., 2023), training reasoning capabilities in large language models (Shao et al., 2024), and searching large design spaces for automated discovery of new molecules (Bengio et al., 2021; Ghugare et al., 2023). However, in many such applications, datasets must be gathered from operational systems or from *high-fidelity* simulations. This requirement acts as a significant barrier to the development and deployment of RL policies: excessive interactions with operational systems are often either infeasible or unsafe, and generating simulated datasets for RL can be prohibitively expensive unless the simulations are both cheap to run and carefully designed to minimize the sim-to-real gap.

On the other hand, *low-fidelity* simulation tools capable of cheaply generating large volumes of data are often available. For example, reduced-order models, linearized dynamics, heuristic reward functions, and generative world models all output useful information for RL, even when approximations of the target dynamics and rewards are very coarse.

Towards enabling the training and deployment of RL policies when expensive, high-fidelity samples are scarce, we develop a novel *multi-fidelity* RL algorithm. The proposed algorithm mixes data from the target environment with data generated by lower-fidelity simulations to improve the sample efficiency of high-fidelity data. The proposed framework not only offers a novel paradigm for sim-to-real transfer, but also provides a principled approach to managing the trade-off between policy performance and data collection costs.

More specifically, we present *multi-fidelity policy gradients (MFPGs)*, an RL framework that mixes a small amount of data from the target environment with a control variate formed from a large volume of low-fidelity data to construct an unbiased, variance-reduced estimator for on-policy policy gradients. We further propose a practical algorithm that instantiates the MFPG estimator with the classical on-policy algorithm REINFORCE (Williams, 1992).

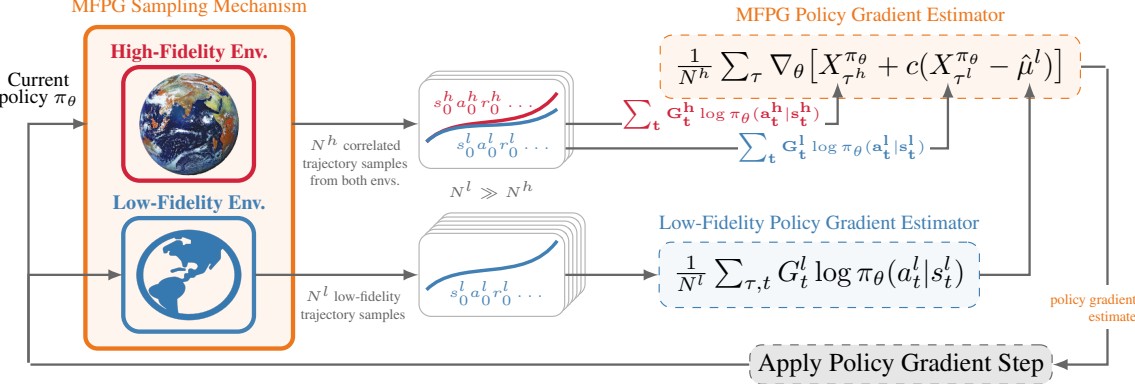

Figure 1: The proposed multi-fidelity policy gradient (MFPG) framework. At each policy update step, MFPG combines a small amount of data from the target (high-fidelity) environment with a large volume of low-fidelity simulation data, thereby forming an unbiased, reduced-variance estimator for the policy gradient.

Figure 1 illustrates the proposed approach. At each policy update step, the algorithm begins by sampling a relatively small number of trajectories from the target environment, which may correspond to real-world hardware or to a high-fidelity simulation. We then propose a method for sampling trajectories from the low-fidelity environments such that the resulting action likelihoods are highly correlated with those from the high-fidelity trajectories. This method hinges on a nuanced approach to action sampling and is critical to the success of our approach. Next, the algorithm uses the low-fidelity environments to sample a much larger quantity of uncorrelated trajectories, which it uses alongside the previously sampled trajectories to compute an unbiased estimate of the policy gradient to be applied. So long as the random variables corresponding to the policy gradient updates between the high and low-fidelity environments are correlated, the approach is guaranteed to reduce the variance of the policy gradient estimates.

Theoretically, we analyze the proposed MFPG estimator for the REINFORCE algorithm and prove that, under standard assumptions, the MFPG estimator guarantees asymptotic convergence to a locally optimal policy (satisfying first-order optimality conditions) in the *target* (high-fidelity) environment. Moreover, when high- and low-fidelity samples exhibit nonzero correlation, MFPG achieves a faster finite-sample convergence rate than using high-fidelity samples alone.

Empirically, we evaluate the proposed algorithm on a suite of benchmark robotic control tasks where the high- and low-fidelity environments differ in transition dynamics, and additionally on a task where the low-fidelity reward is the negative (anti-correlated) version of the high-fidelity reward. We use 20 random seeds per setting and non-parametric bootstrap confidence intervals to assess statistical significance. In high-fidelity sample-scarce regimes, the results reveal three key insights: i) When the dynamics gap is mild, MFPG can substantially reduce the variance of policy gradient estimates compared to standard variance reduction via state-value baseline subtraction. ii) MFPG provides a *reliable* and *robust* way to exploit low-fidelity data. In our baseline comparison experiments, for scenarios where low-fidelity data are neutral or beneficial and dynamics gaps are mild to moderate, MFPG is, among the evaluated off-dynamics RL and low-fidelity-only approaches, *the only* method that consistently and statistically significantly improves the mean performance over a baseline trained solely on high-fidelity data. When low-fidelity data become harmful, MFPG exhibits the strongest robustness against performance degradation among the evaluated off-dynamics RL and low-fidelity-only methods. Strong off-dynamics RL methods tend to exploit low-fidelity data more aggressively, and can achieve high performance at times, but can also fail substantially more severely. iii) MFPG can remain effective even in the presence of reward misspecification in the low-fidelity samples.

## 2 Related Work

This section provides a concise overview of three lines of related work: (i) variance reduction in RL via control variates; (ii) multi-fidelity RL and fine-tuning using high-fidelity data for sim-to-real transfer; and (iii) off-dynamics RL.

**Variance Reduction in RL via control variates.** The method of control variates (CVs) is commonly used for variance reduction in Monte Carlo estimation (Owen, 2013). In RL, particularly in policy gradient (PG) methods (Williams, 1992), there is a long history of using CVs to reduce variance and accelerating learning, e.g., subtracting constant reward baselines (Sutton, 1984; Williams, 1988; Dayan, 1991; Williams, 1992) or state value functions (Weaver & Tao, 2001; Greensmith et al., 2004; Peters & Schaal, 2006; Zhao et al., 2011) from Monte Carlo returns. Even though state values are known to not be optimal for variance reduction (Greensmith et al., 2004), they are also central to modern algorithms, e.g., actor-critic methods (Schulman et al., 2017). More recently, additional CVs baselines have been studied, such as vector-form CVs (Zhong et al., 2021), multi-step variance reduction along trajectories (Pankov, 2018; Cheng et al., 2020) and their more general form (Huang & Jiang, 2020), and state-action dependent baselines (Gu et al., 2017; Liu et al., 2018; Grathwohl et al., 2018; Wu et al., 2018), which offer better variance reduction under certain conditions (Tucker et al., 2018). The literature on using CVs for policy gradients almost exclusively focuses on single-environment settings. In this work, we consider the problem of fusing data from multi-fidelity environments to construct low-variance estimators, cf. Section 4. Moreover, the proposed multi-fidelity approach can also be readily combined with existing single-fidelity CVs for further improved performance.

To the best of our knowledge, the only existing work leveraging multi-fidelity CVs for RL is (Khairy & Balaprakash, 2024). Our approach differs in two main ways. First, we propose a multi-fidelity PG algorithm for Markov decision processes (MDPs) with either continuous or discrete state and action spaces, whereas Khairy & Balaprakash (2024) estimate state-action values in tabular MDPs without function approximation. Second, we propose a novel policy reparameterization trick that enables the sampling of correlated trajectories across multi-fidelity environments. Crucially, this sampling scheme improves the variance reduction of the proposed MFPG algorithm, and it directly supports continuous settings without restricting the MDPs. By contrast, Khairy & Balaprakash (2024) require action sequences to be matched across multi-fidelity environments, which requires additional assumptions about the MDP structure and transition dynamics.

**Multi-fidelity RL and fine-tuning using high-fidelity data.** Multi-fidelity methods for optimization, uncertainty propagation, and inference are well-studied in computational science and engineering, where it is often the case that multiple models for a problem of interest are available (Peherstorfer et al., 2018). Such methods provide principled statistical techniques to manage the computational costs of Monte Carlo simulations, without introducing unwanted biases. However, the adoption of such techniques by RL algorithms is significantly more limited. Two bodies of literature which are closely related to this work are multi-fidelity RL (Cutler et al., 2015) and fine-tuning RL policies using limited target-domain data, e.g., in sim-to-real transfer (Smith et al., 2022). Multi-fidelity RL aims to design *training curricula*, i.e., decide when to train in each simulation fidelities, in order to train highly-performant policies for the highest-fidelity environment with minimum simulation costs. Typically, training begins in the lowest-fidelity simulators, with proposed mechanisms to decide when to transition to other fidelities based on estimated uncertainty (Suryan et al., 2017; Cutler et al., 2015; Agrawal & McComb, 2024; Ryou et al., 2024; Bhola et al., 2023) in predicted actions, state-action values, dynamics, rewards, and/or information gain (Marco et al., 2017). Similarly, in sim-to-real and transfer learning settings (Taylor & Stone, 2009; Zhao et al., 2020; Tang et al., 2024; Da et al., 2025; Niu et al., 2024), the objective is to train a highly-performant policy for the real world using simulation data, sometimes supplemented with a small amount of real-world data. Many approaches leverage models or value functions trained in simulation (source domain) to bootstrap fine-tuning in the real world (target domain) (Rusu et al., 2017; Arndt et al., 2020; Taylor et al., 2007; Smith et al., 2022), guide real-world exploration (Yin et al., 2025), or bidirectionally align state distributions between simulation and real-world agents to improve sample efficiency. Real-world data can also be used to refine the simulator itself (Abbeel et al., 2006; Chebotar et al., 2019; Ramos et al., 2019). The works above introduce new training paradigms and exploration strategies with data from different domains. In contrast, our approach proposes a new policy gradient estimator that incorporates cross-domain data. The proposed approach can be seamlessly integrated into existing paradigms that involve fine-tuning within target domains.

**Off-dynamics RL.** A particularly important setting is when multi-fidelity environments exhibit mismatched transition dynamics. Classical approaches such as system identification (Ljung, 1998) and domain randomization (Tobin et al., 2017) address this problem from a dynamics modeling perspective, but typically require either access to a high-fidelity dynamics model or a pre-specified distribution over dynamics parameters. Other works (Desai et al., 2020b;a) instead learn action transformations to compensate for transitions produced by low-fidelity dynamics.

Another line of work tackles the off-dynamics RL problem from a policy adaptation perspective, i.e., modifying policy learning with samples from multiple domains. DARC (Eysenbach et al.) estimates importance weights for low-fidelity transitions—via a pair of learned classifiers modeling the likelihood ratio under high- and low-fidelity dynamics—and uses these weights to augment the low-fidelity rewards. Guo et al. (2024) extend this idea with generative adversarial imitation learning. Similar in spirit to DARC (Eysenbach et al.), more recent works estimate the dynamics mismatch by comparing observed low-fidelity transitions with a corresponding high-fidelity prediction—either through explicit target dynamics models to evaluate state-value consistency (Xu et al., 2023) or through latent representation learning (Lyu et al., 2024a). The estimated dynamics gap is then used to filter out unrealistic low-fidelity transitions (Xu et al., 2023) or to augment the low-fidelity rewards by penalizing less likely transitions (Lyu et al., 2024a). Van et al. (2024) introduces a mechanism that re-weights and extrapolates low-fidelity transitions while augmenting rewards, in order to mitigate deficient low-fidelity data support under large dynamics gaps. A related line of work

extends these ideas to *offline* RL, leveraging either online low-fidelity simulation data (Niu et al., 2022; 2025) or a large offline low-fidelity dataset (Liu et al., 2022; Lyu et al., 2025; Guo et al., 2025; Anonymous, 2025) to complement a small high-fidelity dataset.

In this work, we focus on the *online*, on-policy setting, where the agent has limited access to interact with the high-fidelity (target) environment but can collect abundant low-fidelity samples for each policy update. We benchmark our proposed MFPG algorithm against DARC (Eysenbach et al.) and PAR (Lyu et al., 2024a). A key distinction between MFPG and most existing off-dynamics RL methods lies in their fundamental design philosophy. Prior approaches often rely on regularizing low-fidelity samples for policy learning (Eysenbach et al.; Xu et al., 2023; Lyu et al., 2024a). In contrast, MFPG explicitly *grounds* policy learning in high-fidelity samples, leveraging low-fidelity data solely as a variance-reduction mechanism. This difference has important theoretical implications. Existing methods (Eysenbach et al.; Xu et al., 2023; Lyu et al., 2024a) bound the global *suboptimality* of policies deployed in the high-fidelity (target) environment; the bounds can degrade as the dynamics gap grows—for example, when discrepancies in value functions (Xu et al., 2023) or learned representations (Lyu et al., 2024a) become more pronounced. DARC (Eysenbach et al.) further assumes that policies optimal in the target domain remain near-optimal in the source domain. By contrast, MFPG requires no such assumption: low-fidelity samples are used exclusively for variance reduction—our asymptotic guarantees ensure that the learned policy is locally *optimal* (up to first order) in the high-fidelity environment, regardless of the dynamics mismatch. This design principle enables us to establish faster convergence rates for MFPG with REINFORCE when the correlation between low- and high-fidelity samples is non-zero. Empirically, MFPG demonstrates greater robustness than DARC and PAR. Moreover, unlike prior methods designed specifically for dynamics mismatch, MFPG can also extend to settings with reward misspecification, as illustrated in Section 6.4.

## 3 Preliminaries

Our objective is to develop RL algorithms capable of leveraging data generated by multiple environments in order to efficiently learn a policy that achieves high performance in a target environment of interest.

**Modeling multi-fidelity environments.** We consider a *high-fidelity* environment and a *low-fidelity* environment, each modeled by a finite-horizon MDP. In particular, the high-fidelity environment is an MDP $\mathcal{M}^h = (S, A, \Delta_{s_I}, \gamma, p^h, R^h, T)$, which we assume represents either an accurate simulator of the target environment, or the target environment itself. Here, $S$ is the set of environment states, $A$ is its set of actions, $\Delta_{s_I}$ is an initial distribution over states, $\gamma \in [0, 1]$ is a discount factor, $p^h(s'|s, a)$ is the probability of transitioning to state $s'$ from state $s$ under action $a$, $r \sim R^h(s, a, s')$ defines the probability of observing a particular reward under a given state-action-state triplet, and $T$ denotes the finite time horizon length. Similarly, the low-fidelity environment is an MDP $\mathcal{M}^l = (S, A, \Delta_{s_I}, \gamma, p^l, R^l, T)$, whose transition dynamics $p^l$ and reward function $R^l$ differ from those of $\mathcal{M}^h$, and do not necessarily accurately represent the target environment.

**Assumptions on the multi-fidelity environments.** We assume that the cost of generating sample trajectories in $\mathcal{M}^h$ is significantly higher than that of generating trajectories in $\mathcal{M}^l$, whose cost we treat as negligible. This may be the case if $\mathcal{M}^h$ were to represent real-world hardware while $\mathcal{M}^l$ were to represent cheap simulations. In addition, the proposed MFPG approach requires that the low-fidelity simulator can be reset to user-specified states on demand, so that we can construct low-fidelity trajectories whose initial states match those of the high-fidelity rollouts and thereby obtain correlated trajectories.

Our objective is to learn a stochastic policy $\pi_\theta(a|s)$, parameterized by $\theta \in \mathbb{R}^d$, that achieves a high expected total reward in the high-fidelity environment $\mathcal{M}^h$. Fixing a policy $\pi_\theta$ defines a distribution over trajectories in both $\mathcal{M}^h$ and $\mathcal{M}^l$. We denote trajectories in $\mathcal{M}^h$ by $\tau^h = s_0^h, a_0^h, r_0^h, s_1^h, a_1^h, r_1^h, \ldots, s_T^h$, where $s_0^h \sim \Delta_{s_I}$, $a_t^h \sim \pi_\theta(\cdot|s_t^h)$, $s_{t+1}^h \sim p^h(\cdot|s_t^h, a_t^h)$, and $r_t^h \sim R^h(s_t^h, a_t^h, s_{t+1}^h)$. Similarly, we use $\tau^l$ to denote trajectories sampled in $\mathcal{M}^l$.

**Policy gradients.** PG algorithms aim to maximize the performance measure $J(\theta) := \mathbb{E}_{\tau \sim \mathcal{M}(\pi_\theta)}[R(\tau)]$—the expected total reward along trajectories $\tau$ sampled from environment $\mathcal{M}$ under policy $\pi_\theta$. They do so by using stochastic estimates of the policy gradient $\nabla_\theta J(\theta)$ to perform gradient ascent directly on the

policy parameters (Sutton & Barto, 2018). For example, the REINFORCE algorithm (Williams, 1988; 1992) uses Monte Carlo estimates of $\mathbb{E}_{\tau \sim \mathcal{M}(\pi_\theta)}[\nabla_\theta X_\tau^{\pi_\theta}]$ to estimate the policy gradient, where $X_\tau^{\pi_\theta} := \frac{1}{T} \sum_{t=0}^{T-1} G_t \log \pi_\theta(a_t|s_t)$ is a random variable defining the contribution of each trajectory $\tau$ to the overall policy gradient. Here, $a_t$, $s_t$, and $G_t = \sum_{n=t}^{T-1} \gamma^{n-t} r_n$, denote the selected action, the state, and the reward-to-go at time $t$ in trajectory $\tau$, respectively.

Different policy gradient algorithms use different expressions for $X_\tau^{\pi_\theta}$ when estimating the policy gradient (e.g., $G_t$ may be replaced with an advantage estimate, or $X_\tau^{\pi_\theta}$ may be entirely replaced by a surrogate per-trajectory gradient estimate, as is done in PPO (Schulman et al., 2017)). However, we note that the overall structure of many on-policy algorithms remains the same: use the current policy to sample trajectories in the environment; use the rewards, actions, and states along these trajectories to compute a variable of interest $X_\tau^{\pi_\theta}$; finally, average the gradients of these sampled random variables to estimate $\nabla_\theta J(\theta)$.

## 4 Multi-Fidelity Policy Gradients

This section introduces our MFPG framework, a mechanism to draw correlated trajectories from multi-fidelity environments, and the corresponding multi-fidelity REINFORCE algorithm.

**Iteration notation.** We consider an iterative policy gradient procedure with parameters $\theta_k \in \mathbb{R}^d$ at iteration $k \in \mathbb{Z}^+$, and write $\pi_{\theta_k}(a \mid s)$ for the corresponding policy. When discussing a single iteration and a fixed parameter vector, all quantities are understood to be evaluated at that iteration, and we omit the subscript $k$ for readability, writing simply $\theta$ and $\pi_\theta$. All subsequent expressions that omit the iteration index should therefore be interpreted as taken at an arbitrary but fixed iteration $k$, and hence apply without loss of generality to every iteration of the algorithm.

**Multi-fidelity policy gradient estimators via control variates.** Because our objective is to optimize the policy performance in the high-fidelity environment $\mathcal{M}^h$, during each step of the policy gradient algorithm we must estimate $\nabla_\theta \mathbb{E}_{\tau^h \sim \mathcal{M}^h(\pi_\theta)}[X_{\tau^h}^{\pi_\theta}]$ from a potentially limited number $N^h$ of sampled high-fidelity trajectories $\tau^h$. In data-scarce settings, existing policy gradient methods can face the challenge of high variance of the gradient estimates (Greensmith et al., 2004). We aim to reduce the estimation variance for the PGs. In this work, we assume that we may also sample a relatively large number $N^l \gg N^h$ of trajectories $\tau^l$ from the low-fidelity environment $\mathcal{M}^l$. We use these low-fidelity samples to construct a so-called *control variate* $X_{\tau^l}^{\pi_\theta}$—a correlated auxiliary random variable whose expected value $\mu^l := \mathbb{E}_{\tau^l \sim \mathcal{M}^l(\pi_\theta)}[X_{\tau^l}^{\pi_\theta}]$ is known. Note that $X_{\tau^l}^{\pi_\theta}$ is obtained by applying the same trajectory functional $X_\tau^{\pi_\theta}(\cdot)$ to trajectories $\tau^l$ sampled from the low-fidelity environment $\mathcal{M}^l(\pi_\theta)$ under $\pi_\theta$. We then use the *control variates technique* (Nelson, 1987) to construct an unbiased, reduced-variance estimator for $\nabla_\theta \mathbb{E}_{\tau^h \sim \mathcal{M}^h(\pi_\theta)}[X_{\tau^h}^{\pi_\theta}]$.

Specifically, we construct a new random variable $Z^{\pi_\theta}(c) := X_{\tau^h}^{\pi_\theta} + c(X_{\tau^l}^{\pi_\theta} - \mu^l)$ with coefficient $c \in \mathbb{R}$. The next lemma formalizes the unbiasedness and variance-reduction properties of this construction; its proof follows standard control variate arguments and is provided in Section B.

**Lemma 1** (Unbiasedness and variance reduction of the multi-fidelity control variate estimator). *Let $X_{\tau^h}^{\pi_\theta}$ and $X_{\tau^l}^{\pi_\theta}$ be the high- and low-fidelity random variables defined above, and let $\mu^l := \mathbb{E}_{\tau^l \sim \mathcal{M}^l(\pi_\theta)}[X_{\tau^l}^{\pi_\theta}]$. For any coefficient $c \in \mathbb{R}$, define*

$$Z^{\pi_\theta}(c) := X_{\tau^h}^{\pi_\theta} + c(X_{\tau^l}^{\pi_\theta} - \mu^l). \tag{1}$$

*Then, $Z^{\pi_\theta}(c)$ is unbiased with respect to the high-fidelity random variable: $\mathbb{E}[Z^{\pi_\theta}(c)] = \mathbb{E}[X_{\tau^h}^{\pi_\theta}]$.*

*Moreover, by choosing the coefficient*

$$c^* = -\frac{Cov(X_{\tau^h}^{\pi_\theta}, X_{\tau^l}^{\pi_\theta})}{Var(X_{\tau^l}^{\pi_\theta})} = -\rho(X_{\tau^h}^{\pi_\theta}, X_{\tau^l}^{\pi_\theta})\frac{\sqrt{Var(X_{\tau^h}^{\pi_\theta})}}{\sqrt{Var(X_{\tau^l}^{\pi_\theta})}} \implies Var\big(Z^{\pi_\theta}(c^*)\big) = \big(1 - \rho^2(X_{\tau^h}^{\pi_\theta}, X_{\tau^l}^{\pi_\theta})\big)Var(X_{\tau^h}^{\pi_\theta}),$$

(2)

*where $\rho(\cdot, \cdot)$ is the Pearson correlation coefficient between the two random variables. In particular, whenever $\rho(X_{\tau^h}^{\pi_\theta}, X_{\tau^l}^{\pi_\theta}) \neq 0$, the multi-fidelity control variate estimator $Z^{\pi_\theta}(c^*)$ has strictly smaller variance than $X_{\tau^h}^{\pi_\theta}$ alone.*

We estimate $\mu^l$ using the $N^l$ low-fidelity trajectory samples, and we provide a method to jointly sample correlated values $X_{\tau^h}^{\pi_\theta}$ and $X_{\tau^l}^{\pi_\theta}$ (described below). In the remainder of the paper, we take $Z^{\pi_\theta} := Z^{\pi_\theta}(c^*)$ as the ideal multi-fidelity random variable. In practice, $c^*$ is unknown and must be estimated from samples. We estimate $c^*$ using the sampled trajectories from $\mathcal{M}^h$ and $\mathcal{M}^l$, reusing samples from previous gradient steps by maintaining moving-average statistics. Specifically, at iteration $k$, given the current policy $\pi_{\theta_k}$, we first compute *batch* estimates for correlation $\hat{\rho}_k^{\text{batch}}(X_{\tau^h}^{\pi_{\theta_k}}, X_{\tau^l}^{\pi_{\theta_k}})$ and standard deviations $\sqrt{\widehat{\text{Var}}_k^{\text{batch}}(X_{\tau^h}^{\pi_{\theta_k}})}, \sqrt{\widehat{\text{Var}}_k^{\text{batch}}(X_{\tau^l}^{\pi_{\theta_k}})}$ using newly sampled trajectories at iteration $k$. Let $\hat{\rho}_{k-1}(X_{\tau^h}^{\pi_{\theta_{k-1}}}, X_{\tau^l}^{\pi_{\theta_{k-1}}}), \sqrt{\widehat{\text{Var}}_{k-1}(X_{\tau^h}^{\pi_{\theta_{k-1}}})}, \text{and} \sqrt{\widehat{\text{Var}}_{k-1}(X_{\tau^l}^{\pi_{\theta_{k-1}}})}$ denote the running estimates from the previous iteration. With moving–average coefficient $\eta_{\text{ma}} \in (0, 1)$, we update

$$\hat{\rho}_k(X_{\tau^h}^{\pi_{\theta_k}}, X_{\tau^l}^{\pi_{\theta_k}}) = \eta_{\text{ma}}\hat{\rho}_{k-1}(X_{\tau^h}^{\pi_{\theta_{k-1}}}, X_{\tau^l}^{\pi_{\theta_{k-1}}}) + (1 - \eta_{\text{ma}})\hat{\rho}_k^{\text{batch}}(X_{\tau^h}^{\pi_{\theta_k}}, X_{\tau^l}^{\pi_{\theta_k}}),$$
$$\sqrt{\widehat{\text{Var}}_k(X_{\tau^h}^{\pi_{\theta_k}})} = \eta_{\text{ma}}\sqrt{\widehat{\text{Var}}_{k-1}(X_{\tau^h}^{\pi_{\theta_{k-1}}})} + (1 - \eta_{\text{ma}})\sqrt{\widehat{\text{Var}}_k^{\text{batch}}(X_{\tau^h}^{\pi_{\theta_k}})},$$
$$\sqrt{\widehat{\text{Var}}_k(X_{\tau^l}^{\pi_{\theta_k}})} = \eta_{\text{ma}}\sqrt{\widehat{\text{Var}}_{k-1}(X_{\tau^l}^{\pi_{\theta_{k-1}}})} + (1 - \eta_{\text{ma}})\sqrt{\widehat{\text{Var}}_k^{\text{batch}}(X_{\tau^l}^{\pi_{\theta_k}})}.$$

(3)

We then form an estimate of the optimal control–variate coefficient as

$$\hat{c}_k^* = -\hat{\rho}_k(X_{\tau^h}^{\pi_{\theta_k}}, X_{\tau^l}^{\pi_{\theta_k}})\frac{\sqrt{\widehat{\text{Var}}_k(X_{\tau^h}^{\pi_{\theta_k}})}}{\sqrt{\widehat{\text{Var}}_k(X_{\tau^l}^{\pi_{\theta_k}})}},$$

(4)

and, when the iteration index is clear, we simply write $\hat{c}^*$.

At every policy gradient step $k$, given the current policy $\pi_{\theta_k}$, the proposed MFPG framework thus proceeds as follows: 1) Use policy $\pi_{\theta_k}$ to sample $N^h$ correlated trajectories from $\mathcal{M}^h$ and $\mathcal{M}^l$, as well as $N^l$ additional trajectories from $\mathcal{M}^l$. 2) Use the sampled trajectories to compute estimates $\hat{\mu}_k^l$, $\hat{c}_k^*$, and the correlated values of the random variables $X_{\tau^h}^{\pi_{\theta_k}}$ and $X_{\tau^l}^{\pi_{\theta_k}}$. 3) Compute the sampled values of $Z^{\pi_{\theta_k}}$. 4) Use the samples of $Z^{\pi_{\theta_k}}$ to compute an unbiased, reduced-variance estimate of the policy gradient $\nabla_{\theta_k}\mathbb{E}[Z^{\pi_{\theta_k}}] \approx \frac{1}{N^h}\sum_\tau \nabla_{\theta_k}\big[X_{\tau^h}^{\pi_{\theta_k}} + \hat{c}_k^*(X_{\tau^l}^{\pi_{\theta_k}} - \hat{\mu}_k^l)\big]$.

**Sampling correlated trajectories from the multi-fidelity environments.** Note that for the random variables $X_{\tau^h}^{\pi_\theta}$ and $X_{\tau^l}^{\pi_\theta}$ to be correlated, they must share an underlying probability space $(\Omega, \mathcal{F}, \mathbb{P})$. In other words, every outcome $\omega \in \Omega$ from this probability space should uniquely define a high-fidelity $\tau^h(\omega)$ and low-fidelity $\tau^l(\omega)$ trajectory, as well as the corresponding values of the random variables $X_{\tau^h}^{\pi_\theta}(\omega)$ and $X_{\tau^l}^{\pi_\theta}(\omega)$. We emphasize that the global outcome $\omega$ is introduced purely as an analytical device. In practice, sampling and coupling the joint outcome $\omega$ across the high- and low-fidelity MDPs is infeasible, since the stochasticity in environment transitions and rewards is typically not directly controllable. Accordingly, our MFPG algorithm only couples the policy randomness and the initial-state randomness across fidelities, while treating transition and reward outcomes as independent, as we will describe below. A careful formulation of the joint probability space is nevertheless useful for conceptual clarity and for describing how to construct correlated trajectories in $\mathcal{M}^h$ and $\mathcal{M}^l$ under the same policy $\pi_\theta$.

Informally, when sampling trajectories from $\mathcal{M}^h$ and $\mathcal{M}^l$, there are six sources of stochasticity: the stochasticity introduced by the policy $\pi_\theta$, that introduced by the initial state distribution $\Delta_{s_I}$, the stochasticity

introduced by the high-fidelity and low-fidelity transition dynamics $p^h$ and $p^l$, and finally the stochasticity introduced by the two reward functions $R^h$ and $R^l$. Note that the stochasticity introduced by $\pi_\theta$ is implemented by the agent, and is thus under the control of the algorithm in the sense that the same policy random outcome $\omega^\pi$ may be used to generate actions from $\pi_\theta$ in both $\mathcal{M}^h$ and $\mathcal{M}^l$. We do so via a policy reparameterization trick (see below). Similarly, the algorithm may fix the initial states in the low-fidelity environment to match those observed from the high-fidelity samples. On the other hand, the transition dynamics and environment rewards are generated by independent sources of stochasticity in the different environments; we do not assume, nor require, any shared transition or reward outcome across fidelities.

We accordingly define the outcome set of the probability space as $\Omega = \Omega_{\Delta_{s_I}} \times \Omega_\pi \times \Omega_{p^h} \times \Omega_{p^l} \times \Omega_{R^h} \times \Omega_{R^l}$. Here, each outcome $\omega^{\Delta_{s_I}} \in \Omega_{\Delta_{s_I}}$ defines a particular shared initial state. Meanwhile, each policy outcome $\omega^\pi \in \Omega_\pi$ corresponds to a sequence $\omega^\pi = \omega_1^\pi, \omega_2^\pi, \ldots, \omega_T^\pi$ that dictates the random sequence of actions selected by the policy $\pi_\theta$ in both environments. In practice, this policy outcome sequence is realized as a sequence of per-timestep action-noise samples used to reparameterize the policy's action sampling at each time step (see the discussion on policy reparameterization below). Conceptually, given the outcome $\omega_t^\pi$ at any timestep $t$ of a trajectory, the policy should deterministically output action $a_t^h = \pi_\theta(s_t^h, \omega^\pi)$ in the high-fidelity environment, and action $a_t^l = \pi_\theta(s_t^l, \omega^\pi)$ in the low-fidelity environment. Note that this does not necessarily imply that $a_t^h = a_t^l$, due to a potential difference in states $s_t^h$ and $s_t^l$ that stems from the different dynamics. Similarly, each outcome $\omega^{p_h} \in \Omega_{p^h}$ is a sequence $\omega^{p_h} = \omega_0^{p_h}, \omega_1^{p_h}, \ldots, \omega_T^{p_h}$ dictating the outcomes $s_{t+1} = p^h(s_t^h, a_t^h, \omega^{p_h})$ of the stochastic transitions in the high-fidelity environment. However, unlike the policy outcome sequence $\omega^\pi \in \Omega^\pi$, the transition outcome sequences $\omega^{p_h} \in \Omega^{p_h}$ and $\omega^{p_l} \in \Omega^{p_l}$, and the reward outcome sequences $\omega^{R_h} \in \Omega^{R_h}$ and $\omega^{R_l} \in \Omega^{R_l}$, are not shared between the high and low fidelity environments. Thus, only the initial-state outcomes $\omega^{\Delta_{s_I}}$ and the policy outcomes $\omega^\pi$ are coupled across fidelities in our algorithm, while the transition and reward outcomes $\omega^{p_h}, \omega^{p_l}, \omega^{R_h}, \omega^{R_l}$ remain independent.

Formulating these separate outcome sets is conceptually helpful. However, practically, we only explicitly sample values for the policy outcomes $\omega_0^\pi, \omega_1^\pi, \ldots, \omega_T^\pi$ (action-noise samples) in our implementation, which we describe next.

---

**Algorithm 1:** Correlated trajectory sampling.

**Input:** Current policy $\pi_\theta$, multi-fidelity environments $\mathcal{M}^h$ and $\mathcal{M}^l$.

**Output:** Sampled trajectories $\{\tau_i^h, \tau_i^l\}_{i=1}^{N^h}$.

1  TrajectoryList $\leftarrow$ EmptyList
2  **for** $i \in \{1, 2, \ldots, N^h\}$ **do**
3  $\quad$ $\omega_0^\pi \ldots \omega_T^\pi \sim SampleActionNoiseSequence()$
4  $\quad$ $s_0^h \sim \Delta_{s_I}; s_0^l \leftarrow s_0^h$
5  $\quad$ **for** $t \in \{0, \ldots, T-1\}$ **do**
6  $\quad\quad$ $a_t^h \leftarrow \pi_\theta(s_t^h, \omega_t^{\pi_\theta})$ // `reparameterization trick`
7
8  $\quad\quad$ $s_{t+1}^h \sim p^h(\cdot | s_t^h, a_t^h)$
9  $\quad\quad$ $r_t^h \sim R^h(s_t^h, a_t^h, s_{t+1}^h)$
10 $\quad$ **for** $t \in \{0, \ldots, T-1\}$ **do**
11 $\quad\quad$ $a_t^l \leftarrow \pi_\theta(s_t^l, \omega_t^{\pi_\theta})$ // `reparameterization trick`
12
13 $\quad\quad$ $s_{t+1}^l \sim p^l(\cdot | s_t^l, a_t^l)$
14 $\quad\quad$ $r_t^l \sim R^l(s_t^l, a_t^l, s_{t+1}^l)$
15 $\quad$ $\tau_i^h \leftarrow s_0^h, a_0^h, r_0^h, \ldots, s_T^h$
16 $\quad$ $\tau_i^l \leftarrow s_0^l, a_0^l, r_0^l, \ldots, s_T^l$
17 $\quad$ TrajectoryList.append($\{\tau_i^h, \tau_i^l\}$)
18 **return** *TrajectoryList*

---

**Correlated action sampling via policy distribution reparameterization.** In order to use the sampled outcomes $\omega_t^\pi$ to deterministically select an action under the parameterized policy $\pi_\theta$, we implement a technique inspired by the so-called *reparameterization trick* used in variational autoencoders (Kingma, 2013). In particular, in continuous action spaces, we draw $\omega_t^\pi \sim \mathcal{N}(0,1)$, and the policy $\pi_\theta(s_t, \omega_t^\pi)$ is trained to output a state-dependent mean and standard deviation which are used to transform $\omega_t^\pi$ into an action $a_t$. Meanwhile, in discrete action spaces, one can draw $\omega_t^\pi \sim \text{Uniform}(0,1)$ and apply the Gumbel-Max trick (Huijben et al., 2022) to sample $a_t$ according to the state-dependent probability distribution defined by the policy $\pi_\theta(s_t, \omega_t^\pi)$.

To summarize, Algorithm 1 outlines the sampling procedure. First, we use the pre-sampled policy action-noise sequence to roll out a high-fidelity trajectory $\tau^h$. Next, the initial state in the low-fidelity environment is fixed to match the initial state of $\tau^h$ and thereby effectively enforcing a common $\omega^{\Delta_{s_I}}$. Finally, we reuse the same sequence of action-noise samples to generate the low-fidelity trajectory $\tau^l$.

**Defining multi-fidelity variant of the REINFORCE algorithm.** To this point, we have defined a mechanism for sampling correlated trajectories $\tau^h$ and $\tau^l$ from multi-fidelity environments $\mathcal{M}^h$ and $\mathcal{M}^l$, as well as a framework for using said trajectories to construct reduced-variance estimators of the policy gradient from trajectory-dependent variables $X_{\tau^h}^{\pi_\theta}$ and $X_{\tau^l}^{\pi_\theta}$. With these elements of the MFPG framework in place, we may implement a multi-fidelity variant of the REINFORCE algorithm. We note that, in principle, MFPG can also be instantiated with other on-policy policy gradient methods, by replacing the way in which the value of $X_\tau^{\pi_\theta}$ is computed from sampled trajectories. In this work, we focus on multi-fidelity REINFORCE in order to ease theoretical analysis, leaving extensions to other algorithms for future work.

We implement a multi-fidelity variant of the REINFORCE algorithm (Williams, 1988; 1992) by defining $X_\tau^{\pi_\theta} := \frac{1}{T} \sum_{t=0}^{T-1} G_t \log \pi_\theta(a_t|s_t)$, as described above. Additionally, to match the commonly used variance-reduced variant of the REINFORCE algorithm (Peters & Schaal, 2006), we subtract state values estimated by a value network from the Monte Carlo returns, i.e., $X_\tau^{\pi_\theta} := \frac{1}{T} \sum_{t=0}^{T-1} (G_t - V_\phi(s_t)) \log \pi_\theta(a_t|s_t) = \frac{1}{T} \sum_{t=0}^{T-1} A_\phi(s_t, a_t) \log \pi_\theta(a_t|s_t)$. Here, $V_\phi(s_t), A_\phi(s_t, a_t)$ denote value and advantage functions estimated from samples. We sample trajectories from the high- and low-fidelity environments to compute Monte Carlo returns $G_t^h$ and $G_t^l$, respectively. We then use a shared value function $V_\phi$ learned from high-fidelity samples $(s_t^h, a_t^h, r_t^h, s_{t+1}^h)$, to compute state-value baselines that are subtracted from both $G_t^h$ and $G_t^l$. This is an implementation choice that simplifies training; moreover, subtracting any state-dependent baseline does not change the expectation of the REINFORCE gradient estimator (Williams, 1992). Alternatively, one can train separate value functions for the high- and low-fidelity Monte Carlo returns. Algorithm 2 in Section A summarizes the MFPG REINFORCE algorithm.

## 5 Theoretical Analysis of Multi-Fidelity Policy Gradients

We now analyze the convergence of multi-fidelity policy gradients for the REINFORCE algorithm. We employ the following assumptions on the high and low-fidelity environment MDPs $\mathcal{M}^h, \mathcal{M}^l$ and the parametrized policy $\pi_\theta$, which are standard in reinforcement learning literature.

**Assumption 1** (Bounded Rewards). The high- and low-fidelity reward functions $R^h, R^l$ are bounded, i.e., there exists $U_R^h, U_R^l > 0$ such that $|R^h(s,a)| \leq U_R^h, |R^l(s,a)| \leq U_R^l$, $\forall (s,a) \in \mathcal{S} \times \mathcal{A}$.

**Assumption 2** (Differentiable Policy). The policy function $\pi_\theta$ is differentiable with respect to $\theta$ everywhere, and the score function $\nabla_\theta \log \pi_\theta(a|s)$ exists everywhere.

**Assumption 3** (Bounded Score Functions). The score function $\nabla_\theta \log \pi_\theta$ is bounded, i.e., there exists $B_\Theta > 0$ such that $\|\nabla_\theta \log \pi_\theta(a|s)\| \leq B_\Theta$, $\forall \theta \in \mathbb{R}^d, (s,a) \in \mathcal{S} \times \mathcal{A}$.

**Assumption 4** (Lipschitz Score Functions). The score function $\nabla_\theta \log \pi_\theta$ is $L_\Theta$-Lipschitz, i.e., $\|\nabla_\theta \log \pi_{\theta_1}(a|s) - \nabla_\theta \log \pi_{\theta_2}(a|s)\| \leq L_\Theta \|\theta_1 - \theta_2\|$, $\forall \theta_1, \theta_2 \in \mathbb{R}^d, (s,a) \in \mathcal{S} \times \mathcal{A}$.

Assumption 1 is common in the policy gradient literature (Zhang et al., 2020; Bhatnagar et al., 2007). Assumptions 2 to 4 are also common in prior works investigating the convergence of various policy gradient methods (Papini et al., 2018; 2022; Pirotta et al., 2015; Zhang et al., 2020). Particularly, the commonly

used Gaussian and Boltzmann parameterized policies are known to satisfy these assumptions (Zhang et al., 2020). We also make the following assumption on the high-fidelity policy gradient estimate.

**Assumption 5** (High-Fidelity Policy Gradient Estimator)**.** The high-fidelity policy gradient estimator $\nabla_\theta X_{\tau^h}^{\pi_\theta}$ is unbiased and has bounded variance, i.e., $\mathbb{E}\left[\nabla_\theta X_{\tau^h}^{\pi_\theta}\right] = \nabla_\theta J(\theta)$, and there exists $\sigma > 0$ such that $\mathbb{E}\left[\|\nabla_\theta X_{\tau^h} - \nabla_\theta J(\theta)\|^2\right] \leq \sigma^2$.

Assumption 5 ensures that the high-fidelity policy gradient estimator is a "good" estimator, in the sense that the high-fidelity policy gradient would have learned a good policy if it had unrestricted access to high-fidelity data. Such an assumption is required to formally show convergence (see, e.g., Papini et al. (2018)). We note that Assumption 5 is an assumption on the underlying high-fidelity policy gradient estimator; we do not require any policy gradient estimates computed from finite samples to be exact. In particular, the classic Monte Carlo REINFORCE policy gradient estimator considered in this work satisfies this requirement (Sutton et al., 1999). Extending our analysis to actor–critic-style estimators is nontrivial and is discussed as a limitation and direction for future work in Section G.

Finally, in order for the proposed approach to be effective, there must be nonzero correlation between random variables computed from the high- and low-fidelity environments, i.e., the coefficient $c^*$ in Eq. (2) is nonzero. Hence, we make the following assumption.

**Assumption 6** (Nonzero Correlation)**.** The Pearson correlation coefficient between the high- and low-fidelity random variables is nonzero: there exists $\rho \in (0, 1]$ such that $\left|\mathrm{Cov}\left(X_{\tau^h}^{\pi_\theta}, X_{\tau^l}^{\pi_\theta}\right) / \sqrt{\mathrm{Var}\left(X_{\tau^h}^{\pi_\theta}\right)\mathrm{Var}\left(X_{\tau^l}^{\pi_\theta}\right)}\right| \geq \rho, \ \forall \ \theta \in \mathbb{R}^d$.

Assumption 6 can be interpreted as a comment on the quality of the low-fidelity estimator: while the low-fidelity estimator may be inaccurate and have return values that are incomparable to the high-fidelity due to different transition dynamics in the two environments, the assumption ensures that the lower fidelity still preserves some meaningful connection with the higher fidelity. This connection is captured by the correlation between the two fidelities.

Under these assumptions, we establish that the variance-reduced MFPG iterates lead to a faster decrease in policy gradient norms in comparison to the high-fidelity-only iterates, when employed for a finite horizon REINFORCE algorithm. Further, we establish the global asymptotic convergence of the MFPG to a first-order stationary point of the high-fidelity objective function. The proof is presented in Section C.

**Theorem 1.** *Under Assumptions 1 to 6, let $\{\theta_k^{MFPG}\}_{k\in\mathbb{Z}^+}$ and $\{\theta_k^h\}_{k\in\mathbb{Z}^+}$ be the sequences of policy parameters generated by running the REINFORCE algorithm with MFPG and high-fidelity-only policy gradient estimators, respectively. Then, for a problem with time horizon $T$, after $N$ iterations, with stepsizes for both iterates taken to be a sequence $\alpha_k$ satisfying $\sum_k \alpha_k = \infty, \sum_k \alpha_k^2 < \infty$, the policy gradient $\nabla J(\theta)$ evaluated in the high-fidelity (target) environment is bounded as follows:*

$$\min_{k\in[N]} \mathbb{E}\left[\|\nabla J(\theta_k^{MFPG})\|^2\right] \leq \frac{J(\theta^*) - J(\theta_1) + \left(1 - \rho^2\right)\frac{\sigma^2 L_T}{2}\sum_{k=1}^N \alpha_k^2}{\sum_{k=1}^N\left(\alpha_k - \frac{\alpha_k^2 L_T}{2}\right)},$$

*where $L_T$ is the Lipschitz constant of the high-fidelity policy gradient, established in Lemma 2. Moreover, we recover the corresponding rate for the high-fidelity only policy iterates $\{\theta_k^h\}_{k\in\mathbb{Z}^+}$ by substituting $\rho = 0$. Finally, we have that $\lim_{k\to\infty} \mathbb{E}\left[\|\nabla J(\theta_k^{MFPG})\|\right] = 0$ almost surely.*

**Proof Sketch.** The non-asymptotic bounds leverage the Lipschitzness of the high-fidelity-only policy gradient estimator, which we establish in Lemma 2 in Section C, while the asymptotic convergence proof utilizes the supermartingale convergence theorem (Robbins & Siegmund, 1971) as follows. We first show that the norm of the MFPG estimator is bounded, then construct a specific non-negative auxiliary random variable that we show is a supermartingale and apply the supermartingale theorem to establish the result. The complete proof is presented in Section C.

Theorem 1 establishes that the minimum (high-fidelity) policy gradient norm after $k$ iterations will be smaller for the MFPG REINFORCE algorithm compared to the corresponding high-fidelity-only algorithm.

Moreover, the *improvement that the MFPG algorithm brings depends on how correlated the high and low-fidelity environments are.* Note that, as $k$ increases, the right side of the inequality in Theorem 1 diminishes to 0, due to the stepsize conditions. Further, the theorem implies that, asymptotically, the MFPG estimator finds a policy that satisfies the first-order optimality conditions for the high-fidelity objective.

## 6 Experiments

This section empirically evaluates the proposed MFPG algorithm in a suite of simulated benchmark RL tasks. In high-fidelity data-scarce regimes, the experimental results demonstrate the following **key insights**[1]:

- (**I1, Variance reduction**) When the multi-fidelity dynamics gap is mild, MFPG can substantially reduce the variance of PG estimates compared to standard variance reduction via state-value baseline subtraction.

- (**I2, Reliability and robustness**) MFPG provides a reliable and robust way to exploit low-fidelity data. In our baseline comparisons in Section 6.3, for scenarios where low-fidelity data are neutral or beneficial (Hopper-gravity, Walker2d-friction) and dynamics gaps are mild to moderate ($0.5\times$ to $2.0\times$), MFPG is, among the evaluated off-dynamics RL and low-fidelity-only approaches, **the only** method that consistently and statistically significantly improves the mean performance over a baseline trained solely on high-fidelity data in **8 out of 8** scenarios. When low-fidelity data become harmful, MFPG exhibits the strongest robustness against performance degradation among the evaluated off-dynamics RL and low-fidelity-only methods. Strong off-dynamics RL methods tend to exploit low-fidelity data more aggressively, and can achieve higher performance than MFPG at times (e.g., in Walker2d) but can also fail substantially more severely (e.g., in HalfCheetah).

- (**I3, Misspecified reward**) MFPG can remain effective even when low-fidelity samples exhibit reward misspecification, e.g., when the low-fidelity reward is the negative (anti-correlated) version of the high-fidelity reward.

### 6.1 Experiment Setup

**Task settings.** We evaluate our approach on the off-dynamics RL benchmark introduced by Lyu et al. (2024b), focusing on three standard MuJoCo tasks (Todorov et al., 2012): (i) Hopper-v3, (ii) Walker2d-v3, and (iii) HalfCheetah-v3. In each task, the low-fidelity environment corresponds to the original dynamics, while the high-fidelity variants include changes in gravity or friction. Gravity and friction are varied across five levels $\{0.5\times, 0.8\times, 1.2\times, 2.0\times, 5.0\times\}$; the intermediate $0.8\times$ and $1.2\times$ levels are not part of the original benchmark but are introduced here to examine settings with mild dynamics shifts. In total, the task set spans 15 distinct settings. In Sections 6.3 to 6.5, we evaluate each setting with 20 random seeds per method.

We provide additional results on 24 further task settings in Section E, evaluated with 5 random seeds per method. Due to the small number of seeds, these additional results are primarily used to illustrate the consistency of qualitative trends, and we do not draw formal statistical conclusions from them.

We note that the benchmark (Lyu et al., 2024b) also includes the Ant-v3 environment. Unlike the other three tasks, the observation space and reward function in Ant-v3 depend explicitly on contact forces, which cannot be reliably reset to user-specified values, as required by our MFPG sampling scheme. Due to this difficulty in matching initial states across fidelities, we focus our evaluations on the other three tasks described above.

**Baselines.** We compare MFPG against the following baselines: **High-Fidelity Only**, which trains solely on the limited high-fidelity samples; **Low-Fidelity Only ($100\times$)**, which trains on abundant low-fidelity data ($100\times$ more samples) and is evaluated in the high-fidelity environment; **DARC** (Eysenbach et al.), which learns domain classifiers to estimate dynamics gaps for the sampled low-fidelity transitions and uses the estimated gaps to augment low-fidelity rewards; and **PAR** (Lyu et al., 2024a), which augments low-fidelity rewards using dynamics mismatch estimated in a learned latent space. Please refer to Section F for a more detailed description of the baselines.

---

[1]Code: https://xinjie-liu.github.io/mfpg-rl/

We instantiate the MFPG framework with the on-policy policy gradient algorithm REINFORCE (Williams, 1992) with state value subtraction. DARC (Eysenbach et al.) and PAR (Lyu et al., 2024a) were originally demonstrated with the off-policy soft actor-critic algorithm (Haarnoja et al., 2018). To isolate the effect of their multi-fidelity mechanisms from the backbone algorithm and enable fair comparison, we adapt both PAR and DARC to the REINFORCE backbone. We include our code in the supplementary materials and will release the project as open source upon acceptance.

**Experiment regime.** We evaluate MFPG in a regime where high-fidelity samples are scarce, reflecting many real-world applications such as robot learning and molecule design. To simulate this setting, we restrict all methods to a buffer of 100 high-fidelity samples per policy update. Multi-fidelity methods may supplement this limited buffer with up to $100\times$ additional low-fidelity samples per update. Training for all methods is conducted for up to 1M high-fidelity environment steps.

**Hyperparameter configuration.** Given the restriction on high-fidelity samples per policy update, we first tune hyperparameters for the High-Fidelity Only REINFORCE baseline (with state-value baselines subtracted). This tuned configuration serves as the common backbone for all multi-fidelity methods (MFPG, DARC, and PAR), in line with standard practice in the off-dynamics RL literature (Xu et al., 2023; Lyu et al., 2024a). Their method-specific hyperparameters, including the number of low-fidelity samples for DARC and PAR, are tuned separately. The Low-Fidelity Only $(100\times)$ baseline is implemented as a direct ablation of the multi-fidelity methods. For further details on hyperparameter setup, see Section F. Overall, our tuning protocol grants the tested multi-fidelity baselines more tuning than MFPG, *underscoring the simplicity and minimal tuning overhead of MFPG.*

Notably, in our tuning process, PAR is particularly sensitive to hyperparameters; cf. Fig. 18 in Section F. Following (Lyu et al., 2024a), we adopt separate configurations **per task** for PAR, whereas all other methods use a single configuration across settings. MFPG has the fewest hyperparameters among all the tested multi-fidelity methods. We tuned only the exponential moving-average weight $\eta_{\mathrm{ma}}$ in Eq. (3) and found that MFPG's median performance consistently exceeded that of the High-Fidelity Only baseline across the tested values of $\eta_{\mathrm{ma}}$, even though the absolute performance metrics varied with $\eta_{\mathrm{ma}}$.

**Result reporting.** For the variance reduction study in Section 6.2, we run 5 random seeds per method, as this experiment already aggregates a large number of gradient estimates within each run and is primarily intended to illustrate variance trends. For the experiments in Sections 6.3 to 6.5, we run 20 random seeds per method and task setting.

Performance is evaluated in the high-fidelity environment every 2000 high-fidelity steps, using 10 evaluation episodes with a deterministic policy each time. We adopt two primary metrics: (i) the *final evaluation return*, defined as the return averaged over the last 20 evaluation steps, and (ii) area under curve (AUC), the area under the evaluation-return curve across training, estimated by integrating evaluation return along the high-fidelity step axis using the composite trapezoidal rule. AUC captures a method's *accumulated* return performance (Stadie et al., 2015; Osband et al., 2016; Hessel et al., 2018). Finally, to assess robustness, we also report the number of *performance collapses*, defined as cases where the median of a method falls below 50% of the median value of the corresponding High-Fidelity Only baseline.

**Uncertainty and significance.** Where reported, we compute two-sided 95% nonparametric bootstrap confidence intervals using $R = 10,000$ resamples (sampling with replacement over seeds). We report point estimates as sample means. In the baseline comparisons in Section 6.3, we bootstrap the mean difference of each method compared to High-Fidelity Only, i.e., $\Delta = \mathrm{mean}_{\mathrm{method}} - \mathrm{mean}_{\mathrm{High\text{-}Fidelity\ Only}}$, by independently resampling seeds within each method and recomputing $\Delta$ for each resample. We declare an improvement statistically significant at $\alpha = 0.05$ when the 95% bootstrap confidence interval for $\Delta$ excludes zero.

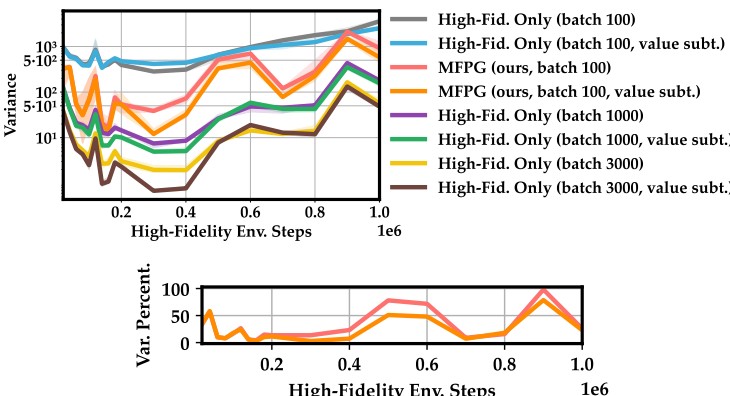

Figure 2: Variance of policy gradient estimates for MFPG versus variants of the High-Fidelity Only baseline on Hopper-v3 (top), and the percentage of MFPG variance relative to single-fidelity counterparts of the same batch size (bottom). Solid lines denote the median across seeds, and shaded regions indicate the corresponding maximum and minimum values. In high-fidelity data-scarce regimes with mild dynamics gaps, MFPG generally exhibits far lower PG variance than the single-fidelity counterparts.

## 6.2 Empirical Variance Reduction

This experiment supports our key insight **I1**: When high-fidelity samples are scarce and the dynamics gap between environments is mild, MFPG can substantially reduce the variance of policy gradient estimates compared to standard variance reduction via baseline subtraction.

We use a Hopper task with $1.2\times$ friction as an example. Figure 2 (top) shows the variance of PG estimates from MFPG versus a few variants of the High-Fidelity Only baseline (more specifically, we plot the variances of the scalar quantities $Z^{\pi_\theta}$ and $X_{\tau^h}^{\pi_\theta}$ before differentiation with respect to policy parameters). The plotted High-Fidelity Only variants include a baseline that has access to the same limited number of high-fidelity samples per policy update as MFPG (i.e., the same batch size), baselines with more high-fidelity samples (a batch size of 1000 and 3000), and variants of these High-Fidelity Only baselines that also use state-value function subtraction in an effort to reduce variance.

We train a policy using the single-fidelity REINFORCE algorithm with state-value subtracted for 1 million steps and save the trained policy at 18 different checkpoints. For each of these saved policies, we collect 200 batches of both high- and low-fidelity data (with low-fidelity data collected at $100\times$ the amount of high-fidelity data), where the size of each high-fidelity batch, i.e., the amount of high-fidelity data, varies between approaches as described above. We then record the empirical variance of the policy gradient estimates from these 200 batches, for each checkpointed policy. We repeat this experiment for 5 random seeds and report aggregate statistics in Figure 2 (top) where each line is based on 21600 batches of policy gradient estimates. Solid lines denote medians and shaded regions (which largely overlap the median lines) indicate the range between maxima and minima across the 5 seeds. Figure 2 (bottom) reports the ratio of median variance of policy gradient estimators for two variants of MFPG (with and without value-function subtraction), measured relative to their single-fidelity counterparts under the same batch size.

As shown in Figure 2, in data-scarce regimes, MFPG can substantially reduce the variance of policy gradient estimators compared to single-fidelity baselines at most training checkpoints. This variance reduction is generally far greater than that achieved by standard state-value subtraction, narrowing the gap to baselines that rely on significantly more high-fidelity samples (e.g., $10\times$).

## 6.3 Baseline Comparison Across Dynamics Variation Types and Levels

This section reports the evaluation of MFPG against baseline methods under the task settings described in Section 6.1 and supports our key insight **I2**.

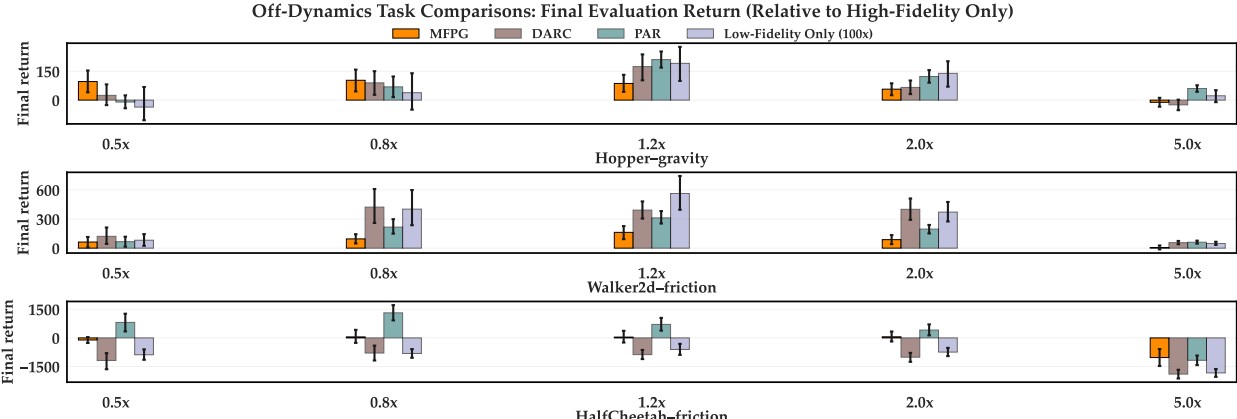

Figure 3: Mean difference in final evaluation return for each method relative to High-Fidelity Only, computed as $\Delta = \text{mean}_{\text{method}} - \text{mean}_{\text{High-Fidelity Only}}$ (0 corresponds to High-Fidelity Only). Bars show the bootstrap point estimate of $\Delta$, and error bars denote two-sided 95% bootstrap confidence intervals for $\Delta$. When low-fidelity data are neutral or beneficial (rows 1-2) and dynamics gaps are mild–moderate ($0.5\times$ to $2.0\times$), MFPG is *the only method* among the evaluated approaches that consistently attains statistically significant improvements over High-Fidelity Only. Strong Off-dynamics RL baselines tend to exploit low-fidelity data aggressively, which can occasionally yield high performance (e.g., in Walker2d) but can also degrade performance substantially more severely than MFPG when low-fidelity data become harmful; cf. row 3 and Fig. 4.

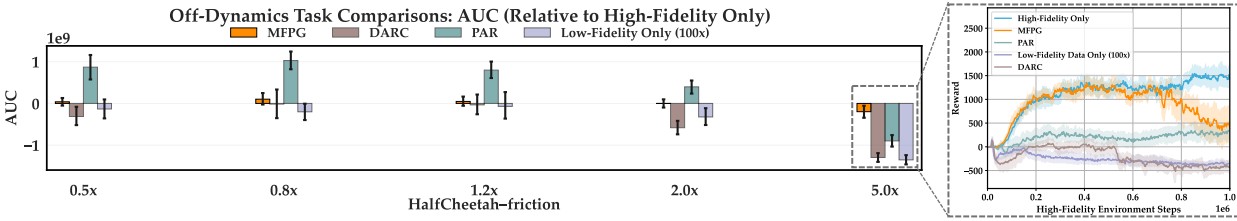

Figure 4: Mean difference in AUC for each method in HalfCheetah–friction relative to High-Fidelity Only, computed analogously to Fig. 3 (0 corresponds to High-Fidelity Only). Bars show the bootstrap point estimate of the mean difference, and error bars denote two-sided 95% bootstrap confidence intervals. When low-fidelity data become harmful, MFPG exhibits the strongest robustness to performance degradation among the evaluated methods.

Fig. 3 presents the mean *difference* in final evaluation return of each method *relative to* the High-Fidelity Only baseline, measured as the difference in mean final evaluation return $\Delta = \text{mean}_{\text{method}} - \text{mean}_{\text{High-Fidelity Only}}$ (so 0 corresponds to High-Fidelity Only), on 15 representative task settings, with error bars indicating two-sided 95% bootstrap confidence intervals for $\Delta$. Each row corresponds to one task, with the high-fidelity environment varying across multiple dynamics levels. For the simpler Hopper-gravity and Walker2d-friction tasks, abundant low-fidelity data tend to be neutral or beneficial and training is relatively stable, so the final evaluation return and AUC metrics are generally consistent; thus, we report only the final evaluation return. In contrast, the HalfCheetah-friction task is more challenging and low-fidelity data tend to be harmful: training exhibits greater fluctuations that occasionally reduce reward performance (baselines fluctuate far more substantially than MFPG). To capture accumulated learning performance more comprehensively in these settings, we report both the final return (Fig. 3, row 3) and the AUC metric (Fig. 4). We compute and visualize the AUC results analogously, i.e., as differences relative to High-Fidelity Only with two-sided 95% bootstrap confidence intervals.

**Hopper and Walker2d tasks.** As shown in Fig. 3 (rows 1-2), in these two tasks, abundant low-fidelity data generally tend to be neutral or beneficial: across all settings, the confidence intervals of the Low-Fidelity Only baseline are never strictly below 0. MFPG is the only evaluated method that reliably and consistently achieves statistically significant improvements when the dynamics gap is mild to moderate ($0.5\times$ to $2.0\times$)—i.e., with error bars strictly above 0 in all 8/8 such scenarios—by leveraging low-fidelity data and cross-fidelity correlation to complement the limited high-fidelity data.

No other off-dynamics RL or low-fidelity-only baseline achieves this level of consistency. For instance, in Hopper-v3 with a $0.5\times$ gravity shift, the Low-Fidelity Only baseline shows neutral performance relative to High-Fidelity Only. Consequently, the evaluated off-dynamics RL baselines fail to significantly improve over High-Fidelity Only, whereas MFPG effectively combines high- and low-fidelity data, leveraging cross-fidelity correlations to achieve a statistically significant performance gain over the High-Fidelity Only baseline.

When low-fidelity data tend to be strongly beneficial, e.g., in Hopper-v3 with a $1.2\times$ gravity shift and Walker2d-v3 with $0.8\times$–$2.0\times$ friction shifts, the evaluated off-dynamics RL baselines aggressively exploit the abundant low-fidelity data to boost performance, achieving significantly higher mean returns than High-Fidelity Only. By contrast, MFPG behaves more cautiously, providing reliable and consistent—yet still statistically significant—improvements over High-Fidelity Only.

We emphasize that this behavior of MFPG is **by design**: MFPG uses low-fidelity data solely as a variance-reduction tool, deliberately trading potentially aggressive (but risky) gains for reliability. This design choice underlies the consistency observed here and the strongest robustness that MFPG exhibits relative to the other evaluated off-dynamics RL methods in the HalfCheetah-v3 task presented below.

In extreme-shift scenarios, i.e., $5.0\times$ dynamics shifts, cross-fidelity correlation drops sharply and MFPG performance becomes effectively neutral relative to High-Fidelity Only.

Another interesting observation is that, even when it improves over High-Fidelity Only, the Low-Fidelity Only baseline tends to exhibit high cross-seed variance, as reflected by its long error bars in Hopper-v3 and Walker2d-v3 with $0.8\times$–$2.0\times$ dynamics shifts in the box plots shown in Figs. 14 and 15. Moreover, whether Low-Fidelity Only significantly improves over High-Fidelity Only does not correlate monotonically with the dynamics gap; for example, low-fidelity data appear more beneficial in Hopper-v3 with a $2.0\times$ gravity shift than with a $0.5\times$ shift. This highlights both the inherent difficulty of predicting when zero-shot transfer will be effective in practice and the importance of having reliable multi-fidelity approaches.

**HalfCheetah task.** As shown in Fig. 3 (row 3) and Fig. 4, in this task, low-fidelity data tend to be harmful: DARC and Low-Fidelity Only consistently and significantly degrade mean final return performance over High-Fidelity Only across mild-to-moderate gaps ($0.5\times$ to $2.0\times$), whereas PAR still manages to improve significantly over High-Fidelity Only and achieves the highest performance among the evaluated methods in these scenarios. However, PAR's performance gains come at the cost of *computationally expensive, per-task* hyperparameter tuning. We empirically observed that PAR is highly sensitive to the reward-augmentation weight $\beta$; see Fig. 18 in Section F. Even with such per-task tuning, PAR's performance often exhibits large cross-seed variance with low minima (cf. $0.5\times$ friction in Fig. 15 and several additional cases in Figs. 16 and 17).

By contrast, MFPG does not require such expensive, careful tuning; thanks to its algorithmic simplicity, we only tuned the exponential moving-average weight $\eta_{\mathrm{ma}}$ in Eq. (3) during our experiments. Across mild-to-moderate gaps, MFPG is *robust* to harmful low-fidelity data and remains effectively neutral with respect to High-Fidelity Only.

At an extreme dynamics shift, i.e., $5.0\times$ friction, the performance of DARC, PAR, Low-Fidelity Only all drops sharply, with the confidence intervals for *both* mean final return and AUC *far below* 0. In this scenario, we observe that the mean return curve of MFPG (cf. Fig. 4) initially closely tracks that of High-Fidelity Only before exhibiting some instability, resulting in a lower mean final return but relatively strong accumulated performance (i.e., AUC). By contrast, all other baselines struggle to achieve meaningful performance throughout training, suggesting that MFPG is the most robust among the evaluated off-dynamics RL and low-fidelity-only methods in this setting.

The observed instability of MFPG may stem from noise in the estimated control-variate coefficient $\hat{c}^*$, and designing more reliable estimation schemes constitutes an interesting direction for future work.

**Robustness.** The trends in Figs. 3 and 4 indicate that MFPG is the most robust among the evaluated off-dynamics RL and low-fidelity-only methods. Quantitatively, across the 15 scenarios in Figs. 3 and 4, the numbers of performance collapses (final return / AUC) are: **MFPG (1 / 0)**, PAR (1 / 1), Low-Fidelity Only (100×) (5 / 3), and DARC (5 / 3). Across the additional 24 settings in Section E, the corresponding collapse counts (final return / AUC) are: **MFPG (1 / 0)**, PAR (3 / 1), Low-Fidelity Only (100×) (8 / 7), and DARC (7 / 5).

*Taken together, these results support our key insight **I2** regarding the **reliability** and **robustness** of MFPG.*

**MFPG performance vs. correlation.** We examine one task in detail to illustrate the qualitative behavior of MFPG under varying dynamics gaps. Figure 5 presents evaluation-return curves of MFPG and High-Fidelity Only in the Hopper task with three levels of gravity variation (0.8×, 2.0×, and 5.0×), along with the estimated Pearson correlation coefficients between high- and low-fidelity policy gradient losses throughout training. Solid lines indicate bootstrap point estimates (means), and shaded regions denote two-sided 95% bootstrap confidence intervals. As the dynamics shift increases, the cross-fidelity correlation decreases (high for 0.8×, moderate for 2.0×, and near-zero for 5.0×). Consistent with this, the gap between MFPG's mean return and that of High-Fidelity Only is largest when the correlation is high (0.8×) and smaller when the correlation is moderate (2.0×); in both cases, MFPG trends above High-Fidelity Only for much of training, although the bootstrap confidence intervals overlap during some portions. At the extreme 5.0× shift—where the correlation collapses—the two methods exhibit very similar learning curves, with largely overlapping bootstrap confidence intervals. Figure 5 shows both algorithms' performance for completeness. Figure 13 further reports the mean difference in evaluation return between MFPG and High-Fidelity Only, computed as $\Delta = \text{mean}_{\text{method}} - \text{mean}_{\text{High-Fidelity Only}}$ ( $\Delta = 0$ indicates no difference). For mild (0.8×) and moderate (2.0×) dynamics gaps, the confidence intervals are often above zero, suggesting improved performance, with larger gains typically observed in the mild-gap case. Under the extreme shift (5.0×), the confidence intervals largely span zero. The AUC results in Fig. 12 are consistent with these trends.

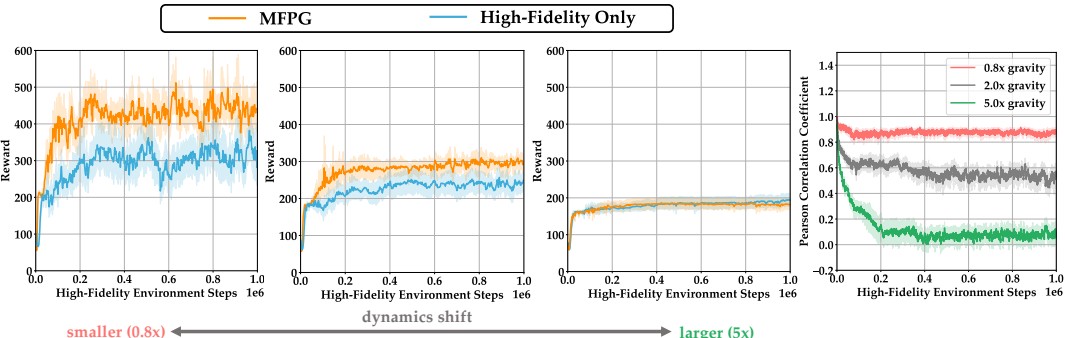

Figure 5: Evaluation return curves of MFPG versus the High-Fidelity Only baseline on Hopper-v3, corresponding to Fig. 3, under mild, moderate, and large gravity variations (left to right: 0.8×, 2.0×, 5.0×), together with estimated Pearson correlation coefficients between high- and low-fidelity policy gradient losses. Solid lines indicate bootstrap point estimates (means), and shaded regions denote two-sided 95% bootstrap confidence intervals. Consistent with the correlation estimates, the gap between MFPG's mean return and that of High-Fidelity Only is more pronounced when cross-fidelity correlation is stronger.

## 6.4 Low-Fidelity Environment with a Misspecified Reward

This experiment is designed to support our key insight **I3**, that when high-fidelity samples are scarce, MFPG can remain effective even in the presence of reward misspecification in the low-fidelity samples, e.g., when the low-fidelity reward is the negative (anti-correlated) version of the high-fidelity reward.

As discussed in Section 4, so long as there is a statistical relationship between the random variables of interest $X_{\tau^h}^{\pi_\theta}$ and $X_{\tau^l}^{\pi_\theta}$ (i.e., $\rho^2(X_{\tau^h}^{\pi_\theta}, X_{\tau^l}^{\pi_\theta})$ is non-negligible), the MFPG framework will reduce the variance of the policy gradient estimates (with respect to the high-fidelity environment).

To demonstrate this point, we examine a situation in which the reward function in the low-fidelity environment is the negative of that from the high-fidelity environment. Figure 6 reports the evaluation return of MFPG in comparison to the High-Fidelity Only and Low-Fidelity Only baselines. Solid lines indicate bootstrap point estimates (means), and shaded regions denote two-sided 95% bootstrap confidence intervals. All approaches in this example use the plain REINFORCE algorithm without state-value subtraction. The core idea extends to the case with value-function subtraction, except that separate value functions must be learned for the multi-fidelity environments because of the misspecified reward.

As expected, the Low-Fidelity Only baseline is entirely ineffective under the drastically misspecified reward: instead of learning to move forward, the agent learns to end the episode as quickly as possible. Meanwhile, the High-Fidelity Only baseline makes limited progress due to the scarcity of high-fidelity samples. In contrast, MFPG effectively combines the two data sources and achieves substantially better performance than both baselines.

Intuitively, although the values of $X_{\tau^l}^{\pi_\theta}$ are entirely different from those of $X_{\tau^h}^{\pi_\theta}$, they are negatively correlated in this example. The MFPG algorithm takes advantage of this relationship to compute a correction for the low-fidelity estimator $\hat{\mu}^l$, using only a small number of high-fidelity samples. Although this is an extreme example in the sense that the low- and high-fidelity tasks are polar opposites of each other, it highlights a useful feature of the MFPG framework: the low-fidelity rewards and dynamics might be very different from the target environment of interest (making direct sim-to-real transfer infeasible), and yet still provide useful information for multi-fidelity policy training.

*These results support our key insight **I3**. Taken together, our experiments validate all three key insights (**I1**–**I3**).*

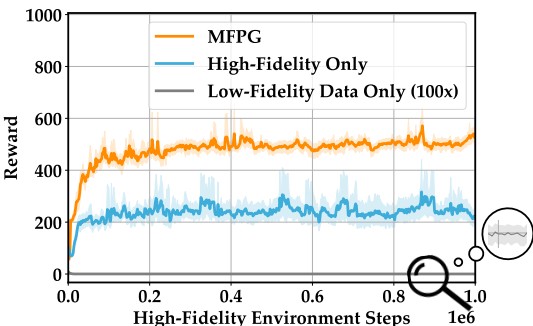

Figure 6: Evaluation return curves of MFPG, High-Fidelity Only, and Low-Fidelity Only on Hopper-v3, where the low-fidelity environment is misspecified with a negated reward. Solid lines indicate bootstrap point estimates (means), and shaded regions denote two-sided 95% bootstrap confidence intervals. Although low-fidelity data alone is ineffective for training, MFPG successfully leverages both fidelities to achieve substantial improvements over High-Fidelity Only in high-fidelity data-scarce regimes.

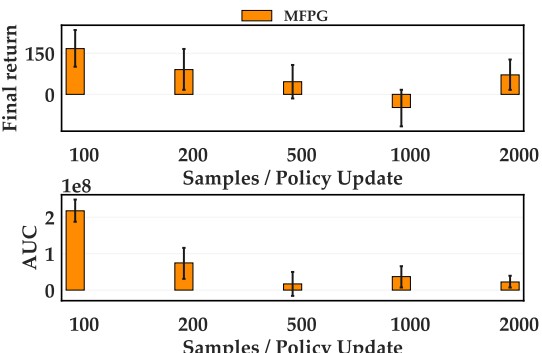

Figure 7: Mean difference in final return (top) and AUC (bottom) between MFPG and High-Fidelity Only on Hopper-v3 with $1.2\times$ friction, across varying high-fidelity batch sizes (i.e., samples per update). Zero corresponds to High-Fidelity Only. Error bars indicate two-sided 95% bootstrap confidence intervals for the mean difference. MFPG provides larger benefits over High-Fidelity Only in high-fidelity sample-scarce regimes.

## 6.5 Additional Experiments

This section presents additional experiments that provide a more detailed examination of different aspects of the MFPG framework. For brevity in the main text, Section D presents additional ablation results.

**MFPG performance with different amounts of high-fidelity samples.** Our main evaluation in Section 6.3 focuses on the high-fidelity data-scarce regime, which reflects many real-world scenarios where collecting large quantities of high-fidelity samples for each policy update is prohibitively expensive (e.g., in robotics or molecule design). This section studies how MFPG compares to the High-Fidelity Only baseline as the number of high-fidelity samples per policy update varies.

Figure 7 reports the mean difference in final return (top) and AUC (bottom) between MFPG and High-Fidelity Only in the Hopper task with 1.2× friction. Methods are trained for 1 million steps, comparing MFPG with 100× low-fidelity data against High-Fidelity Only across different batch sizes (i.e., numbers of high-fidelity samples per update). Error bars indicate two-sided 95% bootstrap confidence intervals for the mean difference. Note that with a fixed total number of high-fidelity environment steps, the number of gradient updates varies across batch sizes; consequently, the metrics for High-Fidelity Only do not necessarily increase monotonically with larger batch sizes. In high-fidelity data-scarce regimes, e.g., batch sizes of 100 and 200, MFPG yields statistically significant improvements over High-Fidelity Only with relatively large mean differences, and the margin decreases as the batch size increases. When high-fidelity samples are abundant, e.g., batch sizes above 500, MFPG is largely neutral relative to High-Fidelity Only (though it still shows a small but statistically significant improvement at a batch size of 2000). These results highlight that MFPG is most effective in high-fidelity data-scarce settings.

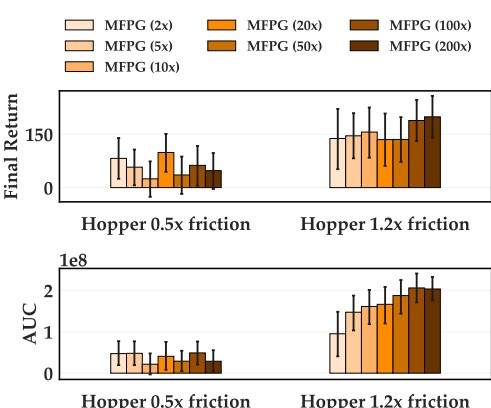

Figure 8: Mean difference in final return (top) and AUC (bottom) of MFPG with increasing amounts of low-fidelity data (light to dark orange) relative to High-Fidelity Only on Hopper-v3 with 0.5× and 1.2× friction (0 corresponds to High-Fidelity Only). Error bars indicate two-sided 95% bootstrap confidence intervals for the mean differences. MFPG is relatively robust to the amount of low-fidelity data: across the tested range of low-fidelity sample sizes, we do not observe statistically significant degradation relative to High-Fidelity Only under our significance test, and MFPG is often statistically significantly better.

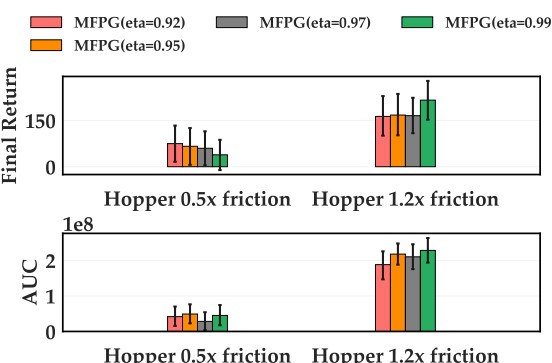

Figure 9: Mean difference in final return (top) and AUC (bottom) of MFPG with varying exponential moving-average weights ($\eta_{\mathrm{ma}}$ in Eq. (3)) relative to High-Fidelity Only (0 corresponds to High-Fidelity Only) on Hopper-v3 with 0.5× and 1.2× friction. Error bars indicate two-sided 95% bootstrap confidence intervals for the mean differences. Across the tested values of $\eta_{\mathrm{ma}}$, MFPG is often statistically significantly better than High-Fidelity Only, and we do not observe statistically significant degradation.

**MFPG performance with different amounts of low-fidelity samples.** We assume throughout this work that low-fidelity samples are cheap to collect, and all methods are given access to an abundant budget of low-fidelity data. Consequently, optimizing the efficiency of low-fidelity sample usage (Cutler et al., 2015) is not the focus of this work. Instead, this section qualitatively examines how the performance of MFPG varies with different amounts of low-fidelity data.

In principle, the performance of MFPG is robust with respect to the amount of low-fidelity samples, so long as the low-fidelity sample amount is sufficient to estimate the low-fidelity sample mean $\hat{\mu}^l$. Figure 8 reports mean differences in final return (top) and AUC (bottom) of MFPG relative to High-Fidelity Only (0 corresponds to High-Fidelity Only) with increasing amounts of low-fidelity samples (indicated from light to dark orange) for estimating low-fidelity sample mean ($\hat{\mu}^l$ in Fig. 1), in Hopper with $0.5\times$ and $1.2\times$ friction. Error bars indicate two-sided 95% bootstrap confidence intervals.

At a high level, MFPG is relatively robust to the amount of low-fidelity data: in our experiment, varying the low-fidelity sample amount over a wide range does not lead to performance that is statistically significantly worse than High-Fidelity Only in both settings, and MFPG is often statistically significantly better. When the dynamics gap is mild (e.g., $1.2\times$ friction), for the AUC metric, increasing the number of low-fidelity samples tends to yield a higher bootstrap mean improvement in AUC over High-Fidelity Only. In particular, the confidence intervals for $100\times$ and $200\times$ low-fidelity data lie clearly above those for $2\times$ low-fidelity data, suggesting a significantly faster learning process. However, under larger dynamics gaps, no clear monotonic trend is observed. Overall, additional low-fidelity samples are more beneficial in mild-gap settings.

**Sensitivity to exponential moving-average weight.** We further examine the sensitivity of MFPG to the choice of the exponential moving-average weight $\eta_{\text{ma}}$ in Eq. (3). Figure 9 reports the mean differences in final return and AUC between MFPG (with different $\eta_{\text{ma}}$ values) and High-Fidelity Only (0 corresponds to High-Fidelity Only) in the Hopper task under $0.5\times$ and $1.2\times$ friction. Error bars indicate two-sided 95% bootstrap confidence intervals for the mean differences. Overall, across the tested values of $\eta_{\text{ma}}$, MFPG is often statistically significantly better than High-Fidelity Only, and we do not observe statistically significant degradation. The mean performance improvement of MFPG over High-Fidelity Only is generally more substantial in settings with smaller dynamics gaps—consistent with the key insights from our baseline comparison in Section 6.3.

## 7 Conclusions

We present a multi-fidelity policy gradient (MFPG) framework, which mixes a small amount of potentially expensive high-fidelity data with a larger volume of cheap lower-fidelity data to construct unbiased, variance-reduced estimators for on-policy policy gradients. We instantiate this framework through a practical, multi-fidelity variant of the standard REINFORCE algorithm. Theoretically, we show that under standard assumptions, the MFPG estimator guarantees asymptotic convergence of multi-fidelity REINFORCE to locally optimal policies (up to first order optimality conditions) in the target high-fidelity environment of interest. Moreover, whenever the statistical correlation between high- and low-fidelity samples is nontrivial, MFPG REINFORCE achieves faster finite-sample convergence than its single-fidelity counterpart. Empirical evaluation of MFPG in high-fidelity data-scarce regimes across a suite of simulated robotics benchmark tasks with varied multi-fidelity transition dynamics highlights both its *reliability* and *robustness*. In our baseline comparison experiments, supported by a 20-seed statistical analysis, for scenarios where low-fidelity data are neutral or beneficial and dynamics gaps are mild to moderate, MFPG is, among the evaluated off-dynamics RL and low-fidelity-only approaches, *the only* method that consistently achieves statistically significant improvements in mean performance over a baseline trained solely on high-fidelity data. When low-fidelity data become harmful, MFPG exhibits the strongest robustness against performance degradation among the evaluated methods. By contrast, strong off-dynamics RL methods tend to exploit low-fidelity data more aggressively, and can achieve high performance at times, but can also fail substantially more severely. Furthermore, an additional experiment in which the high- and low-fidelity environments are assigned anti-correlated rewards demonstrates that MFPG can remain effective even in the presence of reward misspecification in the low-fidelity environment.

In summary, the proposed framework offers a reliable and robust paradigm for leveraging multi-fidelity data in reinforcement learning, providing promising directions for efficient sim-to-real transfer and principled approaches to managing the trade-off between policy performance and data collection costs.

**Limitations & future work.**  Future research spans several avenues. First, we study the classic REIN-FORCE algorithm as a starting point to ease theoretical analysis.   Extending the MFPG framework to modern RL algorithms—including actor–critic, off-policy, and offline methods—raises new questions about bias–variance trade-offs that merit deeper investigation. We defer a more detailed discussion of this point to Section G.

Second, MFPG relies on nontrivial statistical correlation between multi-fidelity samples to achieve variance reduction. Developing systematic techniques to enhance correlation between high- and low-fidelity samples without introducing bias is critical for further performance gains.

Third, our analysis and implementation make several simplifying assumptions, and extending the MFPG framework to more general settings is of great importance. These extensions include (i) integrating data from an arbitrary number of environments, each simulating the target environment with a different fidelity level and cost per sample, (ii) handling multi-fidelity MDPs with different state and action spaces, initial-state distributions, or discount factors, (iii) extending MFPG to infinite-horizon MDPs, and (iv) devising mechanisms to sample correlated trajectories even when low-fidelity simulators do not support resets to arbitrary user-specified initial states.

Finally, applying MFPG to real-world robotic systems represents an important step toward realizing its potential in practical sim-to-real transfer.

## Broader Impact Statement

This work introduces a methodological contribution—multi-fidelity policy gradients—for improving the (high-fidelity) sample efficiency of reinforcement learning in settings with access to both high- and low-fidelity environments. We do not foresee immediate societal risks arising directly from this research. Potential downstream applications may include robotics and autonomous systems, where increased efficiency can reduce computational and data collection costs.

## Acknowledgments

We thank Haoran Xu, Mustafa Karabag, Brett Barkley, and Jacob Levy for their helpful discussions. This work was supported in part by the National Science Foundation (NSF) under Grants 2409535 and 2214939, in part by the Defense Advanced Research Projects Agency (DARPA) under the Transfer Learning from Imprecise and Abstract Models to Autonomous Technologies (TIAMAT) program under grant number HR0011-24-9-0431, and in part by the DEVCOM Army Research Laboratory under Cooperative Agreement Numbers W911NF-23-2-0011 and W911NF-25-2-0021.  The views and conclusions contained in this document are those of the authors and should not be interpreted as representing the official policies, either expressed or implied, of the U.S. Government. The U.S. Government is authorized to reproduce and distribute reprints for Government purposes notwithstanding any copyright notation herein.

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

## A    Complete MFPG REINFORCE Algorithm

Algorithm 2 presents the complete algorithm block for the proposed multi-fidelity variant of the REINFORCE algorithm.

---

**Algorithm 2:** Multi-Fidelity Policy Gradients (MFPGs) - REINFORCE

---

**Input:** Initial policy $\pi_{\theta_1}$, value network $V_\phi$, multi-fidelity environments $\mathcal{M}^h$ and $\mathcal{M}^l$, batch size for correlated trajectory samples $N^h$, batch size for control variate sample mean estimate $N^l$, step sizes $\{\alpha_k\}$, maximum number of iterations $N$.

**Output:** Trained policy $\pi_{\theta_N}$.

**1 for** $k = 1$ **to** $N$ **do**

**2**    `// Current policy at iteration k:` $\pi_{\theta_k}(a \mid s)$

**3**    Collect $N^h$ correlated trajectory samples $\mathcal{D}_1 = \{\tau_i^h, \tau_i^l\}_{i=1}^{N^h}$ using $\pi_{\theta_k}$. ;      `// See Algorithm 1`

**4**    Collect $N^l$ additional uncorrelated low-fidelity trajectory samples $\mathcal{D}_2 = \{\tau_j^l\}_{j=1}^{N^l}$ using $\pi_{\theta_k}$.

**5**    Compute correlated policy gradient random variables using $\mathcal{D}_1$:

$$X_{\tau^h,i}^{\pi_{\theta_k}} = \frac{1}{T} \sum_{t=0}^{T-1} (G_{i,t}^h - V_\phi(s_{i,t}^h)) \log \pi_{\theta_k}(a_{i,t}^h \mid s_{i,t}^h),$$

$$X_{\tau^l,i}^{\pi_{\theta_k}} = \frac{1}{T} \sum_{t=0}^{T-1} (G_{i,t}^l - V_\phi(s_{i,t}^l)) \log \pi_{\theta_k}(a_{i,t}^l \mid s_{i,t}^l).$$

**6**    Estimate the optimal control variate coefficient $\hat{c}_k^*$ using $\mathcal{D}_1$ and Eq. (4).

**7**    Estimate the control variate sample mean using $\mathcal{D}_2$: $\hat{\mu}_k^l = \frac{1}{N^l} \sum_{j=1}^{N^l} X_{\tau^l,j}^{\pi_{\theta_k}}$.

**8**    Construct MFPG estimator: $\nabla_{\theta_k} \mathbb{E}[Z^{\pi_{\theta_k}}] \approx \widehat{g}_k^{\text{MFPG}} = \frac{1}{N^h} \sum_{i=1}^{N^h} \nabla_{\theta_k} [X_{\tau^h,i}^{\pi_{\theta_k}} + \hat{c}_k^*(X_{\tau^l,i}^{\pi_{\theta_k}} - \hat{\mu}_k^l)]$.

**9**    Perform policy update $\theta_{k+1} \leftarrow \theta_k + \alpha_k \widehat{g}_k^{\text{MFPG}}$.

**10**    Fit the value function network $V_\phi$ using high-fidelity samples in $\mathcal{D}_1$.

**11 return** *policy* $\pi_{\theta_N}$

---

## B   Proof of Lemma 1

The proof of Lemma 1 follows standard control variate arguments (Owen, 2013, Ch.8) and proceeds as follows.

*Proof.* We first establish unbiasedness of the control variate estimator for an arbitrary coefficient $c \in \mathbb{R}$. Taking expectation for the defined multi-fidelity control variate random variable over the joint probability space $\Omega$ defined in Section 4 yields

$$\mathbb{E}_{\omega \sim \Omega}[Z^{\pi_\theta}(c)] = \mathbb{E}_{\omega \sim \Omega}[X_{\tau^h}^{\pi_\theta} + c(X_{\tau^l}^{\pi_\theta} - \mu^l)]$$

$$= \mathbb{E}_{\omega \sim \Omega}[X_{\tau^h}^{\pi_\theta}] + c(\underbrace{\mathbb{E}_{\omega \sim \Omega}[X_{\tau^l}^{\pi_\theta}] - \mu^l}_{=0})$$

$$= \mathbb{E}_{\omega \sim \Omega}[X_{\tau^h}^{\pi_\theta}]$$

$$= \mathbb{E}_{\tau^h \sim \mathcal{M}^h(\pi_\theta)}[X_{\tau^h}^{\pi_\theta}],$$

where sampling from the joint probability space $\omega \sim \Omega$ reduces to sampling single-fidelity trajectories $\tau^h \sim \mathcal{M}^h(\pi_\theta)$ and $\tau^l \sim \mathcal{M}^l(\pi_\theta)$ when evaluating the expectation of the corresponding single-fidelity random variables $X_{\tau^h}^{\pi_\theta}$ and $X_{\tau^l}^{\pi_\theta}$, respectively.

Next, we examine the variance of the multi-fidelity control variate estimator:

$$\text{Var}(Z^{\pi_\theta}(c)) = \text{Var}(X_{\tau^h}^{\pi_\theta} + c(X_{\tau^l}^{\pi_\theta} - \mu^l))$$

$$= \text{Var}(X_{\tau^h}^{\pi_\theta} + cX_{\tau^l}^{\pi_\theta})$$

$$= \text{Var}(X_{\tau^h}^{\pi_\theta}) + c^2 \text{Var}(X_{\tau^l}^{\pi_\theta}) + 2c\,\text{Cov}(X_{\tau^h}^{\pi_\theta}, X_{\tau^l}^{\pi_\theta}),$$

where the second line uses the fact that adding/subtracting a constant does not change the variance of a random variable, and the third line uses the standard formula for the variance of a linear combination of random variables. The variance $\mathrm{Var}\big(Z^{\pi_\theta}(c)\big)$ defines a quadratic function in $c$, whose minimizer can be found by setting

$$\nabla_c \mathrm{Var}\big(Z^{\pi_\theta}(c)\big) = 0 \implies c^* = -\frac{\mathrm{Cov}(X_{\tau^h}^{\pi_\theta}, X_{\tau^l}^{\pi_\theta})}{\mathrm{Var}(X_{\tau^l}^{\pi_\theta})} = -\rho(X_{\tau^h}^{\pi_\theta}, X_{\tau^l}^{\pi_\theta}) \frac{\sqrt{\mathrm{Var}(X_{\tau^h}^{\pi_\theta})}}{\sqrt{\mathrm{Var}(X_{\tau^l}^{\pi_\theta})}}.$$

At the minimizer $c^*$, the variance of the multi-fidelity control variate estimator is

$$\begin{aligned}
\mathrm{Var}\big(Z^{\pi_\theta}(c^*)\big) &= \mathrm{Var}(X_{\tau^h}^{\pi_\theta}) + \frac{\mathrm{Cov}^2(X_{\tau^h}^{\pi_\theta}, X_{\tau^l}^{\pi_\theta})}{\mathrm{Var}(X_{\tau^l}^{\pi_\theta})} - \frac{2\mathrm{Cov}^2(X_{\tau^h}^{\pi_\theta}, X_{\tau^l}^{\pi_\theta})}{\mathrm{Var}(X_{\tau^l}^{\pi_\theta})} \\
&= \mathrm{Var}(X_{\tau^h}^{\pi_\theta}) - \frac{\rho^2(X_{\tau^h}^{\pi_\theta}, X_{\tau^l}^{\pi_\theta})\mathrm{Var}(X_{\tau^h}^{\pi_\theta})\mathrm{Var}(X_{\tau^l}^{\pi_\theta})}{\mathrm{Var}(X_{\tau^l}^{\pi_\theta})} \\
&= \big(1 - \rho^2(X_{\tau^h}^{\pi_\theta}, X_{\tau^l}^{\pi_\theta})\big)\mathrm{Var}(X_{\tau^h}^{\pi_\theta}).
\end{aligned}$$

$\square$

## C  Proof of Theorem 1

The proof of Theorem 1 leverages the Lipschitz continuity of the policy gradient for a policy deployed in the MDP $\mathcal{M}^h$, under assumptions Assumptions 1 to 4. Under these assumptions, the Lipschitzness of the policy gradient was proved for the infinite horizon setting by Zhang et al. (2020). We follow a similar procedure to establish the Lipschitzness for the finite-horizon case, stated below.

**Lemma 2** (Adapted from Zhang et al. (2020)). *Under Assumptions 1 to 4, for a problem of horizon $T$, the high-fidelity only policy gradient $\nabla_\theta J(\theta) = \mathbb{E}_{\tau \sim \mathcal{M}^h(\theta)}[R(\tau)]$ is $L_T$-Lipschitz, where*

$$\begin{aligned}
L_T =& \frac{1 - (T+2)\gamma^{T+1} + (T+1)\gamma^{T+2}}{(1-\gamma)^2} \cdot U_R^h L_\Theta \\
&+ \frac{\big(1 - (T+2)^2\gamma^{T+1} + (T+3)(T+1)\gamma^{T+2}\big)(1-\gamma) + 2\gamma\big(1 - (T+2)\gamma^{T+1} + (T+1)\gamma^{T+2}\big)}{(1-\gamma)^3} \cdot U_R^h B_\Theta^2.
\end{aligned}$$

*Proof.* Note that the high-fidelity only policy gradient can be written as

$$\nabla J(\theta) = \sum_{t=0}^{T} \sum_{\kappa=0}^{T-t} \gamma^{t+\kappa} \int r^h(s_{t+\kappa}^h, a_{t+\kappa}^h) \cdot \nabla_\theta \log \pi_\theta \cdot p_{\theta,0:t+\kappa} \cdot ds_{0:t+\kappa}^h \cdot da_{0:t+\kappa}^h,$$

$$\text{where } p_{\theta,0:t+\kappa} = \left(\prod_{u=0}^{t+\kappa-1} p(s_{u+1}^h | s_u^h, a_u^h)\right) \cdot \left(\prod_{u=0}^{t+\kappa} \pi_\theta(a_u^h | s_u^h)\right).$$

Thus, for $\theta_1, \theta_2 \in \mathbb{R}^d$, we have for the high-fidelity only policy gradient,

$$\|\nabla J(\theta_1) - \nabla J(\theta_2)\| = \left\| \sum_{t=0}^{T} \sum_{\kappa=0}^{T-t} \gamma^{t+\kappa} \int \left( \underbrace{r^h(s^h_{t+\kappa}, a^h_{t+\kappa}) \left[ \nabla \log \pi_{\theta_1}(a^h_t \mid s^h_t) - \nabla \log \pi_{\theta_2}(a^h_t \mid s^h_t) \right] p^h_{\theta_1, 0:t+\kappa}}_{A} \right.$$

$$\left. + \underbrace{r^h(s^h_{t+\kappa}, a^h_{t+\kappa}) \nabla \log \pi_{\theta_2}(a^h_t \mid s^h_t) \left( p^h_{\theta_1, 0:t+\kappa} - p^h_{\theta_2, 0:t+\kappa} \right)}_{B} \right) ds^h_{0:t+\kappa} da^h_{0:t+\kappa} \right\|$$

$$\leq \sum_{t=0}^{\infty} \sum_{\kappa=0}^{\infty} \gamma^{t+\kappa} \left[ \underbrace{\int \left( \left| r^h(s^h_{t+\kappa}, a^h_{t+\kappa}) \right| \left\| \nabla \log \pi_{\theta_1}(a^h_t \mid s^h_t) - \nabla \log \pi_{\theta_2}(a^h_t \mid s^h_t) \right\| p^h_{\theta_1, 0:t+\kappa} ds^h_{0:t+\kappa} da^h_{0:t+\kappa} \right.}_{I_1} \right.$$

$$\left. + \underbrace{\int \left| r^h(s^h_{t+\kappa}, a^h_{t+\kappa}) \right| \left\| \nabla \log \pi_{\theta_2}(a^h_t \mid s^h_t) \right\| \left| p^h_{\theta_1, 0:t+\kappa} - p^h_{\theta_2, 0:t+\kappa} \right| ds^h_{0:t+\kappa} da^h_{0:t+\kappa}}_{I_2} \right] \tag{5}$$

From Assumption 4, we have

$$\|A\| \leq U^h_R L_\Theta \|\theta^1 - \theta^2\|$$
$$\implies I_1 \leq U^h_R L_\Theta \|\theta^1 - \theta^2\|. \tag{6}$$

Further, note that

$$p^h_{\theta_1, 0:t+\kappa} - p^h_{\theta_2, 0:t+\kappa} = \left( \prod_{u=0}^{t+\kappa-1} p(s^h_{u+1} | s^h_u, a^h_u) \right) \cdot \left( \prod_{u=0}^{t+\kappa} \pi_{\theta^1}(a^h_u | s^h_u) - \prod_{u=0}^{t+\kappa} \pi_{\theta^2}(a^h_u | s^h_u) \right)$$

From Taylor's theorem, $\exists \; \lambda \in (0,1)$ such that for $\tilde{\theta} := \lambda \theta^1 + (1-\lambda)\theta^2$, we have

$$\left| \prod_{u=0}^{t+\kappa} \pi_{\theta^1}(a^h_u | s^h_u) - \prod_{u=0}^{t+\kappa} \pi_{\theta^2}(a^h_u | s^h_u) \right| = \left| (\theta^1 - \theta^2)^\top \left( \sum_{i=0}^{t+\kappa} \nabla_\theta \pi_{\tilde{\theta}}(a^h_i | s^h_i) \prod_{u=0, u \neq i}^{t+\kappa} \pi_{\tilde{\theta}}(a^h_u | s^h_u) \right) \right|$$

$$\leq \|\theta^1 - \theta^2\| \cdot \sum_{i=0}^{t+\kappa} \left\| \nabla_\theta \log \pi_{\tilde{\theta}}(a^h_i | s^h_i) \right\| \cdot \prod_{u=0}^{t+\kappa} \pi_{\tilde{\theta}}(a^h_u | s^h_u)$$

$$\leq (t+\kappa+1) \cdot B_\Theta \cdot \|\theta^1 - \theta^2\| \cdot \prod_{u=0}^{t+\kappa} \pi_{\tilde{\theta}}(a^h_u | s^h_u) \qquad \text{(Assumption 3)}$$

$$\implies I_2 \leq (t+\kappa+1) \cdot U^h_R \cdot B_\Theta^2 \cdot \|\theta^1 - \theta^2\|. \tag{7}$$

From Equations (5) to (7), we get

$$\|\nabla J(\theta_1) - \nabla J(\theta_2)\| \leq \left( \sum_{t=0}^{T} \sum_{\kappa=0}^{T-t} \left[ U^h_R L_\Theta \gamma^{t+\kappa} + U^h_R B_\Theta^2 (t+\kappa+1) \gamma^{t+\kappa} \right] \right) \|\theta^1 - \theta^2\|$$

Using the fact that

$$\sum_{t=0}^{T} \sum_{\kappa=0}^{T-t} \gamma^{t+\kappa} = \frac{1 - (T+2)\gamma^{T+1} + (T+1)\gamma^{T+2}}{(1-\gamma)^2},$$

$$\sum_{t=0}^{T} \sum_{\kappa=0}^{T-t} (t+\kappa+1)\gamma^{t+\kappa} = \frac{\left(1 - (T+2)^2 \gamma^{T+1} + (T+3)(T+1)\gamma^{T+2}\right)(1-\gamma)}{(1-\gamma)^3}$$

$$+ \frac{2\gamma \left(1 - (T+2)\gamma^{T+1} + (T+1)\gamma^{T+2}\right)}{(1-\gamma)^3},$$

we get that

$$L_T = \frac{1 - (T+2)\gamma^{T+1} + (T+1)\gamma^{T+2}}{(1-\gamma)^2} \cdot U_R^h L_\Theta$$
$$+ \frac{\left(1 - (T+2)^2\gamma^{T+1} + (T+3)(T+1)\gamma^{T+2}\right)(1-\gamma) + 2\gamma\left(1 - (T+2)\gamma^{T+1} + (T+1)\gamma^{T+2}\right)}{(1-\gamma)^3} \cdot U_R^h B_\Theta^2.$$

Note that in the limit of infinite time horizon, we recover the Lipschitz bounds proposed by Zhang et al. (2020) for the infinite horizon setting, given by

$$\lim_{T\to\infty} L_T = \frac{1}{(1-\gamma)^2} \cdot U_R^h L_\Theta + \frac{(1+\gamma)}{(1-\gamma)^3} \cdot U_R^h B_\Theta^2.$$

$\square$

To establish the asymptotic almost sure convergence of $\|\nabla J(\theta^{MFPG})\|$ to 0, we also need the following preliminary result, which is an adaptation of a similar result from Zhang et al. (2020), but for the multi-fidelity case.

**Lemma 3.** *Under Assumptions 1 to 4, the norm of the Multi-Fidelity Policy Gradient estimator is bounded, i.e., $\exists\, \hat{\ell} > 0$ such that $\|\nabla \hat{J}^{MFPG}(\theta)\| \le \hat{\ell}\ \forall\ \theta \in \mathbb{R}^d$. Further, Consider the random variable $W_k$, defined as*

$$W_k := J(\theta_k^{MFPG}) - L_T \hat{\ell}^2 \sum_{j=k}^{\infty} \alpha_j^2.$$

*Let $\mathcal{F}_k$ be a filtration denoting all randomness in $k$ iterations. Then, we have that*

$$\mathbb{E}[W_{k+1}|\mathcal{F}_k] \ge W_k + \alpha_k \|\nabla J(\theta_k^{MFPG})\|^2.$$

*Proof.* We first show the boundedness of the MFPG estimator. We have

$$\left\|\nabla X_{\tau^h,i}^{\pi_\theta}\right\| = \left\|\frac{1}{T}\sum_{t=0}^{T-1} G_{i,t}^h \nabla \log \pi_\theta(a_{i,t}^h \mid s_{i,t}^h)\right\| \le \frac{U_R^h B_\Theta}{T}\sum_{t=0}^{T-1}\gamma^t = \frac{U_R^h B_\Theta}{T}\cdot\frac{1-\gamma^T}{1-\gamma}$$

$$\left\|\nabla X_{\tau^l,i}^{\pi_\theta}\right\| = \left\|\frac{1}{T}\sum_{t=0}^{T-1} G_{i,t}^l \nabla \log \pi_\theta(a_{i,t}^l \mid s_{i,t}^l)\right\| \le \frac{U_R^l B_\Theta}{T}\sum_{t=0}^{T-1}\gamma^t = \frac{U_R^l B_\Theta}{T}\cdot\frac{1-\gamma^T}{1-\gamma}$$

$$\implies \left\|\frac{1}{N^h}\sum_{i=1}^{N^h}\left[\nabla X_{\tau^h,i}^{\pi_\theta} + c^*\left(\nabla X_{\tau^l,i}^{\pi_\theta} - \mathbb{E}\left[\nabla X_{\tau^l,i}^{\pi_\theta}\right]\right)\right]\right\| \le \frac{U_R^h B_\Theta}{TN^h}\cdot\frac{1-\gamma^T}{1-\gamma} + 2c^*\frac{U_R^l B_\Theta}{TN^h}\cdot\frac{1-\gamma^T}{1-\gamma} := \hat{\ell}$$

We now show the stochastic ascent property associated with $W_k$. By Taylor's theorem we have

$$W_{k+1} = J(\theta_k^{MFPG}) + \left(\theta_{k+1}^{MFPG} - \theta_k^{MFPG}\right)^\top \nabla J(\tilde{\theta}_k^{\ MFPG}) - L_T \hat{\ell}^2 \sum_{j=k+1}^{\infty} \alpha_j^2$$

$$= J(\theta_k^{MFPG}) + \left(\theta_{k+1}^{MFPG} - \theta_k^{MFPG}\right)^\top \nabla J(\theta_k^{MFPG})$$

$$+ \left(\theta_{k+1}^{MFPG} - \theta_k^{MFPG}\right)^\top \left(\nabla J(\tilde{\theta}_k^{\ MFPG}) - \nabla J(\theta_k^{MFPG})\right) - L_T \hat{\ell}^2 \sum_{j=k+1}^{\infty} \alpha_j^2$$

$$\geq J(\theta_k^{MFPG}) + \left(\theta_{k+1}^{MFPG} - \theta_k^{MFPG}\right)^\top \nabla J(\theta_k^{MFPG}) - L_T \left\|\theta_{k+1}^{MFPG} - \theta_k^{MFPG}\right\|^2$$

$$- L_T \hat{\ell}^2 \sum_{j=k+1}^{\infty} \alpha_j^2$$

$$\implies \mathbb{E}\left[W_{k+1}|\mathcal{F}_k\right] \geq J(\theta_k^{MFPG}) + \mathbb{E}\left[\theta_{k+1}^{MFPG} - \theta_k^{MFPG}|\mathcal{F}_k\right]^\top \nabla J(\theta_k^{MFPG})$$

$$- L_T \mathbb{E}\left[\left\|\theta_{k+1}^{MFPG} - \theta_k^{MFPG}\right\|^2 |\mathcal{F}_k\right] - L_T \hat{\ell}^2 \sum_{j=k+1}^{\infty} \alpha_j^2$$

$$= J(\theta_k^{MFPG}) + \mathbb{E}\left[\theta_{k+1}^{MFPG} - \theta_k^{MFPG}|\mathcal{F}_k\right]^\top \nabla J(\theta_k^{MFPG})$$

$$- L_T \alpha_k^2 \mathbb{E}\left[\|\nabla J(\theta_k^{MFPG})\|^2 |\mathcal{F}_k\right] - L_T \hat{\ell}^2 \sum_{j=k+1}^{\infty} \alpha_j^2$$

$$\geq J(\theta_k^{MFPG}) + \mathbb{E}\left[\theta_{k+1}^{MFPG} - \theta_k^{MFPG}|\mathcal{F}_k\right]^\top \nabla J(\theta_k^{MFPG})$$

$$- L_T \alpha_k^2 \hat{\ell}^2 - L_T \hat{\ell}^2 \sum_{j=k+1}^{\infty} \alpha_j^2$$

$$\implies \mathbb{E}\left[J(\theta_{k+1}^{MFPG})|\mathcal{F}_k\right] \geq J(\theta_k^{MFPG}) + \mathbb{E}\left[\theta_{k+1}^{MFPG} - \theta_k^{MFPG}|\mathcal{F}_k\right]^\top \nabla J(\theta_k^{MFPG}) - L_T \alpha_k^2 \hat{\ell}^2.$$

To show the bound, we have $\mathbb{E}\left[\theta_{k+1}^{MFPG} - \theta_k^{MFPG}|\mathcal{F}_k\right] = \mathbb{E}\left[\alpha_k \nabla \hat{J}^{MFPG}(\theta^{MFPG})|\mathcal{F}_k\right] = \alpha_k \nabla J(\theta_k^{MFPG})$. *This is the crucial step dependent on unbiasedness of control variates, which allows us to conduct this analysis.* Putting this in the above inequality gives us the desired bound. $\qquad\square$

Using Lemmas 2 and 3, we can now formally prove Theorem 1, which we restate for the reader's convenience.

**Theorem 1.** *Under Assumptions 1 to 6, let $\{\theta_k^{MFPG}\}_{k\in\mathbb{Z}^+}$, $\{\theta_k^h\}_{k\in\mathbb{Z}^+}$ be the sequence of policy parameters generated by running the REINFORCE algorithm with MFPG and high-fidelity only policy gradient estimators. Then, for a problem with time horizon $T$, after $N$ iterations, with step sizes for both iterates taken to be a sequence $\alpha_k$ satisfying $\sum_k \alpha_k = \infty, \sum_k \alpha_k^2 < \infty$, we have*

$$\min_{k\in[N]} \mathbb{E}\left[\|\nabla J(\theta_k^{MFPG})\|^2\right] \leq \frac{J(\theta^*) - J(\theta_1) + \left(1 - \rho^2\right)\frac{\sigma^2 L_T}{2} \sum_{k=1}^{N} \alpha_k^2}{\sum_{k=1}^{N}\left(\alpha_k - \frac{\alpha_k^2 L_T}{2}\right)},$$

*where $L_T$ is the Lipschitz constant of the high-fidelity policy gradient, established in Lemma 2. Moreover, we recover the corresponding rate for the high-fidelity only policy iterates $\{\theta_k^h\}_{k\in\mathbb{Z}^+}$ by substituting $\rho = 0$. Finally, we have that $\lim_{k\to\infty} \mathbb{E}\left[\|\nabla J(\theta_k^{MFPG})\|\right] = 0$ almost surely.*

*Proof.* Let the MFPG estimator be written as $F^{MFPG}(\theta) := \nabla J(\theta) + \xi$, and the high fidelity only policy gradient estimator be written as $F^h(\theta) := \nabla J(\theta) + \tilde{\xi}$. We analyze the sequences $\{\theta_k^{MFPG}\}_{k\in\mathbb{Z}^+}$ and $\{\theta_k^h\}_{k\in\mathbb{Z}^+}$ generated by the two estimators respectively,

$$\theta_{k+1}^{MFPG} = \theta_k^{MFPG} + \alpha_k F^{MFPG}, k \in \mathbb{Z}^+,$$
$$\theta_{k+1}^h = \theta_k^h + \alpha_k F^h, k \in \mathbb{Z}^+,$$
$$\theta_1^h = \theta_1^{MFPG} := \theta_1.$$

$$J(\theta_{k+1}^h) \geq J(\theta_k^h) + \langle \nabla J(\theta_k^h), \theta_{k+1}^h - \theta_k^h \rangle - \frac{L_T}{2} \|\theta_{k+1}^h - \theta_k^h\|^2$$

$$= J(\theta_k^h) + \alpha_k \langle \nabla J(\theta_k^h), F^h \rangle - \frac{\alpha_k^2 L_T}{2} \|F^h\|^2$$

$$= J(\theta_k^h) + \alpha_k \langle \nabla J(\theta_k^h), \nabla J(\theta_k^h) + \xi_k \rangle - \frac{\alpha_k^2 L_T}{2} \|\nabla J(\theta_k^h) + \xi_k\|^2$$

$$= J(\theta_k^h) + \left( \alpha_k - \frac{\alpha_k^2 L_T}{2} \right) \|\nabla J(\theta_k^h)\|^2 + \left( \alpha_k - \alpha_k^2 L_T \right) \langle \nabla J(\theta_k^h), \xi_k \rangle - \frac{\alpha_k^2 L_T}{2} \|\xi_k\|^2$$

$$\implies \left( \alpha_k - \frac{\alpha_k^2 L_T}{2} \right) \|\nabla J(\theta_k^h)\|^2 \leq J(\theta_{k+1}^h) - J(\theta_k^h) - \left( \alpha_k - \alpha_k^2 L_T \right) \langle \nabla J(\theta_k^h), \xi_k \rangle + \frac{\alpha_k^2 L_T}{2} \|\xi_k\|^2$$

$$\implies \sum_{k=1}^{N} \left( \alpha_k - \frac{\alpha_k^2 L_T}{2} \right) \|\nabla J(\theta_k^h)\|^2 \leq J(\theta_{N+1}^h) - J(\theta_1^h) - \sum_{k=1}^{N} \left( \alpha_k - \alpha_k^2 L_T \right) \langle \nabla J(\theta_k^h), \xi_k \rangle$$

$$+ \sum_{k=1}^{N} \frac{\alpha_k^2 L_T}{2} \|\xi_k\|^2$$

$$\leq J(\theta^*) - J(\theta_1) - \sum_{k=1}^{N} \left( \alpha_k - \alpha_k^2 L_T \right) \langle \nabla J(\theta_k^h), \xi_k \rangle + \sum_{k=1}^{N} \frac{\alpha_k^2 L_T}{2} \|\xi_k\|^2.$$

Similarly, for the MFPG iterates, we get

$$\implies \sum_{k=1}^{N} \left( \alpha_k - \frac{\alpha_k^2 L_T}{2} \right) \|\nabla J(\theta_k^{MFPG})\|^2 \leq$$

$$J(\theta^*) - J(\theta_1) - \sum_{k=1}^{N} \left( \alpha_k - \alpha_k^2 L_T \right) \langle \nabla J(\theta_k^{MFPG}), \tilde{\xi}_k \rangle + \sum_{k=1}^{N} \frac{\alpha_k^2 L_T}{2} \|\tilde{\xi}_k\|^2.$$

By taking expectation over the randomness for both sequences, from Assumptions 5 and 6 and the fact that the MFPG is an unbiased estimator for true policy gradient, we get

$$\sum_{k=1}^{N} \left( \alpha_k - \frac{\alpha_k^2 L_T}{2} \right) \mathbb{E}\left[ \|\nabla J(\theta_k^h)\|^2 \right] \leq J(\theta^*) - J(\theta_1) + \frac{\sigma^2 L_T}{2} \sum_{k=1}^{N} \alpha_k^2$$

$$\implies \min_{k \in [T]} \mathbb{E}\left[ \|\nabla J(\theta_k^h)\|^2 \right] \leq \frac{J(\theta^*) - J(\theta_1) + \frac{\sigma^2 L_T}{2} \sum_{k=1}^{N} \alpha_k^2}{\sum_{k=1}^{N} \left( \alpha_k - \frac{\alpha_k^2 L_T}{2} \right)}$$

$$\sum_{k=1}^{N} \left( \alpha_k - \frac{\alpha_k^2 L_T}{2} \right) \mathbb{E}\left[ \|\nabla J(\theta_k^{MFPG})\|^2 \right] \leq J(\theta^*) - J(\theta_1) + \left( 1 - \rho^2 \right) \frac{\sigma^2 L_T}{2} \sum_{k=1}^{N} \alpha_k^2$$

$$\implies \min_{k \in [T]} \mathbb{E}\left[ \|\nabla J(\theta_k^{MFPG})\|^2 \right] \leq \frac{J(\theta^*) - J(\theta_1) + \left( 1 - \rho^2 \right) \frac{\sigma^2 L_T}{2} \sum_{k=1}^{N} \alpha_k^2}{\sum_{k=1}^{N} \left( \alpha_k - \frac{\alpha_k^2 L_T}{2} \right)}.$$

We establish the global asymptotic convergence of the multi-fidelity policy gradient using the arguments developed in Zhang et al. (2020). To show that $\lim_{k \to \infty} \|\nabla J(\theta_k^{MFPG})\| = 0$ almost surely, we first establish that $\liminf_{k \to \infty} \|\nabla J(\theta_k^{MFPG})\| = 0$. Let $J^*$ be the global optimum, we have that $W_k \leq J^*$ by definition of $W_k$. From the stochastic ascent property and corresponding bound established in Lemma 3, we have

$$\mathbb{E}\left[ J^* - W_{k+1} | \mathcal{F}_k \right] \leq (J^* - W_k) - \alpha_k \|\nabla J(\theta_k^{MFPG})\|^2$$

$$\implies \sum_{k=1}^{\infty} \alpha_k \|\nabla J(\theta_k^{MFPG})\|^2 < \infty \text{ almost surely.}$$

(supermartingale convergence theorem (Robbins & Siegmund, 1971) as $J^* - W_k \geq 0$.)

$$\implies \liminf_{k \to \infty} \|\nabla J(\theta_k^{MFPG})\| = 0. \qquad (\because \sum_{k=0}^{\infty} \alpha_k = \infty)$$

Finally, we will establish by contradiction, that $\limsup_{k\to\infty} \|\nabla J(\theta_k^{MFPG})\| = 0$. Assume that $\exists \ \epsilon > 0$ such that $\limsup_{k\to\infty} \|\nabla J(\theta_k^{MFPG})\| = \epsilon$. Thus the set $N_1 = \{\theta_k^{MFPG} : \|\nabla J(\theta_k^{MFPG})\| \geq 2\epsilon/3\}$ is an infinite set. Because $\liminf_{k\to\infty} \|\nabla J(\theta_k^{MFPG})\| = 0$, the set $N_2 = \{\theta_k^{MFPG} : \|\nabla J(\theta_k^{MFPG})\| \leq \epsilon/3\}$ is also an infinite set. We can define the distance between the two sets $D(N_1, N_2) = \inf_{\theta_1 \in N_1} \inf_{\theta_2 \in N_2} \|\theta_1 - \theta_2\|$, which by assumption will be positive. Because both $N_1, N_2$ have an infinite number of sequence members, there must be an index set $\mathcal{I}$ such that the sequence $\{\theta_k^{MFPG}\}_{k\in\mathcal{I}}$ switches between $N_1$ and $N_2$ infinite times, and there must be indices $\{s_i\}_{i\geq 0}, \{t_i\}_{i\geq 0}$ such that $\theta_{s_i} \in N_1, i \geq 0, \theta_{t_i} \in N_2, i \geq 0$, and the iterates $\{\theta_k^{MFPG}\}_{k\in\mathcal{I}} = \{\theta_{s_i+1}^{MFPG}, \dots \theta_{t_i-1}^{MFPG}\}$ are neither in $N_1$ nor in $N_2$. Thus, we have

$$\underbrace{\sum_{i\geq 0} D(N_1, N_2)}_{=\infty} = \sum_i \|\theta_{s_i}^{MFPG} - \theta_{t_i}^{MFPG}\| = \sum_i \left\| \sum_{k=s_i}^{t_i-1} \left( \theta_{k+1}^{MFPG} - \theta_k^{MFPG} \right) \right\|$$

$$\leq \sum_{i\geq 0} \sum_{k=s_i}^{t_i-1} \|\theta_{k+1}^{MFPG} - \theta_k^{MFPG}\|$$

$$= \sum_{k\in\mathcal{I}} \|\theta_{k+1}^{MFPG} - \theta_k^{MFPG}\| \tag{8}$$

We know that

$$\frac{\epsilon}{3} \leq \|\nabla J(\theta_k)\| \ \forall \ k \in \mathcal{I}$$

$$\implies \sum_{k\in\mathcal{I}} \frac{\alpha_k \epsilon^2}{9} \leq \sum_{k\in\mathcal{I}} \alpha_k \|\nabla J(\theta_k^{MFPG})\|^2 < \infty$$

$$\implies \sum_{k\in\mathcal{I}} \alpha_k < \infty$$

$$\implies \sum_{k\in\mathcal{I}} \mathbb{E}\left[ \|\theta_{k+1}^{MFPG} - \theta_k^{MFPG}\| \right] = \sum_{k\in\mathcal{I}} \alpha_k \mathbb{E}\left[ \|\nabla \hat{J}(\theta_k^{MFPG})\| \right] < \infty \qquad \text{(because } \mathbb{E}[\|\nabla \hat{J}\|] \leq \hat{\ell})$$

$$\implies \sum_{k\in\mathcal{I}} \|\theta_{k+1}^{MFPG} - \theta_k^{MFPG}\| < \infty \text{ almost surely,} \qquad \text{(monotone convergence theorem)}$$

which leads us to a contradiction in light of Equation (8), and thus $\limsup_{k\to\infty} \|\nabla J(\theta_k^{MFPG})\| = 0$ almost surely. Thus, we get that $\lim_{k\to\infty} \|\nabla J(\theta_k^{MFPG})\| = 0$.

$\square$

# D   Additional Ablation Study Results

This section presents additional ablation results that complement Section 6.5.

**Reparameterization trick.**   We examine the effect of the reparameterization trick described in Section 4 on our MFPG method. Figure 10 shows evaluation return curves for MFPG with and without the reparameterization trick, along with the High-Fidelity Only baseline, on Hopper-v3 with $0.5\times$ and $1.2\times$ friction shifts. When the dynamics gap is mild (cf., the $1.2\times$ friction setting in Hopper, right panel), reparameterization substantially accelerates learning—for example, around 0.2M steps, the confidence intervals of MFPG with reparameterization lie well above those of its counterpart without reparameterization. In contrast, when the dynamics gap is larger (cf. the $0.5\times$ friction setting, left panel), the difference is much less pronounced.

This pattern is intuitive: when low-fidelity dynamics are close to those of the high-fidelity environment, stochasticity in action selection dominates; reparameterization controls this source of randomness and produces highly correlated rollouts. In the extreme case where the low-fidelity dynamics are perfect (and deterministic), reparameterization makes the low- and high-fidelity rollouts identical, so that $X_{\tau^h}^{\pi_\theta}$ and $X_{\tau^l}^{\pi_\theta}$ cancel out. The policy gradient estimate then reduces to $\mu^l$, which is estimated with abundant (perfect)

low-fidelity samples. By contrast, when the dynamics gap is large, discrepancies in transition dynamics dominate, and even with reparameterization, divergence between high- and low-fidelity rollouts persists, resulting in weaker correlation.

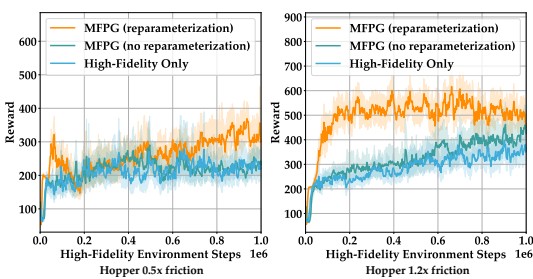

Figure 10: Evaluation return curves for MFPG with and without the reparameterization trick versus High-Fidelity Only on Hopper-v3 with $0.5\times$ (left) and $1.2\times$ (right) friction. Solid lines indicate bootstrap point estimates (means), and shaded regions denote two-sided 95% bootstrap confidence intervals. The reparameterization trick proves especially critical in mild-gap settings (e.g., $1.2\times$ friction).

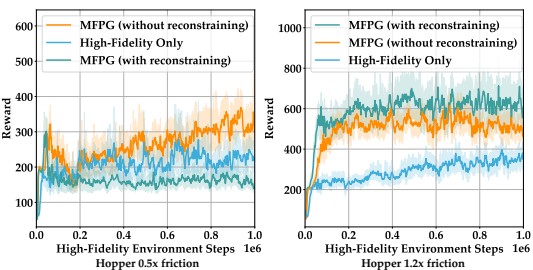

Figure 11: Evaluation return curves for MFPG with and without periodic reconstraining of low-fidelity trajectories versus High-Fidelity Only on Hopper-v3 with $0.5\times$ (left) and $1.2\times$ (right) friction. Solid lines indicate bootstrap point estimates (means), and shaded regions denote two-sided 95% bootstrap confidence intervals. Reconstraining can accelerate learning in some settings—for example, around 0.05M steps in the right panel—but can also harm performance in others, e.g., during the second half of training in the left panel. Accordingly, we do not employ this trick in our evaluations of MFPG.

**Variance-bias trade-off.** In practice, we observed that periodically reconstraining (the correlated) low-fidelity rollouts back to their high-fidelity counterparts can strengthen correlation and accelerate learning in some settings (i.e., every few steps along a low-fidelity rollout, reset the low-fidelity state to match the state of the correlated high-fidelity rollout at the same time step). For example, in the $1.2\times$ friction case in Fig. 11 (right panel), around 0.05M steps, the MFPG variant with periodic reconstraining attains an evaluation return whose confidence interval lies well above that of nominal MFPG without reconstraining. However, this procedure may also introduce bias into the policy gradient estimator, leading to degraded performance in certain cases (e.g., the $0.5\times$ friction setting in Fig. 11, left panel)—the confidence interval of MFPG with reconstraining mostly lies well below that of nominal MFPG without reconstraining throughout the second half of training. These findings highlight a practical variance–bias trade-off. We do not adopt this reconstraining trick in our experiments. Developing systematic, unbiased mechanisms for drawing even more highly correlated samples across environments remains an important direction for future work.

# E   Additional Experimental Results on Off-Dynamics Tasks

For the 15 main task settings in Section 6.3 evaluated with 20 random seeds per method, Fig. 12 reports the mean difference in AUC of each method relative to the High-Fidelity Only baseline (so 0 corresponds to High-Fidelity Only), computed analogously to Fig. 3. Bars show the bootstrap point estimate of the mean difference, and error bars denote two-sided 95% bootstrap confidence intervals. As a complement to Fig. 5, Fig. 13 reports the mean difference in evaluation return between MFPG and the High-Fidelity Only baseline.

Figures 14 and 15 further show box plots of the final evaluation return and AUC for each method and setting across the 20 runs: the center line indicates the median, the box spans the interquartile range (25th–75th percentiles), and whiskers extend to the 5th and 95th percentiles; outliers beyond the whiskers are not shown.

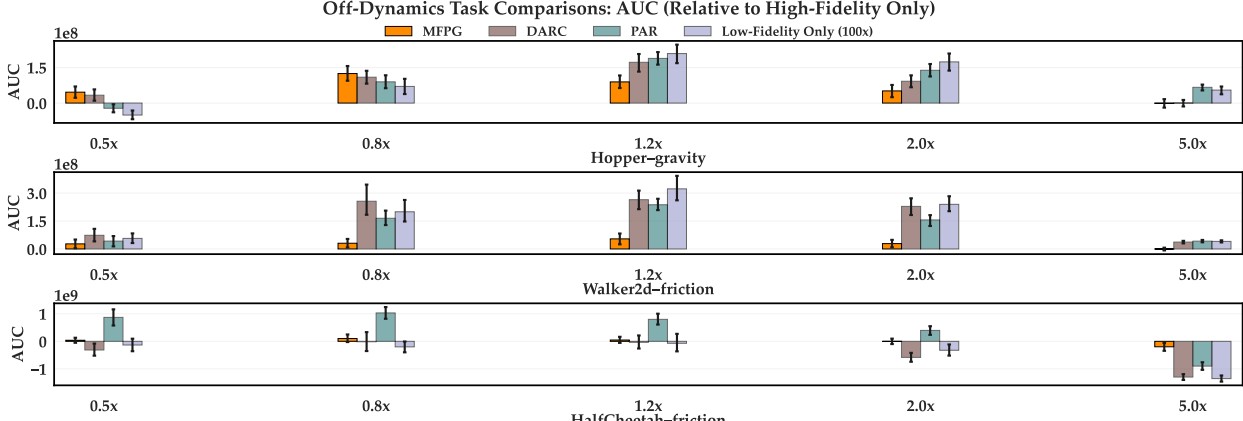

Figure 12: Mean difference in AUC of each method relative to the High-Fidelity Only baseline (0 corresponds to High-Fidelity Only) across the 15 main task settings in Section 6.3, computed analogously to Fig. 3. Bars show the bootstrap point estimate (mean), and error bars denote two-sided 95% bootstrap confidence intervals.

Figures 16 and 17 further reports results on 24 additional task settings, each evaluated with 5 random seeds per method. Due to the small number of seeds, these additional results are primarily used to illustrate the consistency of qualitative trends (e.g., the prevalence of performance collapses), and we do not draw formal statistical conclusions from them. Instead of reporting full distributions, in these figures, bar heights indicate medians and error bars denote the minimum and maximum values across seeds.

In Figs. 16 and 17, kinematic variations are categorized as easy, medium, or hard; each setting constrains a specific joint to a reduced range of motion to mimic actuator damage. For kinematic shifts, we consider cases where the agent's thigh joint is impaired (leg joint for Hopper-v3, since the original benchmark does not include a thigh joint shift for Hopper).

Finally, in Figs. 14 and 16, More High-Fidelity Data (15×) is an reference baseline that trains with access to 15× more high-fidelity samples. Dashed lines denote the median performance of More High-Fidelity Data (15×).

In Fig. 14, we observe that, in general, High-Fidelity Only struggles to match the performance of the More High-Fidelity Data (15×) baseline, highlighting the clear benefit of training on larger high-fidelity sample sizes. Nevertheless, in some settings Low-Fidelity Only (100×) can match or even exceed the More High-Fidelity Data (15×) baseline, albeit often with substantially large cross-seed variance (e.g., Hopper–gravity 1.2×). This does not contradict the role of the More High-Fidelity Data (15×) baseline as a strong reference point: in certain regimes, the dynamics mismatch (bias) can be benign, and abundant (100×) low-fidelity interaction can provide learning signals that transfer unusually well to the high-fidelity environment. In such cases, aggressively exploiting the low-fidelity data can produce large gains, but this behavior is brittle—when low-fidelity data become harmful (cf. HalfCheetah), these approaches can degrade substantially. Moreover, the benefit of low-fidelity data is difficult to predict a priori (e.g., Low-Fidelity Only does not vary monotonically with the dynamics gap). In contrast, MFPG provides a more consistent and reliable mechanism for leveraging low-fidelity data by anchoring updates to high-fidelity gradients while using low-fidelity data only as a variance reduction tool.

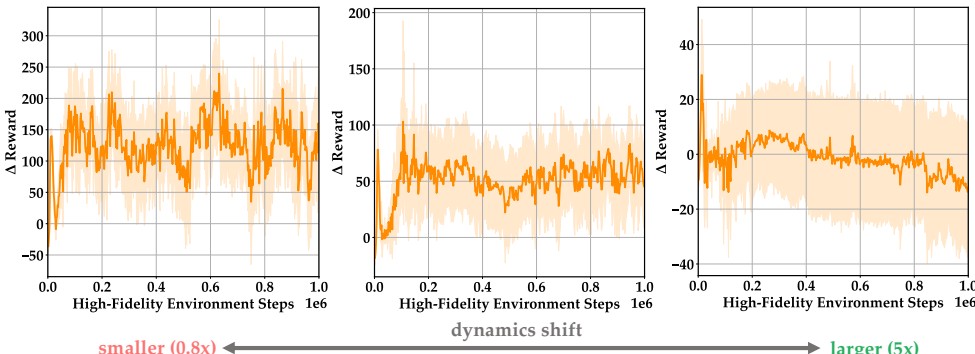

Figure 13: Mean difference in evaluation return of MFPG relative to the High-Fidelity Only baseline (0 indicates no difference) for the settings in Fig. 5. Solid lines show the bootstrap mean estimate, and shaded regions denote two-sided 95% bootstrap confidence intervals.

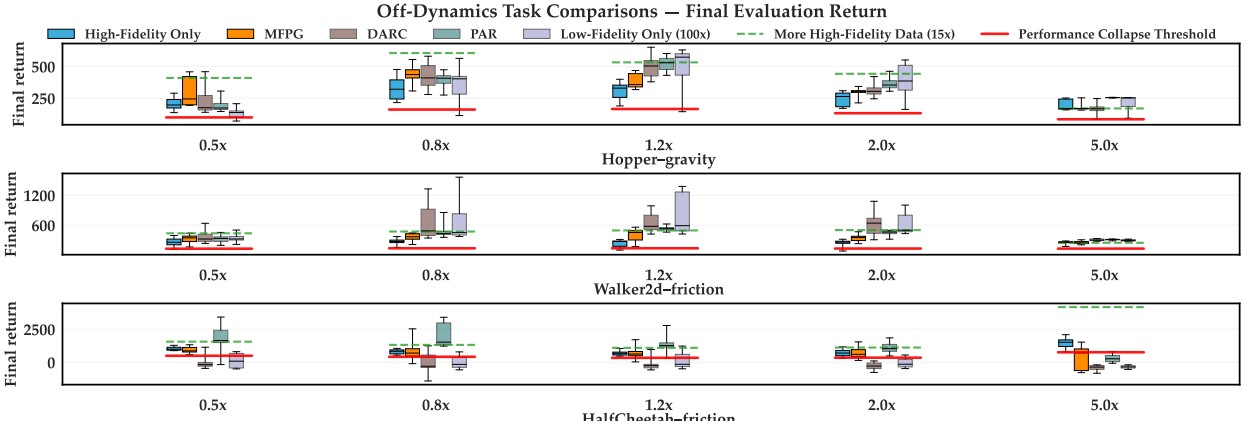

Figure 14: Box plots of the final evaluation return for each method across the 15 main task settings in Section 6.3. The center line indicates the median, the box spans the interquartile range (25th–75th percentiles), and whiskers extend to the 5th and 95th percentiles; outliers beyond the whiskers are not shown.

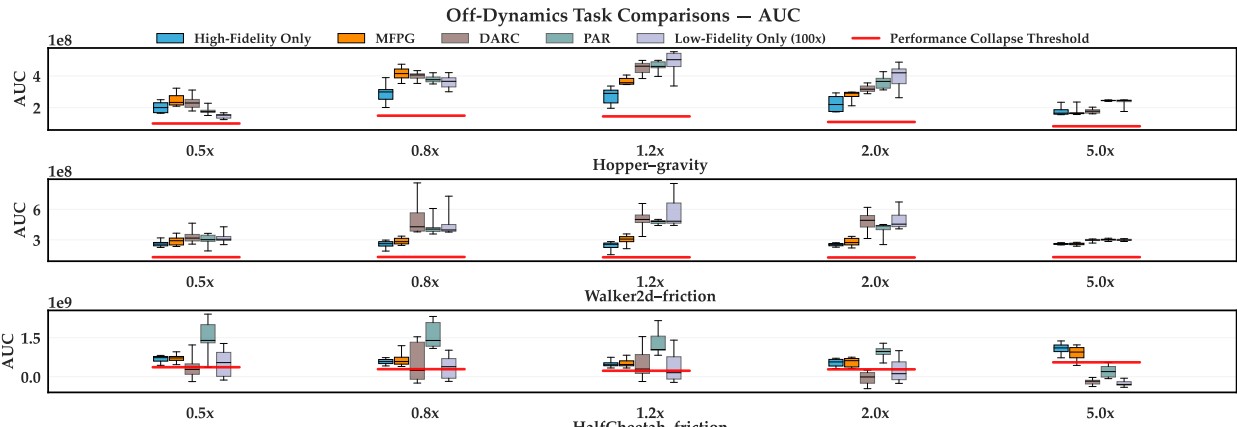

Figure 15: Box plots of the AUC for each method across the 15 main task settings in Section 6.3. The center line indicates the median, the box spans the interquartile range (25th–75th percentiles), and whiskers extend to the 5th and 95th percentiles; outliers beyond the whiskers are not shown.

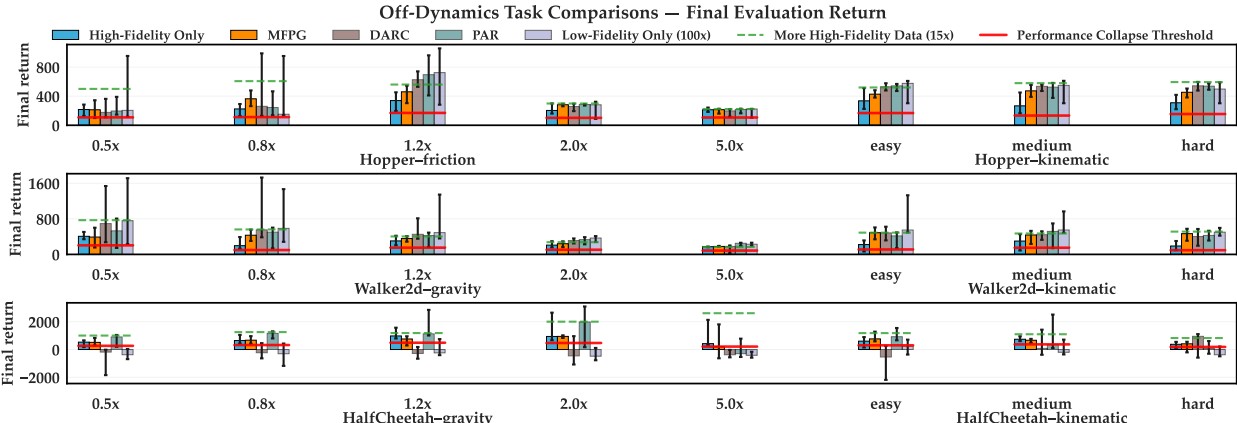

Figure 16: Final evaluation return of each method across an additional 24 task settings. Bar heights indicate medians, and error bars denote the minimum and maximum values across seeds.

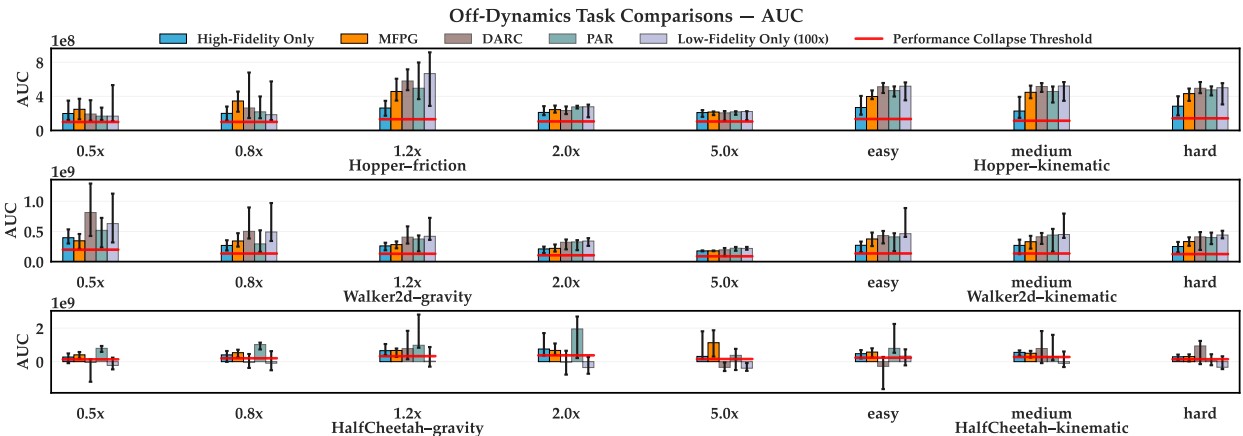

Figure 17: AUC of each method across an additional 24 task settings. Bar heights indicate medians, and error bars denote the minimum and maximum values across seeds.

# F  Implementation Details and Hyperparameter Setup

This section provides implementation details and the hyperparameter setup for all evaluated approaches. Details of the off-dynamics task settings described in Section 6.1 are available in the ODRL benchmark (Lyu et al., 2024b). Table 1 summarizes the hyperparameter configurations for all evaluated methods. To ensure reproducibility, we include our code in the supplementary materials and will release the project as open source upon acceptance.

We build our implementation on top of the RL library Stable-Baselines3 (Raffin et al., 2021). For experimentation, we use random seeds $\{0, 1, 2\}$ for hyperparameter tuning and $\{3, 4, \dots, 22\}$ for evaluation in Sections 6.3 to 6.5 ($\{3, 4, \dots, 7\}$ for the additional 24 tasks in Section E). The High-Fidelity Only baseline is tuned using 3 seeds $\{0, 1, 2\}$. The resulting configuration serves as the shared backbone for the other methods, following standard practice in the off-dynamics RL literature (Xu et al., 2023; Lyu et al., 2024a). All additional hyperparameters are tuned with 2 seeds $\{0, 1\}$, either over more task settings (for the multi-fidelity methods) or longer training steps (for More High-Fidelity Data (15×)). All single-fidelity methods are tuned on the MuJoCo Hopper-v3, Walker2d-v3, and HalfCheetah-v3 tasks with original dynamics, chosen because they represent the midpoint among the varied dynamics settings. Multi-fidelity methods are tuned on the same three tasks under a representative subset of dynamics variations, namely 0.8× and 2.0× gravity. We verified that the tuned hyperparameters generalize well to other variation types and levels (cf. Section E). For the extra sensitive hyperparameters of PAR (reward-augmentation weight) and DARC (the standard deviation of Gaussian noise added to classifier inputs), we additionally tune them on the 0.8× and 2.0× friction variations of the three tasks. We note that this protocol grants the baselines more tuning privilege than MFPG, highlighting the simplicity and minimum tuning overhead of MFPG.

**High-Fidelity Only.**  We implement the standard REINFORCE algorithm (Williams, 1992) with state-value subtraction. Subject to the high-fidelity sample restrictions per policy update described in Section 6.1, we perform grid search over the learning rate $\{1\times10^{-4}, 2\times10^{-4}, 5\times10^{-4}, 7\times10^{-4}, 9\times10^{-4}, 1\times10^{-3}, 2\times10^{-3}\}$, discount factor $\gamma$ $\{0.95, 0.97, 0.98, 0.99, 0.995\}$, advantage normalization $\{$True, False$\}$, maximum gradient norm $\{0.5, 1, 2\}$, and state-value loss weight $\{0.25, 0.5, 1\}$. We adopt the default network architectures for the policy and state-value networks in Stable-Baselines3 (Raffin et al., 2021). The tuned REINFORCE configuration is then adopted as the shared algorithmic backbone for all multi-fidelity methods. The full hyperparameter configuration, along with those for all other methods, is provided in Table 1.

**DARC.**  DARC (Eysenbach et al.)  trains two domain classifiers $q_{\theta_{\mathrm{SAS}}}(\mathrm{high} \mid s_t, a_t, s_{t+1})$ and $q_{\theta_{\mathrm{SA}}}(\mathrm{high} \mid s_t, a_t)$ to predict how likely an observed transition $(s_t, a_t, s_{t+1})$ or state-action pair $(s_t, a_t)$ comes from the high-fidelity environment. The classifiers are trained by minimizing cross-entropy losses over the replay buffers $\mathcal{D}_{\mathrm{high}}$ and $\mathcal{D}_{\mathrm{low}}$ of high- and low-fidelity samples, respectively:

$$\mathcal{L}_{\mathrm{SAS}}(\theta_{\mathrm{SAS}}) = -\mathbb{E}_{\mathcal{D}_{\mathrm{low}}}\left[\log q_{\theta_{\mathrm{SA}}}(\mathrm{low} \mid s_t, a_t, s_{t+1})\right] - \mathbb{E}_{\mathcal{D}_{\mathrm{high}}}\left[\log q_{\theta_{\mathrm{SAS}}}(\mathrm{high} \mid s_t, a_t, s_{t+1})\right]$$
$$\mathcal{L}_{\mathrm{SA}}(\theta_{\mathrm{SA}}) = -\mathbb{E}_{\mathcal{D}_{\mathrm{low}}}\left[\log q_{\theta_{\mathrm{SA}}}(\mathrm{low} \mid s_t, a_t)\right] - \mathbb{E}_{\mathcal{D}_{\mathrm{high}}}\left[\log q_{\theta_{\mathrm{SA}}}(\mathrm{high} \mid s_t, a_t)\right]. \qquad (9)$$

The logits of the trained classifiers are then combined to approximate the dynamics gap, $\log \frac{p(s_{t+1}|s_t,a_t,\mathrm{high})}{p(s_{t+1}|s_t,a_t,\mathrm{low})}$, for each observed (low-fidelity) transition $(s_t, a_t, s_{t+1})$:

$$\Delta r(s_t, a_t, s_{t+1}) = \log p(\mathrm{high} \mid s_t, a_t, s_{t+1}) - \log p(\mathrm{high} \mid s_t, a_t) - \log p(\mathrm{low} \mid s_t, a_t, s_{t+1}) + \log p(\mathrm{low} \mid s_t, a_t). \quad (10)$$

This estimated dynamics gap is used to augment low-fidelity rewards $r^l_{\mathrm{augment},t}(s^l_t, a^l_t, s^l_{t+1}) = r^l_t(s^l_t, a^l_t, s^l_{t+1}) + \Delta r(s^l_t, a^l_t, s^l_{t+1})$. The augmented rewards penalize the agent for exploiting low-fidelity transitions that are implausible in the high-fidelity environment.

We adopt the standard implementation from the ODRL benchmark (https://github.com/OffDynamicsRL/off-dynamics-rl), cross-validated against the authors' original code (https://github.com/google-research/google-research/tree/master/darc). To adapt DARC to the on-policy REINFORCE algorithm, we retain the original classifier-learning and reward-augmentation mechanisms, substituting only the policy optimization backbone. At each update, classifiers are trained using off-policy samples from the high- and low-fidelity replay buffers, as in the original implementation,

by minimizing the cross-entropy losses in Eq. (9). The learned classifiers are then used to augment the rewards, cf., Eq. (10), for low-fidelity, on-policy samples, which are subsequently employed to compute the REINFORCE updates and learn the state-value function.

Building on the shared REINFORCE backbone used by the High-Fidelity Only baseline, we further tune several DARC-specific parameters to which the method is more sensitive: the standard deviation of Gaussian noise added to classifier inputs as a regularizer (an effective stabilization trick noted by the authors (Eysenbach et al.)) $\{0.5, 1, 2\}$, and the number of warm-start, high-fidelity environment steps $\{2000, 10000, 50000\}$ prior to classifier learning and reward augmentation. For other hyperparameters—such as classifier architecture, learning rate, and batch size—we adopt the default values from the original paper (Eysenbach et al.). The resulting classifier training curves are stable.

**PAR.** Similar in spirit to DARC, PAR (Lyu et al., 2024a) also estimates the dynamics gap of low-fidelity transitions for reward augmentation. However, instead of domain classification, PAR adopts a representation-learning approach: it trains a state encoder $z_{1,t} = f_\zeta(s_t)$ and a state–action encoder $z_{2,t} = g_\nu(z_{1,t}, a_t)$ by minimizing a latent dynamics consistency loss over the *high-fidelity* replay buffer:

$$\mathcal{L}(\zeta, \nu) = \mathbb{E}_{(s_t^h, a_t^h, s_{t+1}^h) \sim \mathcal{D}_{\text{high}}} \left[ \| g_\nu(f_\zeta(s_t^h), a_t^h) - \texttt{NoGradient}(f_\zeta(s_{t+1}^h)) \|^2 \right], \tag{11}$$

which essentially learns a dynamics model for the high-fidelity environment in a latent space. The trained encoders are then used to estimate dynamics mismatch for low-fidelity transitions and augment the rewards:

$$r_{\text{augment},t}^l(s_t^l, a_t^l, s_{t+1}^l) = r_t^l(s_t^l, a_t^l, s_{t+1}^l) - \beta \cdot \| g_\nu(f_\zeta(s_t^l), a_t^l) - f_\zeta(s_{t+1}^l) \|^2, \tag{12}$$

where $\beta$ is a weighting hyperparameter. Furthermore, unlike DARC, which trains its policy and critics only on low-fidelity samples with augmented rewards, PAR trains its policy and critics on both high- and low-fidelity samples, with reward augmentation applied to the latter.

We adopt the standard implementation from the ODRL benchmark (https://github.com/OffDynamicsRL/off-dynamics-rl), cross-validated against the authors' original code (https://github.com/dmksjfl/PAR). Analogous to our adaptation of DARC to the on-policy REINFORCE algorithm, we adapt PAR by retaining its original representation-learning and reward-augmentation mechanisms while substituting only the policy optimization backbone. At each update, we draw off-policy, high-fidelity samples from the replay buffer to train the encoders by minimizing Eq. (11). The trained encoders are then applied to augment rewards for on-policy low-fidelity samples, as in Eq. (12). Unlike DARC, which relies solely on augmented low-fidelity samples for policy and value function learning, PAR leverages both on-policy high-fidelity and augmented low-fidelity samples for REINFORCE policy updates and state-value function learning, consistent with the original PAR algorithm.

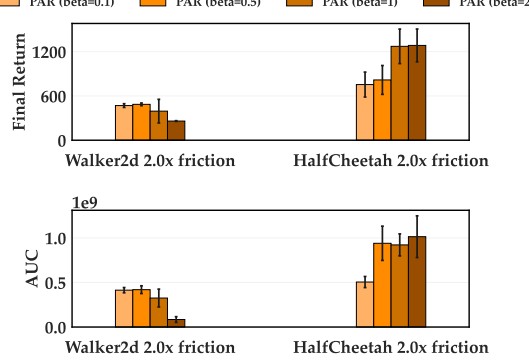

Figure 18: Sweep of the reward-augmentation weight $\beta$ for PAR on two tasks. The relative performance across $\beta$ values varies substantially between tasks. Bar heights indicate medians, and error bars denote the minimum and maximum values.

On top of the shared REINFORCE backbone, we further tune the reward-augmentation weight $\beta$ in Eq. (12) over

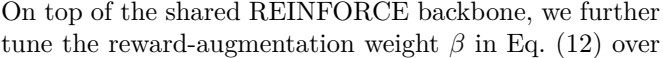

the grid $\{0.1, 0.5, 1, 2\}$, as PAR is particularly sensitive to this parameter; cf. Fig. 18. The original PAR (Lyu et al., 2024a) employs a *per-task* weight for this reason, and we follow the same convention, reporting the selected $\beta$ for each task in Table 1. We note, however, that this requirement introduces a computationally expensive adaptation overhead, unlike other algorithms, which do not require per-task tuning. For other hyperparameters—such as encoder architecture, learning rate, and batch size—we adopt the defaults from the original paper (Lyu et al., 2024a). The resulting encoder training curves are stable.

**MFPG.**  Building on the shared REINFORCE backbone, we perform a grid search over the exponential moving-average weight $\eta_{\mathrm{ma}}$ in Eq. (3) using values $\{0.92, 0.95, 0.99\}$. We find that MFPG's median performance consistently exceeds that of the High-Fidelity Only baseline across the tested values of $\eta_{\mathrm{ma}}$, even though the absolute performance metrics vary with $\eta_{\mathrm{ma}}$ . In off-dynamics task settings, we drop the control variate term $c(X_{\tau^l}^{\pi_\theta} - \hat{\mu}^l)$ whenever the estimated correlation coefficient for the current batch, i.e., $\hat{\rho}_k^{\mathrm{batch}}(X_{\tau^h}^{\pi_{\theta_k}}, X_{\tau^l}^{\pi_{\theta_k}})$ in Eq. (3), is negative. Since high- and low-fidelity samples should in principle be positively correlated in off-dynamics settings, a negative correlation suggests either low-quality samples or noise in correlation estimation. In practice, such cases occur only rarely. This engineering trick helps mitigate noise in estimating the control variate coefficient, which can arise from extreme low-fidelity samples. Developing more principled estimation schemes remains an orthogonal direction for future work.

**Low-Fidelity Only (100×).**  The Low-Fidelity Only baseline serves as an ablation of high-fidelity samples from the multi-fidelity approaches. Accordingly, we adopt the same hyperparameter configuration as the shared REINFORCE backbone, except that each policy and state-value function update is performed using abundant low-fidelity data in place of limited high-fidelity samples.

**More High-Fidelity Data (15×).**  To ensure fair benefit from the additional high-fidelity samples, we tune the sensitive hyperparameters for the oracle baseline More High-Fidelity Data (15×) separately from the shared REINFORCE backbone. Specifically, we tune the learning rate over the grid $\{5 \times 10^{-4}, 7 \times 10^{-4}, 9 \times 10^{-4}, 1 \times 10^{-3}, 2 \times 10^{-3}, 4 \times 10^{-3}\}$ and the batch size (i.e., samples per policy update) over the grid $\{1024, 2048, 3072, 4096, 6144\}$. Because of the expensive, longer training time for this baseline, we first run all configurations for 3 million steps, then select the best 7 configurations and extend them to 15 million steps to determine the final configuration. Other hyperparameters are kept consistent with the shared REINFORCE backbone, as they are relatively insensitive.

**Low-fidelity data amount.**  In baseline comparisons, we assume that low-fidelity samples are cheap to generate, and all multi-fidelity methods may supplement limited high-fidelity data with up to 100× additional low-fidelity samples per policy update. While PAR has been empirically observed to benefit from more low-fidelity data in some task settings (Lyu et al., 2024a), we also found it to be sensitive to the amount of low-fidelity data in certain cases. To maximize baseline performance, we additionally tune the ratio of low- to high-fidelity samples per policy update over the grid $\{10×, 20×, 50×, 100×\}$ for PAR and DARC. For DARC, performance is generally maximized with 100× low-fidelity samples. For PAR, performance is maximized across tasks at 20×. The Low-Fidelity Only baseline is assigned 100× low-fidelity samples to match the budget. We implement parallelized low-fidelity environments, each generating 10× low-fidelity samples per policy update; hence, DARC and Low-Fidelity Only employ 10 environments, while PAR uses 2 environments. For MFPG, we did not tune the amount of low-fidelity samples, since MFPG is in theory stable with respect to low-fidelity sample amounts, so long as the low-fidelity sample amount is sufficient to estimate the low-fidelity sample mean $\hat{\mu}^l$. Our empirical results in Fig. 8 support this point. Therefore, instead of using 10 parallel low-fidelity environments (which would exceed the 100× budget, given that MFPG also requires 1× correlated low-fidelity samples), we adopt 9 parallel environments in all main results, corresponding to 90× low-fidelity samples (plus 1× correlated low-fidelity samples). In some auxiliary variance and sensitivity analysis in Sections 6.2 and 6.5, we use 100× (uncorrelated) low-fidelity samples; we specify the low-fidelity sample amount separately in those sections. Finally, we emphasize that DARC and PAR receive more tuning budget than MFPG, underscoring the simplicity and low adaptation overhead of MFPG.

**Complete hyperparameter configuration.**  Table 1 reports the full hyperparameter configuration for all methods in the baseline comparison results. As noted above, the High-Fidelity Only configuration serves as the shared backbone for other methods, with algorithm-specific hyperparameters listed on top of this backbone. Brown entries indicate hyperparameters that replace the corresponding values in the backbone.

Table 1: Hyperparameter configurations for evaluated methods.

**High-Fidelity Only (REINFORCE backbone)**

| | |
|---|---|
| Optimizer | Adam (Kingma, 2014) |
| Learning rate | $7 \times 10^{-4}$ |
| High-fidelity batch size | 100 |
| Discount factor $\gamma$ | 0.97 |
| State-value loss weight vf_coef | 1.0 |
| Maximum gradient norm | 1.0 |
| Policy network | (64, 64) |
| State value network | (64, 64) |
| Nonlinearity | Tanh |

**DARC**

| | |
|---|---|
| Classifier network | (256, 256) |
| Classifier nonlinearity | ReLU |
| Classifier Gaussian input noise std. $\sigma$ | 0.5 |
| Classifier optimizer | Adam |
| Classifier learning rate | $3 \times 10^{-4}$ |
| Classifier batch size | 128 |
| Replay buffer size (classifier) | $1 \times 10^{6}$ |
| Low-fidelity data amount / policy update | 100× (relative to high-fidelity data) |
| Warm-up high-fidelity steps | 2000 |

**PAR**

| | |
|---|---|
| Encoder network | (256, 256) |
| Encoder nonlinearity | ELU |
| Representation dimension | 256 |
| Encoder optimizer | Adam |
| Encoder learning rate | $3 \times 10^{-4}$ |
| Encoder batch size | 128 |
| Replay buffer size (encoder) | $1 \times 10^{6}$ |
| Polyak averaging coefficient for encoder updates | 0.995 (for the target encoder network) |
| Low-fidelity data amount / policy update | 20× |
| Reward augmentation weight $\beta$ | 0.5 (Hopper), 0.1 (Walker2d), 1.0 (HalfCheetah) |

**MFPG**

| | |
|---|---|
| Low-fidelity data amount / policy update | 90× uncorrelated + 1× correlated |
| Exponential moving-average weight $\eta_{\mathrm{ma}}$ | 0.95 |

**Low-Fidelity Only (100×)**

| | |
|---|---|
| Low-fidelity data amount / policy update | 100× (no high-fidelity data) |

**More High-Fidelity Data (15×)**

| | |
|---|---|
| Learning rate | $9 \times 10^{-4}$ |
| High-fidelity batch size | 3072 |

## G   Extending MFPG to Broader Algorithms and Settings

For the first future-work direction discussed in Section 7, we highlight two subdirections. First, a natural next step is to extend the proposed multi-fidelity control variate approach to a broader class of modern RL algorithms.

A concrete starting point in this direction is modern on-policy actor-critic methods, such as proximal policy optimization (PPO; Schulman et al., 2017). As discussed in Section 4, the MFPG framework can, in principle, be instantiated with other on-policy policy gradient methods by changing how the random variable $X_\tau^{\pi_\theta}$ is computed from sampled trajectories. In this work, we focus on the classic REINFORCE algorithm to simplify the theoretical analysis and establish convergence guarantees. Modern actor-critic methods, however, deliberately trade bias for lower variance in their policy gradient estimates, e.g. via generalized advantage estimation (Schulman et al., 2015). As a result, the high-fidelity policy gradient estimator is typically biased (violating Assumption 5 in Section 5), and extending our convergence analysis to this setting is significantly more challenging. Nevertheless, studying the bias–variance trade-off in this regime may still allow MFPG to yield meaningful variance reduction and performance gain for contemporary on-policy actor-critic methods, and thus merits deeper investigation. Moreover, within the on-policy setting, the proposed MFPG framework could, in principle, be extended in a model-based fashion, where the low-fidelity control-variate random variable $X_{\tau^l}^{\pi_\theta}$ and its sample mean $\hat{\mu}^l$ are computed from a learned dynamics model.

Another important direction is to extend MFPG to off-policy settings. On the one hand, the MFPG framework might be adapted to importance-sampling-based off-policy policy-gradient estimators, e.g. by replacing the high- and low-fidelity random variables with off-policy, importance-weighted policy-gradient estimators. In such settings, variance reduction is arguably even more critical than in on-policy policy-gradient methods (Metelli et al., 2020), making multi-fidelity control variates particularly appealing. On the other hand, if one wishes to extend an MFPG-type approach to modern off-policy algorithms such as soft actor-critic (SAC) (Haarnoja et al., 2018), developing multi-fidelity control variate techniques for off-policy temporal-difference (TD) learning would be an important step. This extension is less direct. In contrast to on-policy policy-gradient methods, where high-variance Monte Carlo returns motivate sophisticated variance-reduction techniques, most modern off-policy algorithms, such as SAC, rely on one-step TD learning and experience replay, which tend to produce relatively low-variance value targets. Their main challenges are primarily bias-related, e.g. value overestimation (Fujimoto et al., 2018), distribution shift (Kumar et al., 2019), and bias accumulation over long horizons (Park et al., 2025). Even so, multi-fidelity control variates may still help reduce noise in bootstrapped targets and the variance of gradient estimates, e.g. in the policy-improvement (policy extraction) steps. In particular, pathwise gradient estimators used in TD-learning–based approaches can still suffer from high variance, depending on the conditioning and smoothness of the learned value function (Mohamed et al., 2020).

A separate line of future work is to extend the multi-fidelity control variate technique to incorporate offline datasets. One scenario is that a large volume of low-fidelity data is available as a static offline dataset, while a smaller amount of high-fidelity data can be gathered online. In principle, one could develop a mechanism similar to the MFPG framework in this work for incorporating such offline (and therefore off-policy) low-fidelity data in a way that aims not to introduce additional bias into the policy gradients estimated from online high-fidelity samples. Concretely, the low-fidelity sample mean $\hat{\mu}^l$ would be estimated from an offline dataset. If, in addition, the offline dataset logs (or allows reconstruction of) the noise samples used for action selection, then one could draw correlated high-fidelity trajectories in the high-fidelity environment via the reparameterization scheme introduced in Section 4 and use the offline low-fidelity trajectories to construct control variates. Another scenario is the standard offline RL setting, in which the high-fidelity samples themselves come from an offline dataset that may be augmented with online low-fidelity simulation data. Extending MFPG to this setting is less direct. The challenges are at least two-fold. First, the specific MFPG framework studied in this work requires the high-fidelity random variable $X_{\tau^h}^{\pi_\theta}$ to be computed from on-policy samples, whereas offline data are inherently off-policy. One might use importance sampling to reweight the offline data and then use online low-fidelity simulation data to reduce the variance of the resulting importance-weighted high-fidelity gradient estimates. Second, extending multi-fidelity control variates to commonly used, TD-learning-based offline RL algorithms (Kumar et al., 2020; Kostrikov et al., 2021) faces challenges similar

to those discussed above for off-policy TD methods: the dominant issues are bias-related, e.g. inaccurate value estimates for out-of-distribution actions and severe distribution shift between the behavior and target policies (Levine et al., 2020).

