# OpenReview forum: "A Multi-Fidelity Control Variate Approach for Policy Gradient Estimation"
_TMLR — Accepted by TMLR_

### Review · Reviewer_KL6A · 2025-11-13

**Summary Of Contributions:**

Summary:

The paper introduces a new way to use data from a low fidelity simulator to augment learning in a higher fidelity environment.  The method uses the data points from the low fidelity simulator not to change the target of the policy gradient algorithm, but to reduce the variance in the update steps.  It does so by adding a ”control variate” term representing the variance to the target function and then weighting that term involving the estimated variances so that the result remains unbiased.  A theoretical result to that effect (and the corresponding convergence result to a local optima) is proven.  Empirical results are then provided comparing the method to two earlier methods that attempt to use low fidelity values in gradient updates by changing the target as well as baselines with only high or low fidelity data.  The new method generally performs on par with the previous methods but is able to succeed in domains where the rewards are anti-correlated between the simulators as well.

Strengths of the paper:
The theoretical results appear correct in the case where variances are exactly known.  I have some concerns about the practical implementation but that doesn’t affect the theoretical result.

The method seems, to me, to be one of the “safer’ uses of lower fidelity data in policy gradient because it manages to not disturb the convergence properties, and has relatively fewer hyper-parameters than other methods.  The empirical results indicate the sensitivity to what hyper-parameters the new method have are not a big concern.

I really like the result in the anti-correlated reward case.  I’m not aware of any other method that would succeed here because they usually assume a positive correlation or at least some sort of bounded dominance between the simulators, but here the math works out that the method can correct the anti-correlated term and use it to its advantage, which seems useful.

**Additional Comments:**

Minor Points:

Alg 1 – The alorithm should have line numbers

Page 15 – bolded capital I in “In this section,”

Page 5 – It appears you are assuming episodes are always finite horizon and that assumption needs to be formalized in the MDP description itself.  If the MDP has unbounded episodes then you need to make gamma \in [0, 1) – that is gamma can’t be 1 as that would allow one to solve NP complete problems.  If instead you are assuming a bounded horizon, the H term should appear in the MDP description.

**Audience:**

Yes

**Audience Explanation:**

Better use of simulated data in RL is a major problem for deploying the field in practical environments and the current method gives a way to do so with fewer hyper-parameters than other methods.

**Broader Impact Concerns:**

I think the Broader Impact statement as written is sufficient.

**Claims And Evidence:**

No

**Claims Explanation:**

The paper makes three main claims and I think two are fully are supported but one of the claims (claim ii) needs to either be made more specific or require more evidence.  In addition, there are statements in the paper that could be construed as claims that need to be cleaned up. See below for details.

1) Variance reduction compared to state-value subtraction.  This seems to be largely covered in the theory and the experiments in Section 6.2

2) The new method outperforms a baseline using high-fidelity simulator data and matches the strongest multi-fidelity approaches.  This claim is partially supported.  I agree the method is outperforming using only high-fidelity data but the small number of replicas in the experiments (see additional comments in sections below) makes it really hard to say anything (even being on par) between the MF methods.  In addition, the paper does not compare to one of the main classes of algorithms they cite in the related work (those that change exploration or simulator usage on the fly) so I think the wording of the claim should not compare to “matching the strongest multi-fidelity methods” but instead "the multi-fidelity methods that make policy gradient adjustments” or "Off-Dynamics RL methods" (the authors' term for this class of methods).  Claiming a comparison to the full set of multi-fidelity methods isn’t backed by evidence.

2b) On the fewer number of collapses experienced by the new method, I agree the evidence is much stronger due to the aggregate statistics reported across experiments.

3) MFPG performs well even with reward misspecification.  I actually think the authors are under-claiming here.  The method manages to deal with anti-correlated rewards between simulators, and and I would add that specific example to the claim itself.

In addition to the claims above, the authors need to be more careful about certain statements in the paper that sound like claims but are actually left to future work.  A casual reader might think the authors are saying that these extensions are obvious or covered in the appendix, but they are not and the authors need to be more clear that they have no evidence that the following statements are true:

Page 5: We remark that… This whole paragraph should be reworked as a list of **limitations** of the current work.  There is no solution here that can deal with multiple simulators (which many MFRL methods do), deal with different state  / actions (again others do) , infinite horizons (again others do), or different discount factors.  The paragraph makes it seem like these are in play but there is no evidence here.

Page 9: Similarly, the inability to deal with Ant3 should be brought up at the beginning as a constraint / limitation of this method.  If you require the ability reset the simulator to arbitrary states at any time, this should be stated in the environment descriptions up front, not relegated to a footnote in the empirical section.

All of those should be reworded to make it clear that these are **limitations** of the current approach and are left to future work without evidence presented here.

**Requested Changes:**

Overall this is a promising new method and I don’t think there is anything completely disqualifying in the idea or execution. However, I have some concerns that need to be addressed about (1) the wording around the limitations that I covered in the claims section, (2) the formal descriptions of the policies on each step (3) the way the omega term is specified in the algorithm / text (4) the wording / comparisons in claim 2 and (5) the lack of replicas.  Each of these seems like they could be resolved, but as they currently stand, they make the paper not fully support (or fully clarify) the claims and assumptions.

(1) was covered above - just reword the pseudo-claims as limitations.  This is crucial but should be straightforward.

(2) The terminology and variables used to describe policies \pi_\theta seem imprecise and may have implications on the theoretical results.  I am concerned the author’s formulation in equation 2 uses  pi_theta interchangeably between the old/new terms.  But \theta is updated between old and new, so really what you have here is \pi_theta_k and \pi_theta_k+1, which I do see used in the appendix proof of Theorem 1, but not in the discussions in the main paper.  The authors need to fix this terminology discrepancy and make sure none of the derivation assumes \pi_old = \pi_new.  This is important but hopefully straightforward and would give greater confidence in Claim 1.

(3) The omega term introduced in the algorithm is impractical to implement the way the algorithm is written.  I scratched my head when I saw that because there is no way to guarantee two simulators have the same noise models between two different transition functions.  The authors avoid this with a reparameterization trick, which seems ok to me, but they ought to be up-front that they are going to do so and state clearly that sampling omega is impractical.  Furthermore, the pseudocode should have the reparameterization trick in it so that future users can follow the implementation rather than trying to build the oracle function currently in there.

(4) The empirical results currently only partially support Claim ii as written.  First, while the authors list several classes of MFRL approaches, they only empirically compared against two “Off Dynamics RL” algorithms, DARC and PAR, while there are no comparisons to the “MFRL” methods mentioned in the paper.  The authors state that those methods use “new training paradigms or exploration strategies”, which is true, but why can’t you compare to one of those methods?  As it stands, claim (ii) states that MFPG “matches the strongest multi-fidelity methods” but that is not the case if you’re not comparing to a whole class.  The claim could be made true by referring to “Off dynamics RL methods” or “multi-fidelity methods that adjust the gradient updates” or something more specific like that. This is crucial but could be just a wording change to the claim unless you want to add more competitors to the experiments.

(5) Finally, the empirical results are all based on a small number of replicas (5 for each bar).  While reporting min/median/max for 5 replicas would be fine if the bars were clearly separated, in all cases except for the mis-specified reward one, the bars heavily overlap, and sometimes the new method has better/worse upper/lower performance than the competitors, and sometimes the opposite.  Given the small number of samples, it’s hard to say much about how these methods compare, although aggregate statistics across the domains like those you did for catastrophic failure are more reasonable (and indeed that claim seems justified). I suggest either running some more replicas or reporting more aggregated statistics so the trends here are clearer.  I also think the authors should say something about why the reduced-variance updates (from Claim 1) are not leading to any actual performance difference in the new algorithm.  Why aren’t the results coming in with lower variance or outperforming methods that have higher variance in their updates?  The crucial part of (5) though is either to add more replicas or to report more "aggregated across domains" statistics to increase the confidence in the claim.

---

> ### Comment · Reviewer_KL6A · 2025-12-12
> **other reviews**
>
> I have read the other reviews now and I largely agree with  hz18 's comments about the small number of replicas in the empirical results.  I pointed out the same thing in my review.  I'm more forgiving of the min/median/max reporting when the distributions clearly don't overlap (say the maxs of one are all below the mins of another) but that's just not the case here.

---

> ### Author Response · Authors · 2025-12-16
> **Authors' response [Part 1]**
>
> We thank the reviewer for the kind summary and the helpful, constructive feedback. We have revised the manuscript to address all comments, and the reviewer’s feedback helped strengthen both the rigor of our empirical evaluation and the overall quality of our manuscript. Below, we summarize the changes in the order of the “Requested Changes” section of the review (all modifications are highlighted in blue in the revised manuscript).
>
>
> ```
> 1. Rewording the pseudo-claims as limitations: "In addition to the claims above, the authors need to be more careful about certain statements in the paper that sound like claims but are actually left to future work. A casual reader...make it clear that these are limitations of the current approach and are left to future work without evidence presented here."
> ```
>
> We agree that these points should be explicitly framed as limitations and as directions for future work that are not addressed in this paper.
>
>
> * We have moved the discussion of these points (multiple low-fidelity environments, differing state/action spaces, different initial distributions or discount factors, infinite-horizon settings, and simulators that do not support resets) into a dedicated “**Limitations & future work**” paragraph at the end of the Conclusions section. There, we explicitly present them as limitations and speculative directions that we leave to future work, and not as current capabilities of MFPG. This paragraph now begins: “Third, our analysis and implementation make several simplifying assumptions, and extending the MFPG framework to more general settings is of great importance...”
>
> * (i) We now explicitly state that MFPG requires the low-fidelity simulator to be resettable to user-specified states on demand, so that we can construct correlated low-fidelity trajectories whose initial states match those of the high-fidelity rollouts. This assumption is highlighted in a dedicated “**Assumptions on the multi-fidelity environments**” paragraph in the Preliminaries section when we introduce the multi-fidelity MDPs. (ii) In addition, we have moved the discussion of the Ant-v3 environment into the main text in the “Task settings” paragraph when we present the experiment setup.
>
> ```
> 2. Policy notation and use of `\pi_\theta`: "The terminology and variables used to describe policies \pi_\theta seem imprecise and ... This is important but hopefully straightforward and would give greater confidence in Claim 1."
> ```
>
> We agree that the original notation could blur the distinction between old and new policies.
>
> * At the beginning of the "Multi-Fidelity Policy Gradients" section, we have added an "**Iteration notation**" paragraph. We now explicitly consider an iterative procedure with parameters $\theta_k$ at iteration k, and we write $\pi_{\theta_k}$ for the corresponding policy. We explain that when discussing an arbitrary but fixed iteration with a fixed parameter vector, all quantities should be understood as evaluated at that iteration, and we omit the subscript k for readability.
>
> * We have removed the “old” / “new” policy notation and now use iteration-indexed notation $\pi_{\theta_k}$ consistently throughout the manuscript. More specifically, we (i) updated Equations (3) and (4) (Equation (2) in the original manuscript) to use $\pi_{\theta_k}$ (and $\pi_{\theta_{k-1}}$ for the previous policy), (ii) revised the paragraph beginning “At every policy gradient step k...” below Equation (4) to use iteration-indexed notation, and (iii) updated Algorithm 2 so that all policies and updates are written in iteration-indexed form.
>
> * We carefully checked the manuscript and confirmed that this was purely a notational issue: none of the theoretical derivations assume $\pi_k = \pi_{k+1}$.

---

> ### Author Response · Authors · 2025-12-16
> **Authors' response [Part 2]**
>
> ```
> 3. Sampling outcome `\omega` and reparameterization trick: "The omega term introduced in the algorithm is impractical to implement ... build the oracle function currently in there."
> ```
>
> We appreciate the reviewer’s concern about the practicality of the $\omega$ term as originally presented and agree that this part of the presentation needed to be clarified. We apologize for the earlier confusion. In the "Multi-Fidelity Policy Gradients" section, we have accordingly revised the paragraphs “Sampling correlated trajectories from the multi-fidelity environments” and “Correlated action sampling via policy distribution reparameterization”:
>
> * We now emphasize upfront that the joint outcome $\omega$ is introduced purely as an analytical device for conceptual clarity. We explicitly state that sampling and coupling the full joint outcome $\omega$ across fidelities is infeasible, since the stochasticity of environment transitions and rewards is not directly controllable. Instead, we clarify which randomness is actually shared across fidelities: MFPG couples only the initial-state randomness and the policy randomness across fidelities and treats transition and reward outcomes as independent. We reiterate this point where relevant in the surrounding text. This makes it clear that MFPG does not assume, nor require, any shared transition or reward outcome across fidelities.
>
>
> * We now introduce the use of the reparameterization trick to enforce coupled policy outcomes across fidelities as soon as we present the notion of policy outcomes. Upon introducing the policy outcomes $\omega^{\pi}$, we clarify that “in practice, this policy outcome sequence is realized as a sequence of per-timestep action-noise samples used to reparameterize the policy’s action sampling at each time step (see the discussion on policy reparameterization below).” We have also revised the surrounding text to consistently and explicitly refer to policy action-noise samples.
>
> * In Algorithm 1, we have revised the pseudocode to incorporate the reparameterization trick, explicitly sampling action-noise sequences rather than invoking an oracle that directly supplies coupled policy outcomes. This now matches our practical implementation.
>
> * Since Algorithm 1 now explicitly relies on the reparameterization trick, we have moved its presentation to follow the reparameterization paragraph, so that the algorithm appears after the necessary concepts have been introduced.
>
> Taken together, these changes clarify the conceptual role of $\omega$ and make MFPG’s sampling scheme and practical implementation clear for future users.
>
> ```
> 4. Claim (ii) wording: "The empirical results currently only partially support Claim ii as written. First, while the authors...This is crucial but could be just a wording change to the claim unless you want to add more competitors to the experiments."
> ```
>
> We thank the reviewer for pointing this out and agree that the previous wording of claim (ii) (“matching the strongest multi-fidelity methods”) should be revised to more precisely reflect the comparison set in our experiments. We have therefore updated claim (ii) to refer only to “strong off-dynamics RL methods” in all occurrences and adjusted the surrounding text accordingly. In addition, based on the comment below and the new experimental results and statistical analysis, we have further refined claim (ii), as detailed next.

---

> ### Author Response · Authors · 2025-12-16
> **Authors' response [Part 3]**
>
> ```
> 5. Number of replicas, aggregate statistics: "Finally, the empirical results are all based on a small number of replicas (5 for each bar). While reporting ... either to add more replicas or to report more "aggregated across domains" statistics to increase the confidence in the claim."
> ```
>
> We appreciate the reviewer’s comment regarding the number of seeds and agree that additional evidence is needed to substantiate the claim. We have made the following changes:
>
> * **More random seeds**: We increased the number of runs for the baseline comparisons (Section 6.3), the negated-reward experiment (Section 6.4), and the additional experiments and ablation studies (Section 6.5) to 20 random seeds per method per task setting.
>
> * **Statistical confidence**: To rigorously assess the evaluated methods, we now report two-sided 95% nonparametric bootstrap confidence intervals (R = 10,000 resamples over seeds/runs) for the means of our primary metrics (final return and AUC).
>
> * **Significance testing**: To verify our claim, for the baseline comparisons in Section 6.3 we bootstrap the mean improvement of each approach relative to the High-Fidelity Only baseline ($\Delta$ = mean(method) - mean(High-Fidelity Only)). We declare a statistically significant improvement over High-Fidelity Only at $\alpha = 0.05$ when the 95% confidence interval for $\Delta$ lies strictly above 0.
>
> * **Figures and text**: Figures that previously used max/min bands have been updated to show bootstrap confidence intervals (error bars for bar plots and shaded regions for learning curves), and we revised the surrounding discussions to only make statements that are supported by significance testing.
>
> * **Empirical claim**: We refined the empirical takeaway (Claim/Insight I2). With the strengthened statistical analysis, our key finding holds and we have revised our claim (ii) accordingly (I2, Reliability and robustness):
>
> MFPG provides a reliable and robust way to exploit low-fidelity data. In our baseline comparison experiments in Section 6.3, for scenarios where low-fidelity data are neutral or beneficial (Hopper-gravity, Walker2d-friction) and dynamics gaps are mild to moderate (0.5× to 2.0×), MFPG is, among the evaluated multi-fidelity and low-fidelity-only approaches, *the only method* that consistently and statistically significantly improves the mean performance over a baseline trained solely on high-fidelity data in *8 out of 8* scenarios. When low-fidelity data become harmful, MFPG exhibits the strongest *robustness* against performance degradation among the evaluated multi-fidelity and low-fidelity-only methods. Strong off-dynamics RL methods tend to exploit low-fidelity data more aggressively, and can achieve higher performance than MFPG at times (e.g., in Walker2d) but can also fail substantially more severely (e.g., in HalfCheetah).
>
> * **Aggregated statistics and robustness**: As suggested, we have retained and clarified the cross-setting robustness summary (performance collapses) and now use it as a complementary aggregated metric alongside the per-setting plots.
>
> * Regarding the review comment "I also think the authors should say something about why the reduced-variance updates (from Claim 1) are not leading to any actual performance difference in the new algorithm.":
>
> Our revised empirical results show benefits of the proposed approach and performance differences between the evaluated methods; see the revised claim (ii) on reliability and robustness above. The evaluated strong off-dynamics RL methods tend to exploit low-fidelity data more aggressively. We have reflected these performance differences in the revised claim (ii). Furthermore, in Section 6.3, we added explanatory text:  "We emphasize that this behavior of MFPG is by design: MFPG uses low-fidelity data solely as a variance-reduction tool, deliberately trading potentially aggressive (but risky) gains for reliability. This design choice underlies the consistency observed here and the strongest robustness that MFPG exhibits relative to the other evaluated multi-fidelity methods in the HalfCheetah-v3 task presented below."
>
>
> These changes primarily affect Section 6 (especially Sections 6.1, 6.3, 6.4, and 6.5, with Section 6.3 most substantially), with small adjacent updates in the Abstract, Introduction, and Conclusion to match the refined empirical statement.

---

> ### Author Response · Authors · 2025-12-16
> **Authors' response [Part 4]**
>
> ```
> Additional comment: MFPG performs well even with reward misspecification. I actually think the authors are under-claiming here. The method manages to deal with anti-correlated rewards between simulators, and and I would add that specific example to the claim itself.
> ```
>
> We thank the reviewer for this encouraging remark. In the revised manuscript, we have strengthened claim (iii) and its references to explicitly mention the anti-correlated reward experiment. In particular, we now write: “An additional experiment, in which the high- and low-fidelity environments are assigned anti-correlated rewards, shows that MFPG can remain effective even when the low-fidelity environment exhibits reward misspecification.”
>
> ```
> Additional comment: "Alg 1 – The algorithm should have line numbers"
> ```
>
> We have added line numbers to both Algorithms 1 and 2.
>
>
> ```
> Additional comment: "Page 15 – bolded capital I in “In this section,”"
> ```
>
> We thank the reviewer for pointing this out. We have double-checked the LaTeX source and We could not reproduce this on our end; nevertheless we adjusted the formatting in that sentence to avoid any ambiguity.
>
>
> ```
> Additional comment: "Page 5 – It appears you are assuming episodes are always finite horizon and that assumption needs to be formalized in the MDP description itself. If the MDP has unbounded episodes then you need to make gamma \in [0, 1) – that is gamma can’t be 1 as that would allow one to solve NP complete problems. If instead you are assuming a bounded horizon, the H term should appear in the MDP description."
> ```
>
> We thank the reviewer for catching this. In the Preliminaries section of the revised manuscript (the “Modeling the multi-fidelity environments” paragraph), we now explicitly formalize that both the high- and low-fidelity environments are finite-horizon MDPs and include the horizon in the MDP tuples.
>
> We thank the reviewer again for their detailed and constructive feedback. We would be happy to address any further questions.

---

### Review · Reviewer_bHiF · 2025-11-17

**Summary Of Contributions:**

This paper proposes Multi-Fidelity Policy Gradients (MFPG), a reinforcement learning framework that efficiently leverages abundant, cheap data from low-fidelity simulators alongside a small amount of high-fidelity data to create an unbiased, variance-reduced policy gradient estimator, enabling more sample-efficient training and robust performance even with significant dynamics gaps. The authors also provide some theoretical analysis of the MFPG.

The contributions are: (1) the MFPG framework for constructing unbiased, low-variance policy gradient estimators; (2) a practical multi-fidelity variant of the REINFORCE algorithm with proven asymptotic convergence guarantees and improved finite-sample convergence rates; and (3) empirical validation demonstrating MFPG's performance gains, robustness to dynamics gaps, and effectiveness under reward misspecification across simulated robotics benchmarks.

**Additional Comments:**

There are some typos in the manuscript, e.g.,
- On page 5, $\gamma \in(0,1]$?
- On page 8,  “goodness"

I do not have further comments. Overall, I think this paper is solid and the requested changes are generally easy to resolve. I recommend a clear acceptance of this paper.

**Audience:**

Yes

**Audience Explanation:**

This paper mainly discusses off-dynamics RL. This field is strongly related to reinforcement learning, and to the best of my knowledge, TMLR has accepted numerous RL papers. Second, this paper is related to policy adaptation in RL, which should be interesting to numerous RL researchers, especially that there are numerous off-dynamics RL papers accepted in top-tier AI conferences like ICML, ICLR, and NeurIPS

**Broader Impact Concerns:**

I think the authors fully address the potential broader impact concerns, and they include the broader impact statement in the manuscript

**Claims And Evidence:**

Yes

**Claims Explanation:**

The authors provide some theoretical analysis (though with some comparatively strong assumptions) to validate the proposed MFPG framework, which is commendable. Furthermore, the authors conduct numerous experiments in Section 6, demonstrating the effectiveness of their approach against some prior methods. It should also be noted that the authors tune the hyperparameters of baseline methods instead of reporting the raw performance without any tuning. This highlights that the advantages of the proposed MFPG framework when compared against baselines.

**Requested Changes:**

1. I think the authors have included numerous off-dynamics RL papers as references. However, since this is a growing area, this paper can still benefit from including more related off-dynamics RL papers [1-3]. Also, there are numerous off-dynamics RL papers submitted to ICLR 2026
2. I understand that this paper mainly focuses on the online off-dynamics RL setting, but it still raises questions on whether the proposed method can be extended to other scenarios, e.g., the source environment is offline while the target environment is online. I do not mean that the authors should conduct experiments for this, but I think it would be better to discuss this in the revised manuscript. That being said, I would like to see some clarification on how the proposed method can be extended to other scenarios. If it could, how, and if it could not , why.
3. The authors provide a long analysis on Page 6, but comment that "However, practically, we only explicitly sample values for the policy outcomes". That kind of makes the previous analysis and statement weak, posing a gap between them. It would be helpful to clearly state the implementation choice and the gap between the analysis and the practical implementation
4. Assumption 5 in the theoretical analysis part is a bit strong. When the high-fidelity data is scarce, how can one ensure that the gradient is unbiased? It would be better to clarify this either in the main text or the appendix
5. The authors comment that they "construct an unbiased, variance-reduced estimator" in the abstract part, but there is no corresponding theoretical analysis of these properties. There is also no empirical evidence on the unbiasedness. The authors should fix such statements
6. The authors comment that "To isolate the effect of their multi-fidelity mechanisms from the backbone algorithm and enable fair comparison, we adapt both PAR and DARC to the REINFORCE backbone". It raises questions on whether the proposed MFPG framework can be extended to the scenario when SAC is adopted as the backbone. The authors do not need to conduct experiments on this, but should add some discussions either in the main text or the appendix
7. Figure 9 shows the parameter study of the introduced hyperparameter $\eta_{\rm ma}$ for the moving average. It turns out that the MFPG framework can be sensitive to different choices of $\eta_{\rm ma}$, while the authors comment on Page 10 that "and found the performance to be insensitive to this choice". This should be clarified and fixed if necessary

## References

[1] Cross-domain offline policy adaptation with optimal transport and dataset constraint. ICLR 2025

[2] Policy Learning for Off-Dynamics RL with Deficient Support. AAMAS

[3] MOBODY: Model Based Off-Dynamics Offline Reinforcement Learning. ArXiv

---

> ### Author Response · Authors · 2025-12-16
> **Authors' response [Part 1]**
>
> We appreciate the reviewer’s encouraging summary and comprehensive, helpful feedback. We have thoroughly updated the manuscript to incorporate these suggestions, and we believe the feedback has
> allowed us to improve the overall quality of our work. Our responses follow the order of the "Requested Changes" section of the review, and all edits are highlighted in blue in the revised manuscript.
>
> ```
> 1. Additional off-dynamics RL references: "I think the authors have included numerous off-dynamics RL papers as references. However, since this is a growing area, this paper can still benefit from including more related off-dynamics RL papers [1-3]. Also, there are numerous off-dynamics RL papers submitted to ICLR 2026"
> ```
>
> We thank the reviewer for these recommendations. We have expanded the Related Work section (specifically the "Off-dynamics RL" subsection) to include the suggested references as well as other recent work in this growing field. Specifically, we have added discussions on:
>
> * Linh Le Pham Van, Hung The Tran, and Sunil Gupta. Policy learning for off-dynamics RL with deficient support. In Proceedings of AAMAS, 2024.
>
> * Jiafei Lyu, Mengbei Yan, Zhongjian Qiao, Runze Liu, Xiaoteng Ma, Deheng Ye, Jing-Wen Yang, Zongqing Lu, and Xiu Li. Cross-domain offline policy adaptation with optimal transport and dataset constraint. In Proceedings of ICLR, 2025.
>
> * Yihong Guo, Yu Yang, Pan Xu, and Anqi Liu. MOBODY: Model based off-dynamics offline reinforcement learning. arXiv preprint arXiv:2506.08460, 2025.
>
> * Anonymous. Cross-domain offline policy adaptation via selective transition correction. Under review for ICLR, 2026.
>
> ```
> 2. Discussion on extending MFPG to other scenarios (e.g., offline source, online target): "I understand that this paper mainly focuses on the online off-dynamics RL setting, but it still raises questions on whether the proposed method ... clarification on how the proposed method can be extended to other scenarios. If it could, how, and if it could not , why."
> ```
>
> We have added a detailed discussion regarding the extension of MFPG to other settings, such as offline RL, in Appendix F, titled “**Extending MFPG to Broader Algorithms and Settings**.” In the last paragraph of that appendix, we now discuss:
>
> * Online high-fidelity data + offline low-fidelity data: We outline how the same multi-fidelity control variate idea could, in principle, be adapted to a setting where abundant low-fidelity data are available only as a fixed offline dataset, while limited high-fidelity data are gathered online. The goal is to reduce the variance of the policy gradient estimates based on online high-fidelity samples without introducing additional bias.
>
> * Offline high-fidelity data: (i) If one directly applies the MFPG framework developed in this work, importance sampling (or a similar correction) would be needed to reweight offline high-fidelity data when computing the high-fidelity random variable, since our current framework is on-policy. (ii) Extending MFPG to standard TD-learning–based offline RL algorithms faces the bias-related challenges inherent in combining multi-fidelity control variates with off-policy TD learning; we discuss these challenges in more detail below in our response to Comment 6.
>
> In addition, Appendix F also discusses how MFPG might be extended to on-policy actor–critic algorithms, model-based algorithms, and off-policy algorithms (with further details again deferred to our response to Comment 6).

---

> ### Author Response · Authors · 2025-12-16
> **Authors' response [Part 2]**
>
> ```
> 3. Implementation choice and gap between analysis and practical implementation: "The authors provide a long analysis on Page 6, but comment that "However, practically, we only explicitly sample values for the policy outcomes". That kind of makes the previous analysis and statement weak, posing a gap between them. It would be helpful to clearly state the implementation choice and the gap between the analysis and the practical implementation."
> ```
>
> We appreciate the reviewer’s concern and agree that this part of the presentation needed clarification.
>
> At a high level, the gap lies in which sources of randomness are actually controllable by the algorithm. In our setting, when sampling trajectories from an MDP, the sources of stochasticity are: (i) initial states, (ii) stochastic policies, (iii) environment transitions, and (iv) rewards. Ideally, one would like to couple the sampling outcomes for *all* of these sources across fidelities. However, the randomness from (iii) and (iv) is typically not directly controllable. Consequently, MFPG couples only (i) (by resetting the low-fidelity simulator to match the initial state of the high-fidelity trajectory) and (ii) (via the reparameterization trick) across fidelities. This is why only (ii) requires explicit sampling of policy outcomes (action-noise samples). The introduction of the joint outcome $\omega$ over all randomness sources is mainly for conceptual clarity in the analysis.
>
> To communicate this more clearly, in the "Multi-Fidelity Policy Gradients" section we have revised the paragraphs “**Sampling correlated trajectories from the multi-fidelity environments**” and “**Correlated action sampling via policy distribution reparameterization**” as follows:
>
> * We now clarify upfront that the joint outcome $\omega$ serves purely as an analytical device for conceptual clarity. We explicitly state that sampling and coupling the full joint outcome $\omega$ across fidelities is practically infeasible, as the stochasticity of environment transitions and rewards cannot be directly controlled. Instead, we clarify that MFPG couples only the initial-state randomness and the policy randomness across fidelities, while treating transition and reward outcomes as independent. We reiterate this point throughout the relevant text to ensure it is clear that MFPG neither assumes nor requires shared transition or reward outcomes across multi-fidelity environments.
>
> * We now introduce the reparameterization trick---used to enforce coupled policy outcomes across fidelities---immediately upon defining policy outcomes. When introducing $\omega^{\pi}$, we clarify that “in practice, this policy outcome sequence is realized as a sequence of per-timestep action-noise samples used to reparameterize the policy’s action sampling at each time step (see the discussion on policy reparameterization below).” The surrounding text has also been updated to consistently and explicitly refer to policy action-noise samples utilized by the reparameterization trick.
>
> * We have revised the pseudocode in Algorithm 1 to explicitly incorporate the reparameterization trick. Instead of invoking an oracle to sample coupled policy outcomes as originally written, the algorithm now samples action-noise sequences, aligning with our practical implementation.
>
>
> * Given that Algorithm 1 now explicitly relies on the reparameterization trick, we have moved its presentation to follow the paragraph on reparameterization, so that the algorithm appears after the necessary concepts have been introduced.
>
> Collectively, these revisions clarify the conceptual role of the joint probability space and sampling outcomes, ensuring that MFPG’s sampling scheme and practical implementation are clear for future users.

---

> ### Author Response · Authors · 2025-12-16
> **Authors' response [Part 3]**
>
> ```
> 4. Assumption 5 (unbiasedness of the high-fidelity policy gradient estimator): "Assumption 5 in the theoretical analysis part is a bit strong. When the high-fidelity data is scarce, how can one ensure that the gradient is unbiased? It would be better to clarify this either in the main text or the appendix."
> ```
>
> We thank the reviewer for catching this. As originally written, Assumption 5 could be slightly misleading, and we apologize for the confusion.
>
> Assumption 5 is intended to be an assumption on the underlying high-fidelity policy gradient *estimator*, rather than on any particular finite-sample estimate. Under this interpretation, the scarcity of high-fidelity data affects only the *variance* of the estimator, while its mean remains unchanged under the standard Monte Carlo sampling scheme.
>
> We have revised Assumption 5 and the surrounding text to make this precise:
>
> * We now write Assumption 5 explicitly as an assumption on the high-fidelity policy gradient estimator (rather than “high-fidelity policy gradient estimate,” as in the original version).
>
> * We have added the following clarification paragraph below the assumption: "We note that Assumption 5 is an assumption on the underlying high-fidelity policy gradient estimator; we do not require any policy gradient estimates computed from finite samples to be exact. In particular, the classic Monte Carlo REINFORCE policy gradient estimator considered in this work satisfies this requirement (Sutton et al., 1999). Extending our analysis to actor–critic-style estimators is nontrivial and is discussed as a limitation and direction for future work in Section F."
>
> We appreciate the reviewer’s careful reading and for highlighting this important nuance.
>
> ```
> 5. Unbiased, variance-reduced estimator: "The authors comment that they "construct an unbiased, variance-reduced estimator" in the abstract part, but there is no corresponding theoretical analysis of these properties. There is also no empirical evidence on the unbiasedness. The authors should fix such statements."
> ```
>
> We thank the reviewer for pointing this out. We have revised the manuscript to formally establish these properties by adding a dedicated result, **Lemma 1** in Section 4, together with its proof in **Appendix B**. Lemma 1 shows that the multi-fidelity control variate estimator is unbiased with respect to the high-fidelity estimator for any coefficient $c$, and that, for the optimal choice $c^*$, its variance is reduced by a factor of $(1-\rho^2)$ relative to the high-fidelity estimator, where $\rho$ is the Pearson correlation coefficient between the high- and low-fidelity random variables. This directly justifies the claim in the abstract that we “construct an unbiased, variance-reduced estimator.”
>
> Regarding “empirical evidence on the unbiasedness,” since unbiasedness is an expectation-level property of the estimator, it is therefore most naturally established analytically rather than via finite-sample experiments, especially on complex continuous-control benchmarks where the true policy gradient is unavailable. Our empirical results instead focus on demonstrating the consequences of this estimator, namely the variance-reduction and performance benefits predicted by Lemma 1.

---

> ### Author Response · Authors · 2025-12-16
> **Authors' response [Part 4]**
>
> ```
> 6. Extension to off-policy learning: "The authors comment that "To isolate the effect of their multi-fidelity mechanisms from the backbone algorithm and enable fair comparison, we adapt both PAR and DARC to the REINFORCE backbone". It raises questions on whether the proposed MFPG framework can be extended to the scenario when SAC is adopted as the backbone. The authors do not need to conduct experiments on this, but should add some discussions either in the main text or the appendix."
> ```
>
> We agree that the potential to extend MFPG to broader RL algorithms and scenarios is important to discuss. In the revised manuscript, we have added a dedicated section, **Appendix F: Extending MFPG to Broader Algorithms and Settings**, which discusses in more detail how the proposed multi-fidelity control variate approach might be extended to on-policy actor–critic, model-based, off-policy, and offline RL settings.
>
> In particular, prior off-dynamics RL methods such as DARC and PAR adopt SAC as their backbone, mainly in off-policy and offline reinforcement learning settings. We discuss potential extensions of MFPG to offline settings in our response to Comment 2 and in Appendix F. For off-policy learning, Appendix F discusses the following points:
>
> * MFPG could, in principle, be adapted relatively directly to importance-sampling-based off-policy policy gradient estimators, e.g., by replacing the high- and low-fidelity random variables with off-policy, importance-weighted policy gradient estimators. In such settings, variance reduction is arguably even more critical than in the on-policy policy gradient setting considered in this work.
>
> * Extending an MFPG-type approach specifically to SAC is less direct and would require developing multi-fidelity control variate techniques for off-policy temporal-difference (TD) learning. Unlike in on-policy policy gradient settings, where high-variance Monte Carlo returns motivate sophisticated variance-reduction techniques, the one-step TD learning and experience replay used by SAC tend to produce relatively low-variance value targets. The main challenges for SAC-type algorithms are primarily bias-related (e.g., value overestimation, distribution shift, and bias accumulation over long horizons). Even so, we believe that developing multi-fidelity control variate techniques for off-policy TD algorithms might still help reduce noise in the bootstrapped targets and in the gradient estimates used in the policy extraction step, and we highlight this as an interesting direction for future work.
>
>
> ```
> 7. Sensitivity to the moving-average coefficient: "Figure 9 shows the parameter study of the introduced hyperparameter `\eta_{ma}` for the moving average. It turns out that the MFPG framework can be sensitive to different choices of `\eta_{ma}`, while the authors comment on Page 10 that "and found the performance to be insensitive to this choice". This should be clarified and fixed if necessary."
> ```
>
>
> We thank the reviewer for noting this. We agree that our previous wording (“insensitive”) was too strong, and we have revised the text to remove such statements and characterize the dependence on $\eta_{\mathrm{ma}}$ more precisely.
>
> * In Section 6.1, we now state: "We tuned only the exponential moving-average weight $\eta_{\mathrm{ma}}$ in Eq. (3) and found that MFPG’s median performance consistently exceeded that of the High-Fidelity Only baseline across the tested values of $\eta_{\mathrm{ma}}$, even though the absolute performance metrics varied with $\eta_{\mathrm{ma}}$."
>
>
> * In addition, we strengthened the $\eta_{\mathrm{ma}}$ parameter study in Section 6.5 by running 20 random seeds and reporting two-sided 95\% bootstrap confidence intervals for the mean improvement of MFPG relative to High-Fidelity Only. We now summarize the result as follows: "Overall, across the tested values of $\eta_{\mathrm{ma}}$, MFPG is often statistically significantly better than High-Fidelity Only, and we do not observe statistically significant degradation."
>
>
> ```
> Additional comment: "There are some typos in the manuscript, e.g., On page 5, `\gamma` \in (0, 1]?"
> ```
>
>
> We thank the reviewer for pointing this out. In the revised manuscript, we now explicitly define both environments as finite-horizon MDPs (Section 3) with horizon $T$ and discount factor $\gamma \in [0, 1]$. This clarifies that we work with finite-horizon problems and also allow the myopic case $\gamma = 0$. We have also verified that this notation is used consistently throughout.
>
>
> ```
> Additional comment: "On page 8, 'goodness'"
> ```
>
> We thank the reviewer for catching this wording issue. We have changed "goodness" to "quality" on page 8 and performed an additional proofreading pass to identify and correct similar instances.
>
> We again thank the reviewer for their detailed, helpful, and encouraging feedback, and we would be happy to address any further questions.

---

> > ### Comment · Reviewer_bHiF · 2025-12-16
> >
> > Thank you for the detailed rebuttal and the revision. My concerns are addressed, and I maintain my positive recommendation of this paper. A minor thing is that the main text now seems to be a bit long. I understand that it is a bit difficult to satisfy all reviewers within a short main text, but it is still better to make the main text more concise. I would suggest that the authors move some content to the appendix. Please consider this when you have thorough discussions with other reviewers. Overall, this is a good work!

---

> > > ### Author Response · Authors · 2025-12-16
> > > **Authors' response**
> > >
> > > Thank you for the follow-up and your positive recommendation! We’re glad the revision addressed your concerns. We agree the main text has become longer after incorporating the revisions; in the next pass we will tighten the presentation and move secondary material to the appendix to improve concision and readability. Thank you again for the helpful suggestion.

---

### Review · Reviewer_hz18 · 2025-12-06

**Summary Of Contributions:**

# Summary

The paper _A Multi-Fidelity Control Variate Approach for Policy Gradient Estimation_ introduces a new variance reduction technique for policy gradient methods when (1) a simulator of the environment is available and (2) data can be gathered from this simulator cheaply. This variance reduction technique uses correlated trajectory samples between the simulator and environment to construct control variates to reduce the variance of the policy gradient proportionally to a function of the Pearson correlation between these trajectories (Equation (1)). The paper studies this variance reduction approach applied to REINFORCE, calling the resulting algorithm the _Multi-Fidelity Policy Gradient_ (MFPG). The paper provides both theoretical justification for MFPG, including convergence to first order stationary points, as well as an empirical investigation of MFPG on a number of continuous-control problems.

**Additional Comments:**

# Small Things

These did **not** affect the scoring of the paper.

1. In Figures 3 and 4, the colours for *DARC* and *Low-Fidelity Only (100⨉)* are hard to distinguish
2. In the caption to Figure 3, the following statement is made
   > Under large dynamics gaps, all multi-fidelity approaches converge toward the High-Fidelity Only baseline performance

   The term *converge* is a bit loaded, and is likely not what is meant here.
3. Section 6.2: "... that also use state-value function subtraction an an effort to variance reduction" → "... that also use state-value function subtraction an an effort to _reduce variance_"
4. The term _performance collapse_ is used before it is fully defined
5. The symbol $X_{\tau^l}^{\pi_\theta}$ is confusing. At the outset, it is remarkably similar to the symbol $X_\tau^{\pi_\theta}$ which is defined as $X_\tau^{\pi_\theta} \doteq \frac{1}{T}\sum_{t=0}^{T-1} G_t \log \pi_\theta(a_t \mid s_t)$. I assumed that $X_{\tau^l}^{\pi_\theta}$ was exactly this value for the low-fidelity simulator, which seems to be incorrect since later $X_{\tau^l}^{\pi_\theta}$ is more or less defined as an arbitrary, trajectory-dependent control variate (paragraph 2 of section 4), which is very confusing. For example, later $X_{\tau^l}^{\pi_\theta}$ has the exact form of $X_\tau^{\pi_\theta}$, except that the advantage function of $\pi_\theta$ in the environment (not the low-fidelity simulator) is used, which is clearly different from the advantage function of $\pi_\theta$ in the low-fidelity simulator.
6. Some of the reported results are a bit surprising, without much discussion. For example, in Figure 3, we see many of the algorithms attaining higher performance than the _More High-Fidelity Data (15x)_ algorithm. In the top two subplots, we see that often the _Low-Fidelity Only (100x)_ algorithm is competitive to the _More High-Fidelity Data (15x)_ baseline. This is quite a surprising result since the _More High-Fidelity Data (15x)_ algorithm was considered as the _oracle_ baseline, outlining the performance we would like to achieve if we had access to lots more data. The paper would likely benefit from a more in-depth discussion of this.

**Audience:**

Yes

**Audience Explanation:**

The paper presents a novel method for variance reduction in policy gradient, which is of interest to much of the audience of TMLR.

**Claims And Evidence:**

No

**Claims Explanation:**

# Main Argument

Overall, the paper motivates the use of low-fidelity data for variance reduction in policy gradient well. The developed algorithm is sensible and theoretically justified by providing improved convergence rates over REINFORCE to a first order stationary point with correlated data between the environment and low-fidelity simulator. Yet, I am concerned about the empirical study of the MFPG algorithm and that the claims made in the paper are not supported by rigorous empirical evidence.

**Too few random seeds**: My main concern is that only 5 experimental repetitions were used throughout the empirical study, which is likely far too few for a rigorous statistical characterization of the algorithms considered [1,2,3]. That reinforcement learning (RL) algorithms exhibit high variance performance distributions is well-known, and recent work has shown that performance measures based on only a few experimental repetitions can be erroneous or confidently incorrect [1,2]. I expect that the reported results would change with more experimental repetitions.

The paper also concedes to this. On page 10, paragraph **Result reporting**, the paper states

> Since five-seed results remain noisy... we report median, maximum, and minimum values.

I would point out that simply reporting *different* performance estimates does not circumvent the problems inherent to using few experimental repetitions. In fact, it is not entirely unlikely that these reported statistics are not representative of the true values of the underlying performance distributions. This complicates an accurate evaluation of the claims made in the paper. For example, in the discussion relating to Figure 3, multiple claims are made regarding the median performance of MFPG and other variance-reduction methods in comparison to each other and to the High-Fidelity Only baseline. But, without an accurate estimate of the true median performance of all these algorithms, these claims are difficult to verify.

**Statistical confidence is not reported and reported errors bars overlap**:  The paper does not report any notion of statistical confidence, likely due to the use of only a few experimental repetitions. Without any measures of statistical confidence, deciphering statistically significant differences in algorithm performance is difficult if not impossible, complicating an accurate evaluation of the presented experimental results.

Instead, the paper replaces notions of statistical confidence with maximum and minimum performance. It is not entirely unlikely that the Student-T confidence intervals for the data presented are of comparable magnitude to the reported max/min regions, and these kinds of confidence intervals would likely be optimistic estimates of confidence [1]. Assuming this is true, the paper then makes multiple claims where estimates of statistical confidence overlap. Even if this assumption is not true, the paper makes claims where the reported performance distributions do overlap, often severely.

I will list a few examples next, noting that in all these examples, the performance distributions overlap often severely throughout the experiment. This perhaps indicates no statistically significant difference among algorithms, though this could be remedied with more experimental repetitions.

---

**Example 1** On page 12, in relation to Figure 3, as noted above.

**Example 2** On page 13, in discussion of Figure 5:

> the median performance gains of MFPG over High-Fidelity Only are greater when the correlation is higher (smaller gap).

In this figure, the reported performance distributions overlap nearly entirely throughout the experiment. Further, it actually seems that the benefit of MFPG over High-Fidelity Only is greater under 2x gravity (lower correlation) than 0.8x gravity (higher correlation), which seems to contradict the statement above. With more random seeds and more rigorous notions of statistical confidence, the relationship between performance and dynamics shift for these algorithms will likely be clearer.

**Example 3** On page 16, in discussion of Figure 10:

> when the dynamics gap is mild... reparameterization substantially accelerates learning

But again, we see that the performance distributions overlap at multiple point at the beginning and throughout the experiment.

**Example 4** On page 16, in discussion of Figure 11:

> In practice, we observed that periodically reconstraining (the correlated) low-fidelity rollouts back to their high-fidelity counterparts can strengthen correlation and improve performance

But again, the performance distributions overlap severely throughout the experiment.

---

While I recognize that the use of relatively few seeds is a common practice in deep RL, this norm does not reflect scientific rigour. I would urge the community－and this paper in particular－to aim to provide rigorous empirical evaluations of algorithms with high statistical confidence. My recommendation is:
1. Report rigorous statistical characterizations of confidence for which all assumptions are satisfied.
2. Use many more experimental repetitions (e.g. ≥ 20) so that the reported levels of confidence are accurate
Bootstrap confidence intervals tend to be a good choice for (1), as they make no assumptions on the distribution of data (unless your dataset satisfies the assumptions of another confidence measure).


# References

[1] Andrew Patterson, Samuel Neumann, Martha White, Adam White. _Empirical Design in Reinforcement Learning_. JMLR, 2024.

[2] Cédric Colas, Olivier Sigaud, Pierre-Yves Oudeyer. _How Many Random Seeds? Statistical Power Analysis in Deep Reinforcement Learning Experiments_. arXiv preprint, 2018.

[3] Peter Henderson, Riashat Islam, Philip Bachman, Joelle Pineau, Doina Precup, David Meger. _Deep Reinforcement Learning that Matters_. AAAI, 2018.

**Requested Changes:**

I would be happy to re-evaluate my score if my concerns in my Main Argument above were addressed. In summary, I am concerned that the empirical study is lacking statistical rigour.

The paper would need to provide a more rigorous statistical characterization of the MFPG algorithm, including increasing the number of experimental repetitions and reporting methods of statistical confidence for which distributional assumptions are satisfied (e.g. bootstrap confidence intervals). This could also include reporting more aggregated performance statistics, as the paper reports when analyzing the aggregate performance collapses across all experimental settings.

---

> ### Author Response · Authors · 2025-12-16
> **Authors' response [Part 1]**
>
> We thank the reviewer for the rigorous, detailed, and constructive feedback. The critique regarding the number of seeds and statistical confidence was well founded. We have carefully revised our empirical evaluation to address these concerns, specifically by re-running our experiments with 20 random seeds and adopting nonparametric bootstrap confidence intervals for all statistical analysis. We believe the reviewer’s feedback has substantially strengthened the empirical evidence supporting our claims. In the revision, we made the following concrete changes (all modifications are highlighted in blue in the revised manuscript).
>
>
> 1. Major Update: Statistical Rigor (20 Random Seeds & Bootstrap Confidence Intervals)
>
> ```
> Review comment: "My main concern is that only 5 experimental repetitions were used throughout the empirical study, which is likely far too few for a rigorous statistical characterization of the algorithms ... 1. Report rigorous statistical characterizations of confidence for which all assumptions are satisfied. 2. Use many more experimental repetitions (e.g. ≥ 20)."
> ```
>
> We have made the following changes:
>
> * **More random seeds**: We increased the number of runs for the baseline comparisons (Section 6.3), the negated-reward experiment (Section 6.4), and the additional experiments and ablation studies (Section 6.5) to 20 random seeds per method per task setting.
>
> * **Statistical confidence**: As suggested, we now report two-sided 95% nonparametric bootstrap confidence intervals (R = 10,000 resamples over seeds/runs) for the means of our primary metrics (final return and AUC).
>
> * **Significance testing**: To verify our claim, for the baseline comparisons in Section 6.3 we bootstrap the mean improvement of each approach relative to the High-Fidelity Only baseline ($\Delta$ = mean(method) - mean(High-Fidelity Only)). We declare a statistically significant improvement over High-Fidelity Only at $\alpha = 0.05$ when the 95% confidence interval for $\Delta$ lies strictly above 0.
>
>
> * **Figures and text**: Figures that previously used max/min bands have been updated to show bootstrap confidence intervals (error bars for bar plots and shaded regions for learning curves), and we revised the surrounding discussions to only make statements that are supported by significance testing (including the discussions around Figures 3, 5, 10, and 11 that the reviewer listed as examples).
>
> * **Empirical claim**: We refined the empirical takeaway (Claim/Insight I2). With the strengthened statistical analysis, our key finding holds and we have revised our claim (ii) accordingly (I2, Reliability and robustness):
>
> MFPG provides a reliable and robust way to exploit low-fidelity data. In our baseline comparison experiments in Section 6.3, for scenarios where low-fidelity data are neutral or beneficial (Hopper-gravity, Walker2d-friction) and dynamics gaps are mild to moderate (0.5× to 2.0×), MFPG is, among the evaluated multi-fidelity and low-fidelity-only approaches, *the only method* that consistently and statistically significantly improves the mean performance over a baseline trained solely on high-fidelity data in *8 out of 8* scenarios. When low-fidelity data become harmful, MFPG exhibits the strongest *robustness* against performance degradation among the evaluated multi-fidelity and low-fidelity-only methods. Strong off-dynamics RL methods tend to exploit low-fidelity data more aggressively, and can achieve higher performance than MFPG at times (e.g., in Walker2d) but can also fail substantially more severely (e.g., in HalfCheetah).
>
> * **Aggregated statistics and robustness**: Following the reviewer’s suggestion, we retained and clarified the cross-setting robustness summary (performance collapses) and use it as a complementary aggregated metric alongside per-setting plots.
>
> These changes primarily affect Section 6 (especially Sections 6.1, 6.3, 6.4, and 6.5, with Section 6.3 most substantially), with small adjacent updates in the Abstract/Introduction/Conclusion to match the refined empirical statement.

---

> ### Author Response · Authors · 2025-12-16
> **Authors' response [Part 2]**
>
> 2. Addressing Additional Comments
>
> ```
> * Review comment: "In Figures 3 and 4, the colours for DARC and Low-Fidelity Only (100⨉) are hard to distinguish."
> ```
>
> We have updated the color of the Low-Fidelity Only (100×) baseline to make it clearly distinguishable, and we apologize for the earlier confusion.
>
> ```
> * Review comment: "In the caption to Figure 3, the following statement is made
>
> Under large dynamics gaps, all multi-fidelity approaches converge toward the High-Fidelity Only baseline performance
>
> The term converge is a bit loaded, and is likely not what is meant here."
> ```
>
> We agree “converge” is too loaded; we have removed the use of the term “converge” in this context and replaced it with more precise wording.
>
> ```
> * Review comment: "Section 6.2: "... that also use state-value function subtraction an an effort to variance reduction" → "... that also use state-value function subtraction an an effort to reduce variance"
> ```
>
> We have corrected the phrasing from “variance reduction” to “reduce variance,” and we thank the reviewer for this suggestion.
>
>
> ```
> * Review comment: "The term performance collapse is used before it is fully defined"
> ```
>
> We have removed all uses of the term “performance collapse” prior to its formal definition in Section 6.1.
>
> ```
> * Review comment: "The symbol $X^{\pi_{\theta}}_{\tau^l}$ is confusing. At the outset, it is remarkably similar to the symbol $X^{\pi_{\theta}}_{\tau}$ which is defined as ... which is clearly different from the advantage function of $\pi_{\theta}$ in the low-fidelity simulator."
> ```
>
> We apologize for the confusion and agree that the original presentation could be confusing. Our intention is that $X_{\tau^{l}}^{\pi_{\theta}}$ denotes the same trajectory functional as
> $X_{\tau}^{\pi_{\theta}}$, but evaluated on trajectories sampled from the low-fidelity environment $\tau^l \sim \mathcal{M}^l(\pi_{\theta})$. In particular, if we used a vanilla REINFORCE algorithm without a state-baseline subtraction from the Monte Carlo returns, i.e., $X_{\tau}^{\pi_{\theta}} =  \frac{1}{T} \sum_{t=0}^{T-1} G_{t} \log \pi_{\theta}(a_{t} | s_{t})$, then $X_{\tau^{l}}^{\pi_{\theta}}$ would match $X_{\tau}^{\pi_{\theta}}$ exactly in form.
>
>
> In our implementation of state-baseline-subtracted REINFORCE, using a shared state-value function as the baseline for both high- and low-fidelity Monte Carlo returns is an implementation choice that simplifies training. We have revised the introduction around this choice of a shared state-value function and added clarification in Section 4 (last paragraph):
>
>
> We sample trajectories from the high- and low-fidelity environments to compute Monte Carlo returns $G^h_t$ and $G^l_t$, respectively. We then use a shared value function $V_{\phi}$ learned from high-fidelity samples $(s^h_t, a^h_t, r^h_t, s^h_{t+1})$ to compute state-value baselines that are subtracted from both $G^h_t$ and $G^l_t$. This is an implementation choice that simplifies training; moreover, subtracting any state-dependent baseline does not change the expectation of the REINFORCE gradient estimator (Williams, 1992). Alternatively, one can train separate value functions for the high- and low-fidelity Monte Carlo returns.
>
> Reference:
>
> Ronald J Williams. Simple statistical gradient-following algorithms for connectionist reinforcement learning. Machine learning, 8:229–256, 1992.

---

> ### Author Response · Authors · 2025-12-16
> **Authors' response [Part 3]**
>
> ```
> * Review comment: Some of the reported results are a bit surprising, without much discussion. For example, in Figure 3, we see many of the algorithms attaining higher performance ... The paper would likely benefit from a more in-depth discussion of this.
> ```
>
> We thank the reviewer for pointing this out. In the revised manuscript, around the discussion of Fig. 13 in Appendix D, we have added the following explanation of this phenomenon:
>
> In Fig. 13, we observe that, in general, High-Fidelity Only struggles to match the performance of the More High-Fidelity Data (15×) baseline, highlighting the clear benefit of training on larger high-fidelity sample sizes. Nevertheless, in some settings Low-Fidelity Only (100×) can match or even exceed the More High- Fidelity Data (15×) baseline, albeit often with substantially large cross-seed variance (e.g., Hopper–gravity 1.2×). This does not contradict the role of the More High-Fidelity Data (15×) baseline as a strong reference point: in certain regimes, the dynamics mismatch (bias) can be benign, and abundant (100×) low-fidelity interaction can provide learning signals that transfer unusually well to the high-fidelity environment. In such cases, aggressively exploiting the low-fidelity data can produce large gains, but this behavior is brittle—when low-fidelity data become harmful (cf. HalfCheetah), these approaches can degrade substantially. Moreover, the benefit of low-fidelity data is difficult to predict a priori (e.g., Low-Fidelity Only does not vary monotonically with the dynamics gap). In contrast, MFPG provides a more consistent and reliable mechanism for leveraging low-fidelity data by anchoring updates to high-fidelity gradients while using low-fidelity data only as a variance reduction tool.
>
> We thank the reviewer again for emphasizing statistical rigor and clarity. The revised experiments (20 seeds and bootstrap confidence intervals) and the updated textual claims substantially strengthen the evidence supporting our empirical conclusions. We would be happy to address any further questions.

---

> ### Comment · Reviewer_hz18 · 2025-12-19
> **Reviewer Response**
>
> # Main Response
>
> I would like to thank the authors for improving the statistical rigour of the empirical study presented in the paper, and for updating the claims made in the paper to be measured and accurate, with more clear supporting evidence. Overall, my main concerns about the paper have been mostly addressed.
>
> My only remaining slight concern is that the following claim at the end of page 15, paragraph **MFPG performance vs. correlation** is still a bit too strong based on the data presented in Figure 5
>
> > ...in both cases [of having smaller or moderate dynamics gaps], MFPG trends above High-Fidelity Only for much of training, although the bootstrap confidence intervals overlap during some portions
>
> My main concern with this statement is that since the confidence intervals do overlap fairly significantly during training for the moderate dynamics shift (2.0×), the statement may not accurately characterize the true mean performance of these algorithms on the considered environment. Can more experimental repetitions be obtained for this experiment to verify the claim? Or, do different metrics such as the AUC outline the difference in performance between MFPG and High-Fidelity Only more clearly?
>
> # A Few Final Points
>
> These points serve only to improve the submission, and they do **not** influence my evaluation of the paper.
>
> **Typo** I noticed a typo at the end of section 6.4, on the last line of page 16. The paper reads
>
> > These results support our key insight ***I2***
>
> When the paper should read:
>
> > These results support our key insight ***I3***
>
>
> **Confusion on what reconstraining is** On page 19, paragraph **Variance-bias trade-off**, the paper reference reconstraining the low-fidelity rollouts back to their high-fidelity counterparts. But, how these rollouts are reconstrained is not completely obvious, and it might be beneficial to better explain how this is accomplished

---

> ### Author Response · Authors · 2025-12-19
> **Authors' response**
>
> We thank the reviewer for the thoughtful follow-up. We appreciate your acknowledgement that the revised empirical analysis has mostly addressed the main concerns. We have made three additional changes in response to your remaining comments:
>
> ```
> 1. Figure 5, overlapping bands: "My main concern with this statement is that since the confidence intervals do overlap fairly significantly during training for the moderate dynamics shift (2.0×), ... do different metrics such as the AUC outline the difference in performance between MFPG and High-Fidelity Only more clearly?"
> ```
>
> **Figure 13 (Appendix D)**: We agree that additional evidence is needed to support our statement about MFPG versus High-Fidelity Only in these settings. To address the interpretability issue caused by overlapping bands in Fig. 5, we additionally plot, for the same settings, the mean difference in evaluation return between MFPG and High-Fidelity Only---computed as $\Delta$ = mean(MFPG) - mean(High-Fidelity Only)---in Figure 13 (Appendix D), together with two-sided 95% bootstrap confidence intervals for $\Delta$ throughout training.
>
> We added the following text description after presenting Fig. 5 (Sec.6.3): "Figure 5 shows both algorithms’ performance for completeness. Figure 13 further reports the mean difference in evaluation return between MFPG and High-Fidelity Only, computed as $\Delta$ = mean(MFPG) - mean(High-Fidelity Only) ( $\Delta = 0$ indicates no difference). For mild (0.8×) and moderate (2.0×) dynamics gaps, the confidence intervals are often above zero, suggesting improved performance, with larger gains typically observed in the mild-gap case. Under the extreme shift (5.0×), the confidence intervals largely span zero. The AUC results in Fig. 12 are consistent with these trends."
>
>
> ```
> 2. Typo in Sec. 6.4 (“I2” → “I3”).
> ```
> We thank the reviewer for catching this. We corrected the sentence to read “These results support our key insight I3.”
>
> ```
> 3. Clarification on “reconstraining” in Sec. 6.5.
> ```
>
> We agree that “reconstraining” was under-specified. We have added a brief, concrete description to explain this: "In practice, we observed that periodically reconstraining (the correlated) low-fidelity rollouts back to their high-fidelity counterparts can strengthen correlation and accelerate learning in some settings (i.e., every few steps along a low-fidelity rollout, reset the low-fidelity state to match the state of the correlated high-fidelity rollout at the same time step)."
>
> We thank the reviewer again for the careful reading and suggestions, which helped further improve the rigor and clarity of the presentation.

---

### Decision · Action_Editor_5s6v · 2026-01-14

**Recommendation:** Accept with minor revision

**Additional Comments:**

Revision noted above.

**Audience:**

Yes

**Audience Explanation:**

This paper seems very relevant for real-world RL. The goal is to build algorithms that make use of data generated from a cheap, low-fidelity simulator with a small amount of high-fidelity data. The proposed policy gradient method works well across settings, has relatively few hyperparameters, and seems more robust to misspecification or errors in the low-fidelity simulator.

**Claims And Evidence:**

Yes

**Claims Explanation:**

Intially, the submitted paper had issues because the number of seeds used was too low, measures of confidence overlapped, and signs of overclaiming (as one reviewer pointed out stating limitations and future work like contributions). The authors have rectified all these issue to the extent that all three reviewers were happy. This paper is correct.

One last thing to do, as flagged by reviewer KL6A:
> The only small issue I still have is that claim (ii) still mostly claims a comparison to "other multi-fidelity methods tested" rather than the more precise "off-dynamics RL methods tested" (by the authors' own terminology this is the subset of MFRL methods they compared to). They mention off-dynamics RL in only the last sentence of claim (ii).